# ODRL: A Benchmark for Off-Dynamics Reinforcement Learning

**Jiafei Lyu**[1][*]   **Kang Xu**[2]   **Jiacheng Xu**[3]   **Mengbei Yan**[1]   **Jingwen Yang**[2]
**Zongzhang Zhang**[3]   **Chenjia Bai**[4][†]   **Zongqing Lu**[5][†]   **Xiu Li**[1][†]

[1]Tsinghua Shenzhen International Graduate School, Tsinghua University
[2]Tencent
[3]National Key Laboratory for Novel Software Technology, Nanjing University
[4]Institute of Artificial Intelligence (TeleAI), China Telecom
[5]School of Computer Science, Peking University
`lvjf20@mails.tsinghua.edu.cn, li.xiu@sz.tsinghua.edu.cn`

## Abstract

We consider off-dynamics reinforcement learning (RL) where one needs to transfer policies across different domains with dynamics mismatch. Despite the focus on developing dynamics-aware algorithms, this field is hindered due to the lack of a standard benchmark. To bridge this gap, we introduce ODRL, the first benchmark tailored for evaluating off-dynamics RL methods. ODRL contains four experimental settings where the source and target domains can be either online or offline, and provides diverse tasks and a broad spectrum of dynamics shifts, making it a reliable platform to comprehensively evaluate the agent's adaptation ability to the target domain. Furthermore, ODRL includes recent off-dynamics RL algorithms in a unified framework and introduces some extra baselines for different settings, all implemented in a single-file manner. To unpack the true adaptation capability of existing methods, we conduct extensive benchmarking experiments, which show that no method has universal advantages across varied dynamics shifts. We hope this benchmark can serve as a cornerstone for future research endeavors. Our code is publicly available at https://github.com/OffDynamicsRL/off-dynamics-rl.

## 1 Introduction

Human beings are able to transfer the policies swiftly to a structurally similar task. This ability is also expected in decision-making agents, especially embodied AI [62, 13, 53]. For instance, we may train the robot in a simulated environment (i.e., *source domain*) and deploy the learned policy in real-world tasks (i.e., *target domain*), where the dynamics gap may pertain between the simulation and reality. It is anticipated that the robot is able to adapt itself to real-world dynamics quickly.

How to generalize policies across different domains with dynamics discrepancies efficiently remains an open problem in reinforcement learning (RL). Such a problem setup is referred to as *off-dynamics RL* in [16], but it is restricted in demanding the online source domain and target domain. We extend its scope and formally define a general off-dynamics RL setting (Definition 1), where the source domain and the target domain only differ in their transition dynamics. Existing researches realize policy adaptation under dynamics mismatch via system identification [82, 10], domain randomization [63, 66, 55], learning domain classifiers [16, 41], etc. However, we argue that this field lacks a standard and unified benchmark. Upon checking the latest off-dynamics RL methods [16, 76, 41], we found that they often manually construct their customized environments with dynamics shifts and

---

[*]Work done while working as an intern at Tencent IEG. [†] Corresponding Authors.

38th Conference on Neural Information Processing Systems (NeurIPS 2024) Track on Datasets and Benchmarks.

conduct experiments on them. The superior performance reported in these papers does not necessarily indicate that they are indeed state-of-the-arts, due to the lack of comprehensive evaluations and comparison using a unified testbed. Eventually, the progress in off-dynamics RL can be impeded.

To mitigate this concern, we introduce ODRL, the first benchmark for off-dynamics RL. ODRL covers numerous domain categories, including locomotion, navigation, and dexterous manipulation, and a wide spectrum of varied dynamics shifts, e.g., kinematic and morphology mismatch, as depicted in Figure 1. Our benchmark provides 4 types of experimental settings in a unified framework (see Figure 2), where the source and target domain can be either online or offline. Such versatility guarantees a thorough and trustworthy assessment of off-dynamics RL algorithms under different conditions.

Furthermore, ODRL encompasses a variety of recent off-dynamics RL algorithms within diverse experimental configurations, in conjunction with some baseline methods proposed on our own. All algorithms share similar code styles and are implemented in separate single files, making it easier to recognize core algorithmic designs and performance-relevant details. It also ensures consistency and allows for a fair benchmarking comparison. Notably, only a limited budget of target domain data is permitted in ODRL to better evaluate the policy adaptation efficiency of the agent.

To summarize, we have made the following efforts: (a) we give a formal definition of the general off-dynamics RL setting; (b) we propose the first off-dynamics RL benchmark, which offers different experimental settings, diverse task categories, and dynamics shift types under a unified framework; (c) we isolate algorithm implementations into single files to facilitate a straightforward understanding of the key algorithmic designs; (d) we conduct extensive experiments to investigate the performance of existing methods under different dynamics shifts and experimental settings, and conclude some key observations and insights. It is our hope that ODRL can aid researchers in developing dynamics-aware methods and pave the way for adaptable algorithms in real-world applications.

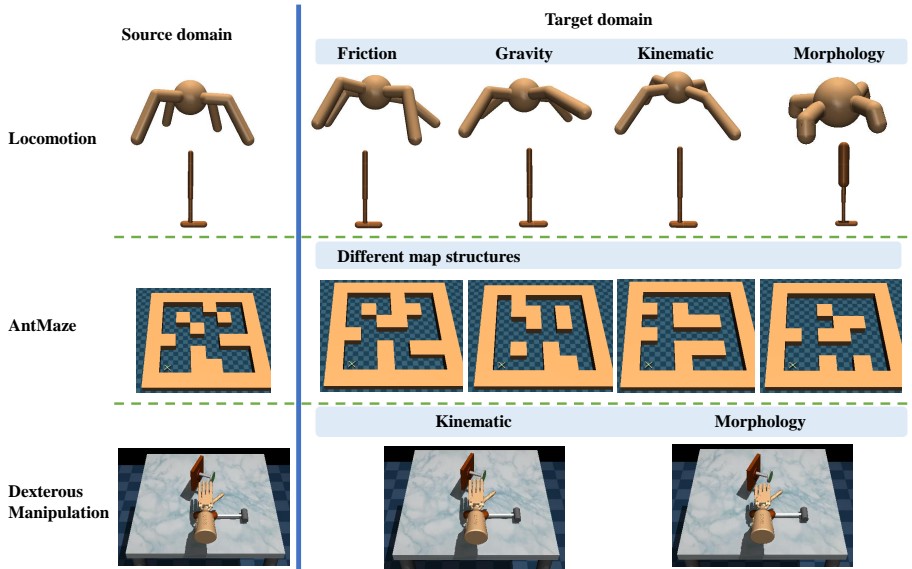

Figure 1: **An overview of selected benchmark tasks.** ODRL includes multiple domains with various types of dynamics shifts, making it a reliable platform for evaluating policy adaptation ability.

## 2 Background

**Reinforcement Learning (RL).** We study sequential decision-making problems in a Markov Decision Process (MDP), which is specified by the tuple $\langle \mathcal{S}, \mathcal{A}, P, r, \rho_0, \gamma \rangle$, where $\mathcal{S}, \mathcal{A}$ are state space and action space, respectively, $P(s'|s,a) : \mathcal{S} \times \mathcal{A} \to \Delta(\mathcal{S})$ is the transition probability, where $\Delta$ is the probability simplex, $r : \mathcal{S} \times \mathcal{A} \to \mathbb{R}$ is the scalar reward signal, $\gamma \in [0,1)$ is the discount factor, $\rho_0$ is the initial state distribution. The goal of RL is to find a policy $\pi(\cdot|s)$ to maximize $J(\pi) = \mathbb{E}_\pi[\sum_{t=0}^{\infty} \gamma^t r(s_t, a_t)]$.

**Off-dynamics RL.** In off-dynamics RL, we consider two infinite-horizon MDPs, the *source domain* $\mathcal{M}_{\text{src}} := \langle \mathcal{S}, \mathcal{A}, P_{\text{src}}, r, \rho_0, \gamma \rangle$ and the *target domain* $\mathcal{M}_{\text{tar}} := \langle \mathcal{S}, \mathcal{A}, P_{\text{tar}}, r, \rho_0, \gamma \rangle$. Note that the

two domains only differ in their transition probabilities, and other components like state space, action space, and reward functions are kept the same. Formally, we define our problem setting below.

> **Definition 1** (Off-dynamics RL setting). *The agent has access to sufficient data from the source domain $\mathcal{M}_{\mathrm{src}}$ and a limited budget of data from the target domain $\mathcal{M}_{\mathrm{tar}}$, where there exist dynamics shifts between $\mathcal{M}_{\mathrm{src}}$ and $\mathcal{M}_{\mathrm{tar}}$. The agent aims at getting better performance in the target domain $\mathcal{M}_{\mathrm{tar}}$ by leveraging data from both domains.*

## 3 Related Work

Generalizing policies across varied domains is a broad and critical topic in RL, where domains can differ in transition dynamics [16, 69, 77, 12], observation spaces or action spaces [22, 6, 23, 83, 4, 79, 25], etc. As a benchmark paper, it is difficult to include all possible domain discrepancies. We set our focus on the policy adaptation across domains with dynamics mismatch, because this kind of problem often occurs in real-world applications [53, 36, 1, 85], e.g., enabling the quadruped robot that previously trained under indoor scenes to adapt its policy to outdoor varied landscapes.

Previous studies mainly handle this challenge by domain randomization [63, 49, 70, 35], system identification [10, 86, 15, 74, 17, 87, 11, 60], and meta learning approaches [50, 58, 2, 73]. These methods often rely on a manipulable simulator [9], or require access to the distribution of training environments. System identification can be expensive due to the need of calibrating the dynamics of the source domain, and domain randomization relies on manually-specified randomized parameters and nuanced randomization distributions. Another line of research tries to address the issue from the perspective of designing dynamics-aware algorithms without changing the parameters of the training environment. Typical methods include recognizing the dynamics change by training expressive models [24, 31, 75], selectively sharing transitions from the source domain by contrastive learning [71] or the proximity of paired value estimate targets across the two domains [76], learning domain classifiers for measuring the dynamics gap to penalize source domain rewards [16, 41, 68] or performing importance weighting [52, 51], capturing the representation mismatch between two domains for reward modification [44], leveraging action transformation methods that utilizing the trained dynamics models of the two domains convert transitions from the source domain [27, 14]. Furthermore, some studies close the dynamics gap between two domains by using the expert demonstration from the target domain [37, 28, 18, 61]. Despite the success, existing papers often conduct experiments within self-proposed environments, which is unhealthy for the advances of this field because it fails to truly reveal the merits of the proposed method. Meanwhile, the constructed tasks often lack sufficient diversity, and different papers set focus on distinct settings (e.g., whether the source domain is offline). These motivate us to develop a unified and standard benchmark for off-dynamics RL.

**Comparison against other benchmarks.** There are numerous benchmarks that are related to ODRL, including Gym-extensions [29], D4RL [19], DMC suite [65], Meta-World [81], RLBench [32], CARL [5], Continual World [72]. We compare ODRL against these benchmarks below.

Table 1: **A comparison between ODRL and other RL benchmarks.**

| Benchmark | Offline Datasets | Diverse Domains | Multi-task | Single-task Dynamics Shift |
|---|---|---|---|---|
| D4RL [19] | ✔ | ✔ | ✘ | ✘ |
| DMC suite [65] | ✘ | ✔ | ✘ | ✘ |
| Meta-World [81] | ✘ | ✘ | ✔ | ✘ |
| RLBench [32] | ✔ | ✘ | ✔ | ✘ |
| CARL [5] | ✘ | ✔ | ✘ | ✔ |
| Gym-extensions [29] | ✘ | ✘ | ✔ | ✔ |
| Continual World [72] | ✘ | ✘ | ✔ | ✘ |
| ODRL (this work) | ✔ | ✔ | ✘ | ✔ |

Among shown in Table 1, D4RL only contains single-domain offline datasets and does not focus on the off-dynamics RL. DMC suite contains a wide range of tasks, but it does not offer offline datasets and does not handle the off-dynamics RL. Meta-world is designed for the multi-task RL setting.

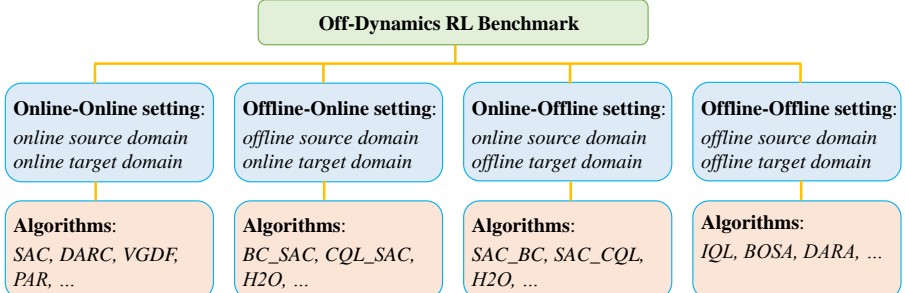

Figure 2: **Benchmark setting and algorithmic implementations.** Our benchmark involves 4 varied experimental settings, where the source domain and the target domain can be designated as online or offline. Numerous baselines and off-dynamics RL algorithms are implemented for each setting.

RLBench provides demonstrations for numerous tasks but it does not involve the dynamics shift in a single task. CARL focuses on the setting where the context of the environment (e.g., reward, dynamics) can change between different episodes (i.e., it does not have a source domain or target domain, but only one domain where the dynamics or rewards can change depending on the context). CARL also does not provide offline datasets. Continual World is a benchmark for continual learning in RL. It also supports multi-task learning and can be used for transfer RL policies. ODRL, instead, focuses on the setting where the agent can leverage source domain data to facilitate the policy training in the target domain, where the task in the source domain and the target domain remain identical. Note that Gym-extension involves some tasks that are similar to some MuJoCo tasks in ODRL, but Gym-extensions is intrinsically designed for multi-task scenarios and ODRL covers more diverse task domains, dynamics shift types and also provides offline datasets for each task.

## 4 Benchmark Details

ODRL mainly contains three task categories, including locomotion, navigation, and dexterous hand manipulation. Different domains are equipped with distinct and representative dynamics shift tasks. We treat the vanilla environments within these domains as the *source domain* and environments with dynamics shifts as the *target domain*. Both the source domain and the target domain in ODRL are allowed to be either online or offline, resulting in four varied training paradigms as illustrated in Figure 2, e.g., *Online-Offline* setting denotes that the source domain is online while the target domain is offline. The goal is to learn policies that can achieve better performance in the target domain by leveraging transition from the *single* source domain (with dynamics discrepancies) and *single* target domain. ODRL offers a diverse collection of tasks and covers all possible single-task policy adaptation settings, making it a reliable and unified benchmark for off-dynamics RL.

### 4.1 Benchmark Tasks

**Locomotion.** We adopt four tasks (Ant, Hopper, HalfCheetah, Walker2d) from the popular OpenAI Gym library [7], simulated by MuJoCo [67]. We consider four kinds of dynamics shifts:

(a) *Friction shift.* In MuJoCo, friction is represented by three components: static, dynamic, and rolling friction, where static friction means the frictional force that needs to be overcome when the robot is stationary, while dynamic friction and rolling friction stand for the frictional force between objects when they are in motion and rolling, respectively. Modifying the friction attribute can significantly change the motion characteristics of the simulated robots. We modify all friction components and consider shift levels $\{0.1, 0.5, 2.0, 5.0\}$, where the friction components of the target domain are set to be 0.1, 0.5, 2.0, and 5.0 times the friction components of the source domain to span across both slight shifts and serious shifts.

(b) *Gravity shift.* The gravity attribute corresponds to the gravitational force acting on objects within the simulation. We only modify the strength of the gravity and keep its direction unchanged. Similarly, we consider shift levels $\{0.1, 0.5, 2.0, 5.0\}$ where the gravity in the target domain is established at 0.1 times, 0.5 times, 2.0 times, and 5.0 times the gravity within the source domain to cover both minor shift cases and extreme shift cases.

(c) ***Kinematic shift.*** The kinematic discrepancies are realized by constraining the rotation angle ranges of certain joints in the simulated robot, i.e., some joints are broken such that it becomes infeasible to exhibit some motion characteristics in the target domain. We offer two choices of broken joints for each robot, which occur at varied parts of their body and are distinct across different robots since they possess varied structures and appearances, e.g., broken hips or broken ankles can occur in *ant* tasks, while the thigh joint or the foot joint can be broken for the *halfcheetah* task. For a specific type of kinematic shift, one can choose from three different shift levels (*easy*, *medium*, *hard*), where the joint's rotation angle range is restricted to distinct values.

(d) ***Morphology shift.*** We achieve the morphology shifts by modifying the size of specific limbs or torsos of the simulated robot, without altering the state space space and action space. We also provide two types of morphological change for each robot, along with three different shift levels (*easy*, *medium*, *hard*) for each type, where the body part in the target domain is revised to different sizes, depending on the specific task. For example, the leg size in the *ant* task can be 1/2 of its leg size in the source domain given a medium shift level.

Generally, ODRL involves 4 friction shift tasks, 4 gravity shift tasks, 6 kinematic shift tasks, and 6 morphology shift tasks for a single robot, resulting in a total of **80** tasks with dynamics mismatch in the locomotion domain. If one considers the source domain to be offline, different qualities of source domain datasets can incur a substantial number of tasks, e.g., given a fixed target domain, the source domain datasets can be medium or expert. We adopt the offline source domain datasets from D4RL [19], which contains 6 different types of offline datasets (random, medium, medium-replay, medium-expert, full-replay, expert), leading to **480** MuJoCo tasks for the **Offline-Online** setting in principle. Similarly, abundant tasks can be used for evaluation under other settings like **Online-Offline**.

**AntMaze.** The AntMaze domain requires navigating an 8-DoF Ant quadruped robot to the goal position within the maze. The morphologically sophisticated quadruped robot attempting to reach goals can mimic real-world navigation tasks. Following D4RL [19], we use a sparse 0-1 reward and a +1 reward is only assigned when the goal is reached. We employ three map sizes (*small*, *medium*, *large*), and construct 6 different map layouts for each map size (please see some examples in Figure 1), leading to an aggregate of **18** tasks. Note that the embodied Ant robot is unchanged, and only the map structures are modified, targeting examining the policy adaptation ability to varied landscapes or potential obstacles. The starting position and the *unique* goal position in the revised target map remain consistent with the original source domain map.

**Dexterous Manipulation.** We consider four tasks (*pen*, *door*, *relocate*, *hammer*) from Adroit [59] that is tailored for dexterous hand manipulation. It demands controlling a 24 DoF shadow hand robot to master tasks like opening a door, hammering a nail, etc. This domain is chosen because it resembles real-world hand manipulation tasks. It is challenging since it expects fine-grained hand operations, making it an ideal testbed for measuring the policy adaptation capability of the agent when encountering high-dimensional, sparse reward tasks. We do not alter the operated objects (e.g., the hammer), but modify the robotic hand to comprise the following two kinds of dynamics shifts:

(a) ***Kinematic shift.*** Akin to MuJoCo tasks, we simulate broken joints by limiting the rotation ranges of some hand joints, and there are numerous design choices to fulfill that due to the complexity of the hand robot. We modify rotation ranges of all finger joints in the index finger and the thumb, which should cause substantial troubles in completing tasks like twirling a pen, since it becomes much harder for the dexterous hand to grasp an object. ODRL involves three kinds of shift levels (*easy*, *medium*, *hard*) for individual tasks, where the rotation angle ranges of the index finger and thumb are set to be 1/2, 1/4, and 1/8 the rotation angle range of the paired source domain task, respectively.

(b) ***Morphology shift.*** We shrink the sizes of the proximal, intermediate, and distal phalanges in the index, middle, ring, and little fingers to realize the morphological mismatch. Despite that the thumb is unmodified, it is still very challenging to perform fine-grained manipulations under such shifts. We also provide three shift levels (*easy*, *medium*, *hard*) for each task where the phalanges sizes are configured as 1/2, 1/4, and 1/8 the phalanges sizes of the source domain robot hand, respectively.

For each task in Adroit, we offer 3 environments with kinematic shifts and 3 environments with morphology shifts, yielding a total of **24** tasks. Combined with the aforementioned tasks, it is evident that

ODRL contains a wide spectrum of dynamics shift tasks. Note that we consider these three domains because they are widely adopted in research-oriented RL papers [40, 43, 39, 20, 80, 47, 84, 78]. It is friendly for newcomers to quickly get familiar with the benchmark, develop new algorithms, and run experiments. The naming rule for benchmark tasks gives `[domain]-[shift_type]-[shift_part (optional)]-[shift_level]`, e.g., *ant-friction-0.5* denotes that the target domain has 0.5 times the friction components of the source domain, *hopper-kinematic-footjnt-medium* means that the foot joint in *hopper* robot is broken and the shift level gives medium. We defer the full list of ODRL task names, more task details, and visualization results to Appendix D.

## 4.2 Offline Datasets

Since ODRL allows the target domain with dynamics discrepancies to be offline, we provide target domain offline datasets with distinct qualities (*random*, *medium*, *expert*) for locomotion and dexterous manipulation tasks, where the medium datasets are gathered by an early-stopped RL algorithm[2] that has approximately one third or one half the performance of the expert policy. For the AntMaze domain, we only provide a mixing dataset for each map layout that contains both successful goal-reaching trajectories and unsuccessful ones. It is worth noting that we constrain the target domain offline dataset sizes (5000 transitions for locomotion and dexterous manipulation tasks, and 10000 samples for AntMaze tasks) owing to the fact that existing offline RL algorithms [39, 40, 21, 46, 20] can learn effective policies if a large amount of target domain data is available, potentially obviating the need of a source domain. It becomes difficult to train solely on the target domain dataset under such a low data regime. This is also reasonable in real-world scenarios, where collecting offline experiences can be expensive or time-consuming, and only limited data can be accessed, but sufficient data from another domain is available, e.g., a possibly biased simulator.

## 4.3 Evaluation Protocol

In ODRL, we suggest two metrics for evaluating the performance of the learned policy, the achieved return and its normalized score in the *target domain*. We do not care about the performance of the agent in the source domain. The normalized score (NS) is calculated by: $NS = \frac{J_\pi - J_r}{J_e - J_r} \times 100$, where $J_\pi$ is the return acquired by the learned policy, $J_r$ is the return by a random policy, and $J_e$ is the return of an expert policy. We offer reference values of $J_r$ and $J_e$ for all tasks.

## 4.4 Baseline Algorithms

We implement various off-dynamics RL algorithms and baselines in ODRL, categorized into 4 settings. **Online-Online:** DARC [16] that trains domain classifiers for penalizing source domain rewards, VGDF [76] that performs source domain data filtering from a value estimate perspective, PAR [44] that penalizes source domain data by measuring the representation mismatch between two domains; **Offline-Online:** H2O [52] that leverages the domain classifier for importance weighting, BC_VGDF [76] and BC_PAR [44] that incorporate the behavior cloning (BC) term for the source domain data; **Online-Offline:** H2O and PAR_BC that introduces the BC term to the target domain data; **Offline-Offline:** DARA [41], which is exactly the offline version of DARC and BOSA [42] that employs two support-constrained objective for regularization.

Furthermore, we assemble the following baselines. **Online-Online**: SAC [26] that trains on data from both domains, SAC_IW that adopts the domain classifier for importance weighting, SAC_tune that first learns an SAC policy in the source domain and directly finetunes it in the target domain. **Offline-Online:** BC_SAC, CQL_SAC, MCQ_SAC that apply SAC loss on target domain samples and adopts BC loss, CQL [40] loss, MCQ [46] loss for source domain data, respectively, RLPD [3] that leverages random ensemble distillation and layer normalization for efficient online learning. **Online-Offline:** SAC_BC, SAC_CQL, SAC_MCQ where SAC loss is applied upon source domain data training and BC loss, CQL loss and MCQ loss are integrated for target domain data. **Offline-Offline:** IQL [39] and TD3_BC [20] that train on offline samples from both domains. Most of these methods are proposed to examine whether it is feasible to train a good policy by treating the two domains as one mixed domain (i.e., the transition is sampled from $\hat{P}_{\text{mix}} = \lambda P_{\text{src}} + (1 - \lambda)P_{\text{tar}}, \lambda \in \{0, 1\}$).

---

[2]MuJoCo datasets are gathered using SAC [26] while Adroit datasets are collected with DARC [16] because SAC fails to learn meaningful policies even after 5M environmental steps in the target domain.

The algorithmic details can be found in Appendix B. We reproduce all methods by following their official codes or respective papers within a *single* file. This is motivated by the CORL [64] and CleanRL [30] projects, targeting at bringing together all core algorithm designs and making it "easy-to-hack" for practitioners. The aforementioned baseline algorithms share similar code styles in ODRL and are equipped with separate `yaml` configuration files for different tasks.

## 5 Experiments

In this section, we investigate the dynamics adaptation capability of the approaches involved in ODRL. We run experiments across multiple experimental settings and examine how these methods behave given different dataset qualities and domain gaps. For each task, we report the final mean performance along with the standard deviations *achieved in the target domain* across 5 varied random seeds. The empirical evaluations are accompanied by some critical observations (**Obs**) and insights. Due to space limits, we defer wider experimental evaluations to Appendix E.

### 5.1 Benchmark Results

Considering the extensive array of tasks in our benchmark, it is costly to run algorithms on all of them. To enable a comprehensive evaluation as much as possible, we opt for two locomotion tasks (*ant*, *walker2d*), each featuring 4 types of dynamic shifts (*friction*, *gravity*, *kinematic*, *morphology*), with each shift comprising two tasks (0.5/5.0 times friction/gravity, medium/hard shift levels for kinematic and morphology tasks[3]). This allows us to examine the performance of existing methods under varying dynamics discrepancies. Additionally, we consider AntMaze tasks with two distinct map sizes (*small*, *medium*), each involving two maze structures (*empty*, *centerblock* for the small maze, and map type *1*, *2* for the medium-size maze), to reveal the transfer capabilities of these methods across diverse landscapes and obstacles. Moreover, we include two dexterous manipulation tasks (*pen*, *door*) with two dynamic shifts (*broken-joint*, *shrink-finger*) and the *easy* shift level to examine whether off-dynamics RL methods are also effective in high-dimensional and sparse reward tasks. We initially turn our attention to the **Online-Online** setting. Recall that we only have a limited budget of data from the target domain. Hence, we run all algorithms for 1M environmental steps in the source domain, and 0.1M steps in the target domain. The aggregated normalized score comparison of baselines is depicted in Figure 3. Based on the results, we summarize the following key observations.

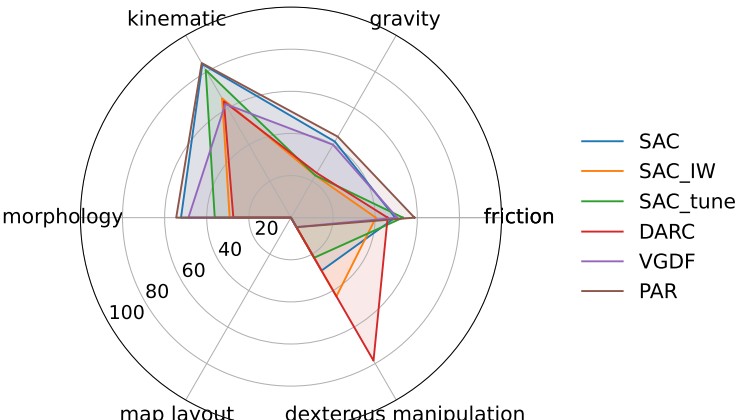

Figure 3: **Radar chart comparison of different methods.** We report the aggregated normalized score of tasks within each category given the online source domain and target domain.

**Obs 1.** *No single off-dynamics RL algorithm can exhibit advantages across all scenarios.*

**Obs 2.** *PAR achieves the best performance on locomotion tasks but fails on the Antmaze domain and Adroit domain.*

**Obs 3.** *AntMaze tasks are extremely challenging and no algorithm can achieve meaningful returns, indicating that adapting policies across barriers is hard for state-based methods.*

---

[3]We use *ant-kinematic-anklejnt*, *ant-morph-alllegs*, *walker2d-kinematic-footjnt*, *walker2d-morph-leg* tasks.

**Obs 4.** *Off-dynamics RL methods often fail on dexterous hand manipulation tasks. Only DARC and SAC_IW can achieve comparatively good performance on them.*

These findings suggest that there is still a considerable distance to explore in developing a truly general dynamics-aware RL algorithm. It appears that current methods excel at addressing specific types of dynamics shifts. The lack of success with PAR in dexterous manipulation tasks can be credited to the poor reward penalty terms on the source domain data. In fact, the severe value overestimation phenomenon is often observed when running Adroit tasks with PAR. Importantly, although VGDF performs on par with PAR in locomotion tasks, it also falls short in the Adroit domain. This could be because VGDF requires training dynamics models of the target domain, which can be challenging for complex tasks like dexterous manipulation [48, 34, 45]. The fact that SAC_IW often results in inferior performance than DARC indicates that leveraging the learned domain classifiers for importance sampling weighting is not preferred in the online setting.

**Obs 5.** *Surprisingly, simply training the SAC agent using both the source domain data and target domain samples can incur good performance on numerous kinds of dynamics shifts.*

The last observation is quite intriguing, as it shows that merely considering the source and target domains as a single mixed domain and training on data from both domains without introducing any additional components can be effective in many tasks. The direct finetuning approach (i.e., SAC_tune) has advantages in certain tasks, but it frequently results in suboptimal performance when faced with morphology and gravity shifts.

## 5.2 Performance Comparison Given Offline Datasets

We then proceed to examine the performance of baseline methods when either the source domain or the target domain is offline. This presents substantial challenges for the algorithm to learn transferable policies because (a) if the source domain is offline, finding dynamics-consistent transitions in the dataset may be difficult due to limited coverage; (b) if the target domain is offline, gathering high-quality experiences that are closely aligned with the target domain may become challenging, and poor samples from the source domain can negatively impact the learning process of the agent.

**Offline Source Domain.** We pick two tasks, *ant-gravity* with shift level 5.0, and *walker2d-friction* with shift level 0.5. We run experiments by leveraging three varied qualities of source domain datasets from D4RL (*medium-replay*, *medium*, *expert*). The agent undergoes training for 1M gradient steps and is allowed to interact with the target domain every 10 gradient steps, i.e., 0.1M environmental steps. We present the results in Figure 4 and have the corresponding observations.

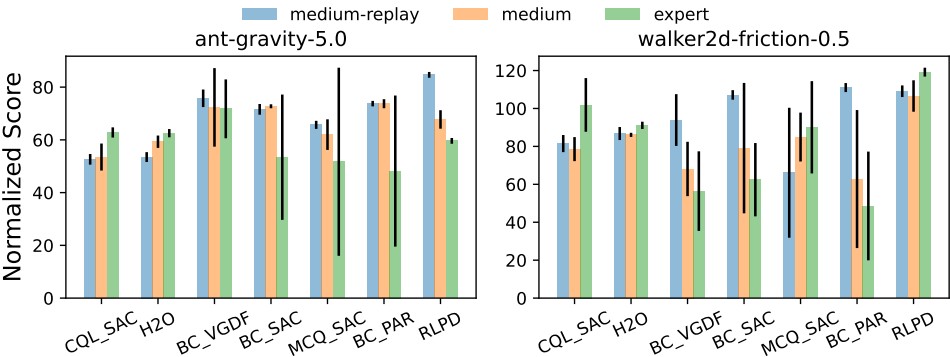

Figure 4: **Normalized score comparison of methods under distinct source domain datasets.** We report the final average normalized score in the target domain, along with the standard deviation.

**Obs 6.** *A higher quality of the source domain dataset does not necessarily imply better performance in the target domain, even when an expert source domain dataset is provided.*

**Obs 7.** *Baseline methods that treat two domains as one mixed domain can achieve good performance on some tasks, sometimes even surpassing off-dynamics methods like BC_PAR, BC_VGDF, and H2O.*

**Obs 8.** *Methods that leverage the conservative value penalties (e.g., CQL_SAC) can outperform methods that involve the BC term (e.g., BC_PAR), given expert source domain datasets.*

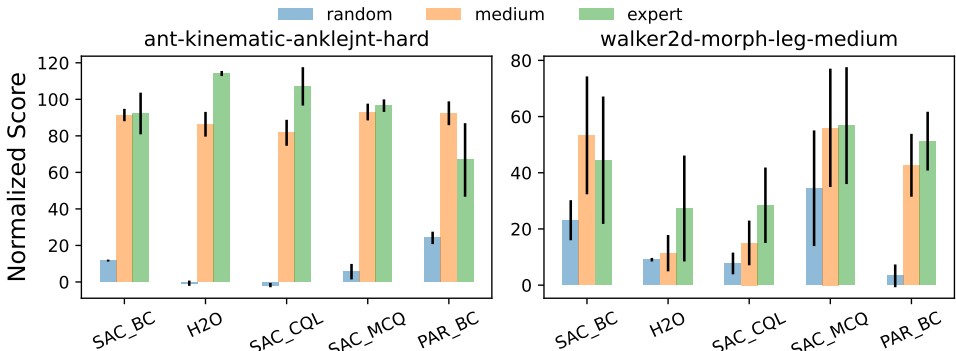

Figure 5: **Normalized score comparison of baselines given varied qualities of target domain datasets.** The final mean performance and its standard deviation in the target domain are presented.

**Offline Target Domain.** We choose two tasks (*ant-kinematic-anklejnt* with a hard shift level, and *walker2d-morph-leg* with a medium shift level) and three types of target domain dataset qualities (*random*, *medium*, *expert*). We execute the implemented algorithms in the **Online-Offline** setting category for 500K gradient steps, allowing the agent to interact with the source domain at each step. We do not run for 1M steps since the offline dataset sizes of these tasks are limited (5000 transitions), and the agent may suffer from the overfitting issue if a large amount of gradient steps is employed. The results can be found in Figure 5, where we find that **Obs 7** and **Obs 8** still hold when the target domain is offline. We also conclude some new observations below.

**Obs 9.** *The agent's performance can also be inferior given expert target domain datasets (e.g., SAC_BC) and it is challenging to achieve a meaningful score using random target domain datasets.*

**Obs 10.** *A larger dynamics gap from the source domain does not indicate that it becomes harder for the agent to learn effective policies, e.g., almost all methods can achieve quite good performance in the ant-kinematic-anklejnt task with a shift level hard.*

### 5.3 Connections Between Source Domain Performance and Target Domain Performance

We investigate if there is a positive relationship between the agent's performance in the source domain and the target domain. We select the *ant-friction-0.5* task in the **Online-Online** setting, and *walker2d-kinematic-footjnt-medium* task in the **Online-Offline** setting with a medium-quality target domain dataset. We conduct experiments on these tasks using the respective algorithms within each category. We chart the learning curves in both the source domain and the target domain, and display the results in Figure 6. We observe that VGDF demonstrates strong performance in the target domain for the *ant* task, but its policy performance in the source domain is considerably weak. Conversely, DARC and SAC_IW learn relatively effective policies in the source domain, but they typically struggle to achieve good performance in the target domain. For other methods, they generally exhibit good performance in both domains. It is then interesting to conclude the following observation.

**Obs 11.** *There is no definitive correlation between the policy's performance in the source domain and the target domain (it depends on the specific algorithm).*

This observation is vital because it conveys that the performance in the source domain is not a reliable indicator for estimating the policy's performance in the target domain.

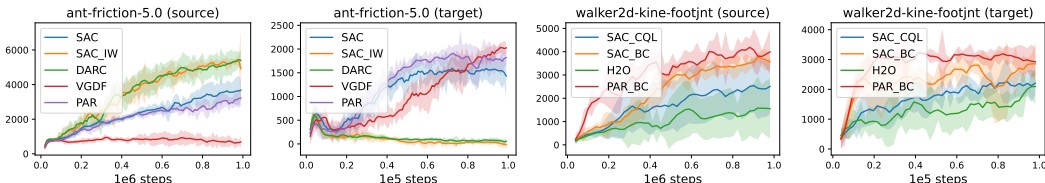

Figure 6: **Return comparison across two domains on selected tasks.** The solid lines represent the mean return and the shaded region captures the standard deviation. Kine denotes kinematic.

# 6 Conclusion and Limitations

In this paper, we propose ODRL, the first benchmark for off-dynamics RL. ODRL allows the source or target domain to be either online or offline, and introduces a wide spectrum of dynamics shift tasks to facilitate a comprehensive and persuasive evaluation. We implement many off-dynamics RL algorithms within a single file to bring together core algorithm designs, and additionally propose some baselines for each experimental setting. Our empirical results show that no existing method can lead all types of dynamics shifts. We conclude some critical empirical observations, which can serve as valuable takeaways for readers. Our benchmark is promised to be actively maintained. We hope our benchmark can pave the way for developing general dynamics-aware RL algorithms.

**Limitations.** ODRL primarily supports sim-to-sim policy adaptation tasks. Adapting the policy to real-world scenarios may be more complicated (e.g., a combination of various dynamics shifts can occur). However, the proposed dynamics shifts in our study are common, and our benchmark should serve as a valuable testbed for developing more advanced off-dynamics RL algorithms.

# Acknowledgments

This work was supported by the STI 2030-Major Projects under Grant 2021ZD0201404, the National Natural Science Foundation of China (NSFC) under Grant 62450001 and 62476008. The authors would like to thank the anonymous reviewers for their valuable comments and advice.

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

# A Benchmark Summary

In this section, we offer a brief overview of our benchmark, as shown in Table 2. ODRL primarily consists of three domain categories: locomotion, navigation, and dexterous hand manipulation. Different domains feature various types of dynamics shifts and are designed to explore specific challenges. For each domain:

**Locomotion:** We consider four tasks (*ant, halfcheetah, hopper, walker2d*), each with four kinds of dynamics shifts (*friction shift, gravity shift, kinematic shift, morphology shift*). We have 4 shift levels (*0.1, 0.5, 2.0, 5.0*) for friction and gravity shift tasks, and 3 shift levels (*easy, medium, hard*) for kinematic and morphology tasks. ODRL provides 4 tasks for friction mismatch, 4 tasks for gravity mismatch, 6 tasks for kinematic mismatch, and 6 tasks for morphology mismatch, resulting in **20** dynamics shift tasks for a single robot and a total of **80** tasks for the locomotion domain. As both the source and target domains can be offline, our benchmark can offer a vast number of tasks to facilitate a comprehensive evaluation of the agent's policy adaptation capability. We collect offline datasets for the target domain by training the SAC [26] agent solely in the target domain. We provide three types of target domain datasets (*random, medium, expert*), where the medium-quality datasets have approximately half the return of the expert trajectories. The offline target domain dataset size is limited to 5,000 transitions. The locomotion tasks aim to address the following open problem:

> **Open problem:** How can we develop a general enough algorithm that can handle various kinds of dynamics shifts?

**Navigation (AntMaze):** The AntMaze navigation task consists of three different maze sizes (*small*, *medium*, *large*). For each map structure, we provide 6 distinct types of maze layouts, resulting in a total of **18** tasks for the AntMaze domain. The target domain offline datasets are collected by training a goal-reaching policy and using it in conjunction with the high-level waypoint generator from Maze2D, which provides sub-goals to guide the agent towards the goal. We do not use SAC for locomotion tasks since it fails to reach goals after 5 million environmental steps in the target domain. We only provide one mixed offline target domain dataset for each map layout, where the offline dataset contains both unsuccessful and successful trajectories. The dataset size is limited to 10,000. Note that the embodied ant robot in the maze remains unmodified, and only the underlying maze layout changes to address the following open problem:

> **Open problem:** How can we enable the agent to transfer across different landscapes and map layouts given limited data from the target domain?

**Dexterous Manipulation:** We adopt four tasks from Adroit (*pen*, *door*, *relocate*, *hammer*) and primarily focus on two dynamics shift problems: kinematic shifts and morphology shifts, each with three shift levels (*easy*, *medium*, *hard*). This results in a total of **24** tasks for the dexterous hand manipulation domain. For the target domain offline datasets, we provide three types of dataset qualities (*random*, *medium*, *expert*), where the medium-quality and expert-quality datasets are collected using the DARC [16] algorithm because we find that the SAC agent cannot learn any meaningful performance even after 5 million environmental steps of training in the target domain. The medium-quality datasets have approximately half the return of the expert datasets. The dataset size is limited to 5,000 transitions. By designing this domain, we aim to explore the following open problem:

> **Open problem:** How can we achieve efficient policy adaptation in complex, high-dimensional, and sparse reward dexterous hand manipulation tasks?

# B Off-Dynamics RL Algorithms and Baselines

In this section, we provide detailed descriptions of the implemented off-dynamics RL algorithms and the additional baselines that we introduce ourselves. We primarily introduce a fine-tuning method for the **Online-Online** setting (i.e., SAC_tune) and the rest of the methods that train on data from

Table 2: **An overview of Benchmark tasks.** We mainly include three distinct domains and equip each domain with varied dynamics shift tasks. Offline target domain datasets are available for all tasks to facilitate the **Online-Offline** setting and **Offline-Offline** setting evaluations.

| Task Domain | Friction | Gravity | Kinematic | Morphology | Map Structure | Offline datasets |
|---|---|---|---|---|---|---|
| Locomotion | ✔ | ✔ | ✔ | ✔ | ✗ | ✔ |
| Navigation | ✗ | ✗ | ✗ | ✗ | ✔ | ✔ |
| Dexterous Manipulation | ✗ | ✗ | ✔ | ✔ | ✗ | ✔ |

both the source and target domains by treating the two domains as one mixed domain. Denote $P_{\mathrm{src}}, P_{\mathrm{tar}}$ as the transition dynamics of the source domain and the target domain, respectively, then the transition dynamics of the mixed domain gives $\hat{P}_{\mathrm{mix}} = \lambda P_{\mathrm{src}} + (1 - \lambda) P_{\mathrm{tar}}$, where $\lambda \in \{0, 1\}$ is a random variable. That is, we suppose that the samples in the two domains are sampled from $\hat{P}_{\mathrm{mix}}$ and hence can simply train a single model on data from the two domains. For clarity, we name the algorithm by what objectives are adopted in the source domain and the target domain. For example, BC_VGDF is an **Offline-Online** algorithm where the BC term is applied to the source domain data, while the vanilla VGDF objectives are used for the target domain data. Similarly, CQL_SAC is also an **Offline-Online** algorithm where the CQL loss is adopted for source domain training, while the vanilla SAC loss is used for target domain data training. Conversely, SAC_CQL is an **Online-Offline** algorithm where the SAC loss is used for source domain data and the CQL loss is adopted for target domain data.

## B.1 Online-Online setting

We implement the following off-dynamics RL methods for this setting:

**DARC:** We follow the original paper and train two domain classifiers $q_{\theta_{\mathrm{SAS}}}(\mathrm{target}|s_t, a_t, s_{t+1})$, $q_{\theta_{\mathrm{SA}}}(\mathrm{target}|s_t, a_t)$ parameterized by $\theta_{\mathrm{SAS}}$ and $\theta_{\mathrm{SA}}$, respectively. These domain classifiers are trained using the following losses:

$$\mathcal{L}(\theta_{\mathrm{SAS}}) = \mathbb{E}_{\mathcal{D}_{\mathrm{tar}}}\left[\log q_{\theta_{\mathrm{SAS}}}(\mathrm{target}|s_t, a_t, s_{t+1})\right] + \mathbb{E}_{\mathcal{D}_{\mathrm{src}}}\left[\log(1 - q_{\theta_{\mathrm{SAS}}}(\mathrm{target}|s_t, a_t, s_{t+1}))\right],$$
$$\mathcal{L}(\theta_{\mathrm{SA}}) = \mathbb{E}_{\mathcal{D}_{\mathrm{tar}}}\left[\log q_{\theta_{\mathrm{SA}}}(\mathrm{target}|s_t, a_t)\right] + \mathbb{E}_{\mathcal{D}_{\mathrm{src}}}\left[\log(1 - q_{\theta_{\mathrm{SA}}}(\mathrm{target}|s_t, a_t))\right],$$

where $\mathcal{D}_{\mathrm{src}}, \mathcal{D}_{\mathrm{tar}}$ are replay buffers of the source domain and the target domain, respectively. We set the Gaussian standard deviation $\sigma = 1$ when training these domain classifiers, which is recommended by the authors. DARC compensates the source domain rewards by estimating the dynamics gap: $\log \frac{P_{\mathcal{M}_{\mathrm{tar}}}(s_{t+1}|s_t, a_t)}{P_{\mathcal{M}_{\mathrm{src}}}(s_{t+1}|s_t, a_t)}$. DARC approximates this term by leveraging the trained domain classifiers. Formally, its reward penalty term, $\delta r$, gives

$$\delta r(s_t, a_t) = -\log \frac{q_{\theta_{\mathrm{SAS}}}(\mathrm{target}|s_t, a_t, s_{t+1}) q_{\theta_{\mathrm{SA}}}(\mathrm{source}|s_t, a_t)}{q_{\theta_{\mathrm{SAS}}}(\mathrm{source}|s_t, a_t, s_{t+1}) q_{\theta_{\mathrm{SA}}}(\mathrm{target}|s_t, a_t)}. \tag{1}$$

The source domain rewards are modified accordingly via:

$$\hat{r}_{\mathrm{src}}^{\mathrm{DARC}} = r_{\mathrm{src}}(s_t, a_t) - \delta r. \tag{2}$$

We implement DARC by following its official code repository[4].

**VGDF:** The core idea of VGDF is to filter source domain data that share similar value estimates as those in the target domain. To fulfill that, it trains an ensemble of dynamics model [33, 54, 57, 8, 56] in the *raw state-action space* of the target domain to predict the next state that follows the transition dynamics of the target domain given source domain data $(s_{\mathrm{src}}, a_{\mathrm{src}})$. Then it measures the mean and variance of the value ensemble $\{Q(s'_i, a'_i)\}_{i=1}^{M}$ to build a Gaussian distribution, where $s'_i$ is the predicted next state, $a'_i$ is sampled from the policy, and $M$ is the ensemble size. Subsequently, the rejection sampling approach is employed to select a fixed percentage ($\xi\%, \xi \in [0, 100]$) of source domain data with the highest likelihood estimation and share them with the target domain, i.e., VGDF optimizes the following objective:

$$\begin{aligned}
\mathcal{L}_{\mathrm{critic}} = \mathbb{E}_{(s,a,r,s') \sim \mathcal{D}_{\mathrm{tar}}}\left[(Q_{\theta_i}(s, a) - y)^2\right] \\
+ \mathbb{E}_{(s,a,r,s') \sim \mathcal{D}_{\mathrm{src}}}\left[\mathbb{1}(\Lambda(s, a, s') > \Lambda_{\xi\%})(Q_{\theta_i}(s, a) - y)^2\right], i \in \{1, 2\},
\end{aligned} \tag{3}$$

[4]https://github.com/google-research/google-research/tree/master/darc

where $\mathbb{1}(\cdot)$ is the indicator function, $\Lambda(s, a, s')$ is the fictitious value proximity (FVP) representing the likelihood of the source domain state value, $\Lambda_{\xi\%}$ denotes the top $\xi$-quantile likelihood estimation of the source domain batch data, $y$ is the target value. VGDF also trains an additional exploration policy for improved exploration. We reproduce VGDF by following its official implementation [5]. We train VGDF in the source domain for 1M environmental steps and allow the agent to interact with the target domain every 10 steps.

**PAR:** PAR detects the dynamics mismatch by capturing the representation mismatch, i.e., the representation deviation between the source domain state-action pair and its subsequent next state using the state encoder $f$ and state-action encoder $g$ trained only in the target domain. Note that PAR only trains one single state encoder along with one single state-action encoder. The encoders are updated via the following objective:

$$\mathcal{L}(\psi, \xi) = \mathbb{E}_{(s,a,s')\sim\mathcal{D}_{\text{tar}}} \left[ (g_\xi(f_\psi(s), a) - \text{SG}(f_\psi(s')))^2 \right], \tag{4}$$

where $\text{SG}$ denotes the stop gradient operator. Then PAR modifies the source domain reward by using the deviations between the representations:

$$\hat{r}_{\text{PAR}} = r_{\text{src}} - \beta \times \left[ g_\xi(f_\psi(s_{\text{src}}), a_{\text{src}}) - f_\psi(s'_{\text{src}}) \right]^2, \tag{5}$$

where $\beta$ is the penalty coefficient that controls the strengths of the penalty. Afterwards, the critic and the actor are optimized by leveraging data from both domains. We reproduce PAR by following its official implementation [6]. We train PAR for 1M environmental steps in the source domain, and let the policy interact with the target domain every 10 steps.

We also include the following baselines proposed by ourselves:

**SAC:** The vanilla SAC policy [26] that simply train on both the source domain and the target domain data with the following critic loss:

$$\mathcal{L}_{\text{critic}} = \mathbb{E}_{(s,a,r,s')\sim\mathcal{D}_{\text{src}}\cup\mathcal{D}_{\text{tar}}} \left[ (Q_{\theta_i}(s, a) - y)^2 \right], i \in \{1, 2\}, \tag{6}$$

and the policy loss gives,

$$\mathcal{L}_{\text{actor}} = \mathbb{E}_{s\sim\mathcal{D}_{\text{src}}\cup\mathcal{D}_{\text{tar}}, a\sim\pi_\phi(\cdot|s)} \left[ \min_{i=1,2} Q_{\theta_i}(s, a) - \alpha \log \pi_\phi(\cdot|s) \right]. \tag{7}$$

Similarly, we train the agent for 1M environmental steps in the source domain, and collect experiences in the target domain every 10 steps.

**SAC_IW:** This approach denotes SAC with importance sampling weights. It generally resembles vanilla DARC, except that it does not perform reward correction for the source domain data, but utilizes that estimated dynamics gap as an importance sampling term for critic updates, i.e.,

$$\mathcal{L}_{\text{critic}} = \mathbb{E}_{(s,a,r,s')\sim\mathcal{D}_{\text{src}}} \left[ \omega(s, a, s') \left( Q_{\theta_i}(s, a) - y \right)^2 \right], i \in \{1, 2\}, \tag{8}$$

where

$$\omega(s, a, s') = \frac{q_{\theta_{\text{SAS}}}(\text{target}|s, a, s') q_{\theta_{\text{SA}}}(\text{source}|s, a)}{q_{\theta_{\text{SAS}}}(\text{source}|s, a, s') q_{\theta_{\text{SA}}}(\text{target}|s, a)}. \tag{9}$$

To ensure training stability, we clip the value of the importance sampling weight $\omega(s, a, s')$ to lie in the range of $[10^{-4}, 1]$. We train SAC_IW agent in the source domain for 1M environmental steps and let it interact with the target domain every 10 steps.

**SAC_tune:** This method simply first trains the SAC agent in the source domain for 1M environmental steps, and then directly finetunes the learned policy in the target domain for 0.1M environmental steps without sampling source domain data. This baseline is designed to investigate the effectiveness of a direct fine-tuning of the learned policy.

---

[5]https://github.com/Kavka1/VGDF
[6]https://github.com/dmksjfl/PAR

## B.2 Offline-Online setting

We implement the following dynamics-aware algorithms:

**H2O:** H2O also trains domain classifiers to estimate the dynamics gap, while using it as importance sampling weights for critic optimization. It additionally combines CQL loss to inject conservatism. Note that the vanilla H2O is designed for **Online-Offline** setting, i.e., the source domain is online, while the target domain is offline. To match the **Offline-Online** setting, we slightly modify the objective functions of H2O. To be specific, its critic objective function is formulated as:

$$\mathcal{L}_{\mathrm{critic}} = \mathbb{E}_{\mathcal{D}_{\mathrm{tar}}}\left[(Q_{\theta_i}(s,a) - y)^2\right] + \mathbb{E}_{\mathcal{D}_{\mathrm{src}}}\left[\omega(s,a,s')(Q_{\theta_i}(s,a) - y)^2\right]$$
$$+ \beta_{\mathrm{CQL}}\left(\mathbb{E}_{s\sim\mathcal{D}_{\mathrm{src}},\tilde{a}\sim\pi_\phi(\cdot|s)}[\omega(s,a,s')Q_{\theta_i}(s,\tilde{a})] - \mathbb{E}_{\mathcal{D}_{\mathrm{src}}}[\omega(s,a,s')Q_{\theta_i}(s,a)]\right), i \in \{1,2\}, \tag{10}$$

where $\omega(s,a,s')$ is the importance sampling weights specified by the dynamics gap estimation term in Equation 9, $\beta_{\mathrm{CQL}}$ is the penalty coefficient. We set $\beta_{\mathrm{CQL}} = 10.0$ instead of $\beta_{\mathrm{CQL}} = 0.01$ that is recommended in the H2O paper because we find $\beta_{\mathrm{CQL}} = 10.0$ incurs a better performance. We reproduce H2O by using the official codebase[7]. We train H2O for $10^6$ gradient steps, and allow the policy to gather data in the target domain every 10 steps.

**BC_VGDF:** This variant has the same critic objective function as VGDF (Equation 3). What makes it different is that we incorporate a behavior cloning (BC) term into its policy training to realize conservatism injection, i.e.,

$$\mathcal{L}_{\mathrm{actor}} = \lambda \cdot \mathbb{E}_{\substack{s\sim\mathcal{D}_{\mathrm{src}}\cup\mathcal{D}_{\mathrm{tar}},\\ a\sim\pi_\phi(\cdot|s)}}\left[\min_{i=1,2} Q_{\theta_i}(s,a) - \alpha\log\pi_\phi(\cdot|s)\right] + \mathbb{E}_{\substack{(s,a)\sim\mathcal{D}_{\mathrm{src}},\\ \hat{a}\sim\pi_\phi(\cdot|s)}}[(a - \hat{a})^2], \tag{11}$$

where $\lambda = \dfrac{\nu}{\frac{1}{N}\sum_{(s_j,a_j)}\min_{i=1,2} Q_{\theta_i}(s_j,a_j)}$, and $\nu \in \mathbb{R}^+$ is the normalization coefficient. We train BC_VGDF for $10^6$ gradient steps, with $10^5$ interactions with the target domain.

**BC_PAR:** Akin to BC_VGDF, an additional behavior cloning term is involved for the source domain data in the PAR algorithm to give birth to BC_PAR. The reward modification term and the critic optimization objective are kept unchanged and the policy network is updated via:

$$\mathcal{L}_{\mathrm{actor}} = \lambda \cdot \mathbb{E}_{\substack{s\sim\mathcal{D}_{\mathrm{src}}\cup\mathcal{D}_{\mathrm{tar}},\\ a\sim\pi_\phi(\cdot|s)}}\left[\min_{i=1,2} Q_{\theta_i}(s,a) - \alpha\log\pi_\phi(\cdot|s)\right] + \mathbb{E}_{\substack{(s,a)\sim\mathcal{D}_{\mathrm{src}},\\ \hat{a}\sim\pi_\phi(\cdot|s)}}[(a - \hat{a})^2], \tag{12}$$

where $\lambda = \dfrac{\nu}{\frac{1}{N}\sum_{(s_j,a_j)}\min_{i=1,2} Q_{\theta_i}(s_j,a_j)}$ and $\nu \in \mathbb{R}^+$ is the normalization coefficient. We train BC_PAR for $10^6$ gradient steps, and let it interact with the target domain every 10 gradient steps.

Furthermore, we introduce the following additional baseline for this setting. These baselines are proposed to examine whether it is feasible to directly treat the two domains with dynamics discrepancies as one mixed domain and simply train on data from both domains without introducing extra techniques such as reward correction.

**BC_SAC:** This baseline leverages both offline source domain data and online target domain transitions for learning a policy. Since learning from offline data requires conservatism, while learning from online samples does not, we simply incorporate a behavior cloning term in the policy update objective of the SAC algorithm, i.e., the critics are updated via Equation 6 and the actor is updated via:

$$\mathcal{L}_{\mathrm{actor}} = \lambda \cdot \mathbb{E}_{\substack{s\sim\mathcal{D}_{\mathrm{src}}\cup\mathcal{D}_{\mathrm{tar}},\\ a\sim\pi_\phi(\cdot|s)}}\left[\min_{i=1,2} Q_{\theta_i}(s,a) - \alpha\log\pi_\phi(\cdot|s)\right] + \mathbb{E}_{\substack{(s,a)\sim\mathcal{D}_{\mathrm{src}},\\ \hat{a}\sim\pi_\phi(\cdot|s)}}[(a - \hat{a})^2], \tag{13}$$

where $\lambda = \dfrac{\nu}{\frac{1}{N}\sum_{(s_j,a_j)}\min_{i=1,2} Q_{\theta_i}(s_j,a_j)}$ and $\nu \in \mathbb{R}^+$ is the normalization coefficient. We train BC_SAC for 1M gradient steps and let it interact with the target domain every 10 steps.

**CQL_SAC:** We introduce a different conservative objective against BC_SAC by leveraging the conservative Q-learning (CQL [40]) approach to facilitate offline source domain data training. That is,

---

[7]https://github.com/t6-thu/H2O

the conservatism is realized by penalizing the learned value functions instead of the policy constraint. We train critics by updating source domain data with the CQL loss while the online target domain data with the SAC loss, $i.e.$,

$$
\begin{aligned}
\mathcal{L}_{\mathrm{critic}} = \mathbb{E}_{\mathcal{D}_{\mathrm{src}} \cup \mathcal{D}_{\mathrm{tar}}} & \left[ (Q_{\theta_i}(s,a) - y)^2 \right] \\
& + \beta_{\mathrm{CQL}} \left( \mathbb{E}_{s \sim \mathcal{D}_{\mathrm{src}}, \tilde{a} \sim \pi_\phi(\cdot|s)}[Q_{\theta_i}(s,\tilde{a})] - \mathbb{E}_{\mathcal{D}_{\mathrm{src}}}[Q_{\theta_i}(s,a)] \right), i \in \{1,2\},
\end{aligned}
\tag{14}
$$

where $\beta_{\mathrm{CQL}}$ is the CQL penalty coefficient. We generally follows the official implementation of CQL[8]. We train CQL_SAC for $10^6$ gradient steps, with $10^5$ interactions with the target domain.

**MCQ_SAC:** This baseline explores another value-based conservative approach, mildly conservative Q-learning (MCQ) [46], due to its good performance in the offline D4RL benchmark tasks and offline-to-online tasks. MCQ actively trains the OOD actions by assigning them pseudo target values. Similar to CQL_SAC, we train the source domain offline data with the MCQ loss and the target domain online data with the SAC loss, and the critic objective function gives,

$$
\begin{aligned}
\mathcal{L}_{\mathrm{critic}} = \mathbb{E}_{\mathcal{D}_{\mathrm{tar}}} & \left[ (Q_{\theta_i}(s,a) - y)^2 \right] + \lambda \mathbb{E}_{\mathcal{D}_{\mathrm{src}}} \left[ (Q_{\theta_i}(s,a) - y)^2 \right] \\
& + (1 - \lambda) \mathbb{E}_{s \sim \mathcal{D}_{\mathrm{src}}, \tilde{a} \sim \pi_\phi(\cdot|s)}[(Q_{\theta_i}(s,\tilde{a}) - y')^2], i \in \{1,2\},
\end{aligned}
\tag{15}
$$

where $y' = \min_{j=1,2} \mathbb{E}_{\{a'_i\}^N \sim \hat{\mu}} \left[ \max_{a' \sim \{a'_i\}^N} Q_{\theta_j}(s,a') \right]$ is the pseudo target value measured by MCQ, $\hat{\mu}$ is the learned behavior policy, $N$ is the number of sampled actions, $\lambda$ is the weighting coefficient tha that balances the in-distribution training and OOD data training. We follow the MCQ paper and set $N = 10$ by default. The policy objective is the same as SAC. We implement MCQ by following the official codebase[9]. We train MCQ_SAC for 1M gradient steps, and allow the policy to interact with the target domain every 10 steps.

**RLPD:** RLPD [3] is a method that is carefully designed for learning online with offline data. It leverages random ensemble distillation and layer normalization for mitigating overestimation bias and fulfilling sample efficient online training. To be specific, it learns an ensemble of critics, and sample a subset from the ensemble to calculate the target value. It does not introduce any explicit conservative terms like CQL or policy regularization methods, but resorts to the layer normalization technique for alleviating the overestimation issue. We implement this method by following the official codebase[10]. We set the update-to-data ratio as 1 here to make it a fair comparison between different algorithms.

## B.3 Online-Offline setting

We provide the following off-dynamics RL approaches in this category.

**H2O:** H2O is initially designed for the **Online-Offline** setting. Its core idea is to leverage the biased online source domain to facilitate the training of the target domain, which is offline and contain a limited coverage of trajectories. Since simply combining the source domain data and target domain data may incur a *potential* distribution shift issue, H2O addresses this problem by estimating the dynamics gap between the source domain and the target domain, $\frac{P_{\mathcal{M}_{\mathrm{tar}}}(s_{t+1}|s_t,a_t)}{P_{\mathcal{M}_{\mathrm{src}}}(s_{t+1}|s_t,a_t)}$, and leverages this dynamics gap term for correcting the biased online samples. To that end, it trains domain classifiers and estimate the dynamics gap. The critics in H2O are optimized by:

$$
\begin{aligned}
\mathcal{L}_{\mathrm{critic}} = \mathbb{E}_{\mathcal{D}_{\mathrm{tar}}} & \left[ (Q_{\theta_i}(s,a) - y)^2 \right] + \mathbb{E}_{\mathcal{D}_{\mathrm{src}}} \left[ \omega(s,a,s')(Q_{\theta_i}(s,a) - y)^2 \right] \\
& + \beta_{\mathrm{CQL}} \left( \mathbb{E}_{s \sim \mathcal{D}_{\mathrm{src}}, \tilde{a} \sim \pi_\phi(\cdot|s)}[\tilde{\omega}(s,a)Q_{\theta_i}(s,\tilde{a})] - \mathbb{E}_{\mathcal{D}_{\mathrm{tar}}}[Q_{\theta_i}(s,a)] \right), i \in \{1,2\},
\end{aligned}
\tag{16}
$$

where $\omega(s,a,s')$ is estimated via Equation 9, $\beta_{\mathrm{CQL}}$ is the penalty coefficient, and $\tilde{\omega}(s,a)$ gives:

$$
\tilde{\omega}(s,a) := \frac{u(s,a)}{\sum_{\bar{s},\bar{a}} u(\bar{s},\bar{a})}, u(s,a) := \mathbb{E}_{s' \sim P_{\mathrm{tar}}} \left[ \log \frac{P_{\mathrm{tar}}(s'|s,a)}{P_{\mathrm{src}}(s'|s,a)} \right].
\tag{17}
$$

Note that the objective here (Equation 16) is quite different from H2O in the **Offline-Online** setting (Equation 10). We train H2O for 500K gradient steps, where the agent is also allowed to interact with the source domain for 500K environmental steps.

---

[8] https://github.com/aviralkumar2907/CQL
[9] https://github.com/dmksjfl/MCQ
[10] https://github.com/ikostrikov/rlpd

**PAR_BC:** This baseline generally follows the same way of calculating the reward modification term as vanilla PAR and BC_PAR, but it adds the behavior cloning term to the target domain data since they are offline. The policy is then updated via:

$$\mathcal{L}_{\text{actor}} = \lambda \cdot \mathbb{E}_{\substack{s \sim \mathcal{D}_{\text{src}} \cup \mathcal{D}_{\text{tar}}, \\ a \sim \pi_\phi(\cdot|s)}} \left[ \min_{i=1,2} Q_{\theta_i}(s,a) - \alpha \log \pi_\phi(\cdot|s) \right] + \mathbb{E}_{\substack{(s,a) \sim \mathcal{D}_{\text{tar}}, \\ \hat{a} \sim \pi_\phi(\cdot|s)}} [(a - \hat{a})^2], \quad (18)$$

where $\lambda = \dfrac{\nu}{\frac{1}{N} \sum_{(s_j, a_j)} \min_{i=1,2} Q_{\theta_i}(s_j, a_j)}$ and $\nu \in \mathbb{R}^+$ is the normalization coefficient. We train PAR_BC for $5 \times 10^5$ gradient steps, and let it interact with the source domain for 500K steps.

We further provide the following baseline methods that treat the source domain and the target domain as one mixed domain.

**SAC_BC:** This baseline introduces an additional behavior cloning term to the target domain data and uses the SAC as the base algorithm. Its policy is optimized by:

$$\mathcal{L}_{\text{actor}} = \lambda \cdot \mathbb{E}_{\substack{s \sim \mathcal{D}_{\text{src}} \cup \mathcal{D}_{\text{tar}}, \\ a \sim \pi_\phi(\cdot|s)}} \left[ \min_{i=1,2} Q_{\theta_i}(s,a) - \alpha \log \pi_\phi(\cdot|s) \right] + \mathbb{E}_{\substack{(s,a) \sim \mathcal{D}_{\text{tar}}, \\ \hat{a} \sim \pi_\phi(\cdot|s)}} [(a - \hat{a})^2], \quad (19)$$

where $\lambda = \dfrac{\nu}{\frac{1}{N} \sum_{(s_j, a_j)} \min_{i=1,2} Q_{\theta_i}(s_j, a_j)}$ and $\nu \in \mathbb{R}^+$ is the normalization coefficient. We train SAC_BC for 500K gradient steps and let it gather experiences in the source domain for 500K steps.

**SAC_CQL:** This baseline introduces the CQL loss to the target domain offline data and apply the SAC loss to the online source domain data. Its critic optimization formula is given by:

$$\mathcal{L}_{\text{critic}} = \mathbb{E}_{\mathcal{D}_{\text{src}} \cup \mathcal{D}_{\text{tar}}} \left[ (Q_{\theta_i}(s,a) - y)^2 \right] \\ + \beta_{\text{CQL}} \left( \mathbb{E}_{s \sim \mathcal{D}_{\text{tar}}, \tilde{a} \sim \pi_\phi(\cdot|s)}[Q_{\theta_i}(s, \tilde{a})] - \mathbb{E}_{\mathcal{D}_{\text{tar}}}[Q_{\theta_i}(s,a)] \right), i \in \{1, 2\}, \quad (20)$$

where $\beta_{\text{CQL}}$ is the CQL penalty coefficient. Similarly, this method is trained for 500K gradient steps and can interact with the source domain for 500K environmental steps.

**SAC_MCQ:** This method introduces the MCQ loss to the offline target domain samples and adopts the vanilla SAC loss for the online source domain transitions, i.e.,

$$\mathcal{L}_{\text{critic}} = \mathbb{E}_{\mathcal{D}_{\text{src}}} \left[ (Q_{\theta_i}(s,a) - y)^2 \right] + \lambda \mathbb{E}_{\mathcal{D}_{\text{tar}}} \left[ (Q_{\theta_i}(s,a) - y)^2 \right] \\ + (1 - \lambda) \mathbb{E}_{s \sim \mathcal{D}_{\text{tar}}, \tilde{a} \sim \pi_\phi(\cdot|s)} [(Q_{\theta_i}(s, \tilde{a}) - y')^2], i \in \{1, 2\}, \quad (21)$$

where $y' = \min_{j=1,2} \mathbb{E}_{\{a'_i\}^N \sim \hat{\mu}} \left[ \max_{a' \sim \{a'_i\}^N} Q_{\theta_j}(s, a') \right]$ is the pseudo target value given by MCQ. We train SAC_MCQ for 500K gradient steps and allow it to collect transitions in the source domain for 500K environmental steps.

### B.4 Offline-Offline setting

**DARA:** DARA [41] is exactly the offline version of DARC. It also trains the domain classifiers to calculate the reward penalty term $\delta r$ in Equation 1 and corrects the rewards in the source domain dataset via:

$$\hat{r}_{\text{DARA}} = r - \lambda \times \delta_r, \quad (22)$$

where $\lambda$ is the reward penalty coefficient that controls the strengths of the penalty, which is set to be 0.1 by default. After that, DARA is trained on data from both domains for 500K gradient steps. We implement DARA by following the original paper and clip the reward penalty term to lie in $[-10, 10]$. We use IQL [39] as the base algorithm for DARA.

**BOSA:** BOSA [42] identifies that there may exist the distribution shift issue when learning policies based on datasets from two domains with dynamics mismatch. It handles the OOD state actions problem through a supported policy optimization and addresses the OOD dynamics issue through a supported value optimization. To be specific, the critics in BOSA are updated via:

$$\mathcal{L}_{\text{critic}} = \mathbb{E}_{(s,a) \sim \mathcal{D}_{\text{src}}}[Q_{\theta_i}(s,a)] \\ + \mathbb{E}_{(s,a,r,s') \sim \mathcal{D}_{\text{src}} \cup \mathcal{D}_{\text{tar}}, a' \sim \pi_\phi(\cdot|s)} \left[ \mathbb{1}(\hat{P}_{\text{tar}}(s'|s,a)) > \epsilon)(Q_{\theta_i}(s,a) - y)^2 \right], \quad (23)$$

where $\mathbb{1}(\cdot)$ is the indicator function, $\hat{P}_{\text{tar}}(s'|s,a) = \arg\max \mathbb{E}_{(s,a,s')\sim\mathcal{D}_{\text{tar}}}[\log \hat{P}_{\text{tar}}(s'|s,a)]$ is the estimated target domain transition dynamics, $\epsilon$ is the selection threshold, $i \in \{1,2\}$. The policy in BOSA is updated via another supported optimization objective:

$$\mathcal{L}_{\text{actor}} = \mathbb{E}_{s\sim\mathcal{D}_{\text{src}}\cup\mathcal{D}_{\text{tar}}, a\sim\pi_\phi(s)}[Q_{\theta_i}(s,a)], \;\; \text{s.t.} \;\; \mathbb{E}_{s\sim\mathcal{D}_{\text{src}}\cup\mathcal{D}_{\text{tar}}}[\hat{\pi}_{\phi_{\text{mix}}}(\pi_\phi(\text{s})|\text{s})] > \epsilon' \tag{24}$$

where $\epsilon'$ is the policy selection threshold, $\hat{\pi}_{\phi_{\text{mix}}}$ is the learned behavior policy of the mixed dataset $\mathcal{D}_{\text{src}} \cup \mathcal{D}_{\text{tar}}$. The above objective is transformed to a relaxed Lagrangian form for optimization. Both the behavior policy and the dynamics models are modeled by the CVAE. We implement BOSA by following the instructions in the original paper. BOSA is trained for 500K gradient steps in practice by drawing samples from both the source domain dataset and the target domain dataset.

Moreover, we provide the following two baselines.

**IQL:** IQL [39] is a popular offline RL algorithm that learns the policy completely within the span of the offline datasets. It learns the state value function and state-action value function simultaneously by expectile regression:

$$\mathcal{L}_V = \mathbb{E}_{(s,a)\sim\mathcal{D}_{\text{src}}\cup\mathcal{D}_{\text{tar}}}[L_2^\tau(Q_{\theta'}(s,a) - V_\psi(s))], \tag{25}$$

where $L_2^\tau(u) = |\tau - \mathbb{1}(u<0)|u^2$, $\mathbb{1}(\cdot)$ is the indicator function, $\theta'$ is the target network parameter. The state-action value function is then updated by:

$$\mathcal{L}_Q = \mathbb{E}_{(s,a,r,s')\sim\mathcal{D}_{\text{src}}\cup\mathcal{D}_{\text{tar}}}[(r(s,a) + \gamma V_\psi(s') - Q_\theta(s,a))^2]. \tag{26}$$

We then can obtain the advantage function $A(s,a) = Q(s,a) - V(s)$. The policy is optimized by the advantage-weighted behavior cloning:

$$\mathcal{L}_{\text{actor}} = \mathbb{E}_{(s,a)\sim\mathcal{D}_{\text{src}}\cup\mathcal{D}_{\text{tar}}}\left[\exp(\beta \times A(s,a)) \log \pi_\phi(a|s)\right], \tag{27}$$

where $\beta$ is the inverse temperature coefficient. We implement IQL by following its official codebase[11] and also train the IQL agent on offline datasets from both domains for 500K gradient steps.

**TD3_BC:** TD3_BC [20] is a simple but effective offline RL approach that incorporates an additional behavior cloning term to the objective function of the vanilla TD3, which gives:

$$\mathcal{L}_{\text{actor}} = \lambda \cdot \mathbb{E}_{s\sim\mathcal{D}_{\text{src}}\cup\mathcal{D}_{\text{tar}}}\left[\min_{i=1,2} Q_{\theta_i}(s,a)\right] + \mathbb{E}_{(s,a)\sim\mathcal{D}_{\text{src}}\cup\mathcal{D}_{\text{tar}}}[(a-\pi_\phi(s))^2], \tag{28}$$

where $\lambda = \dfrac{\nu}{\frac{1}{N}\sum_{(s_j,a_j)}\min_{i=1,2} Q_{\theta_i}(s_j,a_j)}$ and $\nu \in \mathbb{R}^+$ is the normalization coefficient. We implement TD3_BC by following its official codebase[12]. Similarly, TD3_BC is trained for 500K gradient steps by leveraging offline datasets from both domains.

## C  Hyperparameters

In this section, we elaborate on the specific hyperparameter configurations for our implemented algorithms. Table 3 outlines the hyperparameters for algorithms under the **Online-Online** setting, Table 4 for the **Offline-Online** setting, Table 5 for the **Online-Offline** setting, and Table 6 for the **Offline-Offline** setting.

To ensure a fair and consistent comparison, most of the algorithms utilize similar hyperparameters, such as the actor learning rate and neural network structures. While these settings may slightly deviate from those in the original papers, our preliminary experiments on selected tasks have indicated that these modifications have a negligible impact on performance. Generally, we employ a batch size of 256. For the fine-tuning method, SAC_tune, we use a batch size of 256 for both the source and target domain training. For other methods, we use a batch size of 128 each for the source and target domain data, since these methods concurrently sample from the replay buffers of both domains for policy training. This symmetric sampling technique has also been adopted in previous studies [3, 44, 76].

---

[11] https://github.com/ikostrikov/implicit_q_learning
[12] https://github.com/sfujim/TD3_BC

Table 3: **Hyperparameter setup for Online-Online algorithms.** For consistency, most of these methods share many hyperparameter setup.

| Hyperparameter | Value |
| --- | --- |
| Shared (SAC) | |
|     Actor network | $(256, 256)$ |
|     Critic network | $(256, 256)$ |
|     Learning rate | $3 \times 10^{-4}$ |
|     Optimizer | Adam [38] |
|     Discount factor | 0.99 |
|     Replay buffer size | $10^6$ |
|     Warmup steps | 256 |
|     Nonlinearity | ReLU |
|     Target update rate | $5 \times 10^{-3}$ |
|     Temperature coefficient | 0.2 |
|     Maximum log std | 2 |
|     Minimum log std | $-20$ |
| SAC_tune | |
|     Source domain Batch size | 256 |
|     Target domain Batch size | 256 |
| DARC and SAC_IW | |
|     Classifier Network | $(256, 256)$ |
|     Source domain Batch size | 256 |
|     Target domain Batch size | 256 |
|     Target domain interaction interval | 10 |
| VGDF | |
|     Dynamics model network | $(200, 200, 200, 200, 200)$ |
|     Ensemble size | 7 |
|     Data selection ratio | 25% |
|     Source domain Batch size | 128 |
|     Target domain Batch size | 128 |
|     Exploration policy network | $(256, 256)$ |
|     Target domain interaction interval | 10 |
| PAR | |
|     Encoder Network | $(256, 256)$ |
|     Representation dimension | 256 |
|     Reward penalty coefficient $\beta$ | 0.1 |
|     Source domain Batch size | 128 |
|     Target domain Batch size | 128 |
|     Target domain interaction interval | 10 |

Table 4: **Hyperparameter setup for Offline-Online algorithms.**

| Hyperparameter | Value |
| --- | --- |
| Shared | |
|     Actor network | $(256, 256)$ |
|     Critic network | $(256, 256)$ |
|     Learning rate | $3 \times 10^{-4}$ |
|     Optimizer | Adam [38] |
|     Discount factor | 0.99 |
|     Replay buffer size | $10^6$ |
|     Warmup steps | 256 |
|     Nonlinearity | ReLU |
|     Target update rate | $5 \times 10^{-3}$ |
|     Temperature coefficient | 0.2 |
|     Maximum log std | 2 |
|     Minimum log std | $-20$ |
|     Source domain Batch size | 128 |
|     Target domain Batch size | 128 |
|     Target domain interaction interval | 10 |
| H2O | |
|     Classifier Network | $(256, 256)$ |
|     CQL penalty coefficient $\beta_{\mathrm{CQL}}$ | 10.0 |
| BC_VGDF | |
|     Dynamics model network | $(200, 200, 200, 200, 200)$ |
|     Ensemble size | 7 |
|     Data selection ratio | 25% |
|     Normalization coefficient $\nu$ | 5.0 |
| BC_PAR | |
|     Encoder Network | $(256, 256)$ |
|     Representation dimension | 256 |
|     Reward penalty coefficient $\beta$ | 0.1 |
|     Normalization coefficient $\nu$ | 5.0 |
| CQL_SAC | |
|     CQL penalty coefficient $\beta_{\mathrm{CQL}}$ | 10.0 |
| BC_SAC | |
|     Normalization coefficient $\nu$ | 5.0 |
| MCQ_SAC | |
|     CVAE encoder hidden dimension | 750 |
|     CVAE decoder hidden dimension | 750 |
|     CVAE hidden layers | 3 |
|     CVAE learning rate | $1 \times 10^{-3}$ |
|     CVAE latent dimension | $2 \times |\mathcal{A}|$ |
|     Number of sampled actions | 10 |
|     weighting coefficient $\lambda$ | 0.8 |
| RLPD | |
|     Critic ensemble size | 10 |
|     Update-to-data ratio | 1 |
|     Entropy backup | True for locomotion tasks and False otherwise |
|     Clipped double Q-learning | False for AntMaze tasks and True otherwise |

Table 5: **Hyperparameter setup for Online-Offline algorithms.**

| Hyperparameter | Value |
|---|---|
| Shared | |
|     Actor network | $(256, 256)$ |
|     Critic network | $(256, 256)$ |
|     Learning rate | $3 \times 10^{-4}$ |
|     Optimizer | Adam [38] |
|     Discount factor | 0.99 |
|     Replay buffer size | $10^6$ |
|     Warmup steps | 256 |
|     Nonlinearity | ReLU |
|     Target update rate | $5 \times 10^{-3}$ |
|     Temperature coefficient | 0.2 |
|     Maximum log std | 2 |
|     Minimum log std | $-20$ |
|     Source domain Batch size | 128 |
|     Target domain Batch size | 128 |
|     Source domain interaction interval | 1 |
| H2O | |
|     Classifier Network | $(256, 256)$ |
|     CQL penalty coefficient $\beta_{\mathrm{CQL}}$ | 0.01 |
|     Number of sampled states | 10 |
| PAR_BC | |
|     Encoder Network | $(256, 256)$ |
|     Representation dimension | 256 |
|     Reward penalty coefficient $\beta$ | 0.1 |
|     Normalization coefficient $\nu$ | 5.0 |
| SAC_CQL | |
|     CQL penalty coefficient $\beta_{\mathrm{CQL}}$ | 10.0 |
| SAC_BC | |
|     Normalization coefficient $\nu$ | 5.0 |
| SAC_MCQ | |
|     CVAE encoder hidden dimension | 750 |
|     CVAE decoder hidden dimension | 750 |
|     CVAE hidden layers | 3 |
|     CVAE learning rate | $1 \times 10^{-3}$ |
|     CVAE latent dimension | $2 \times |\mathcal{A}|$ |
|     Number of sampled actions | 10 |
|     weighting coefficient $\lambda$ | 0.8 |

Table 6: **Hyperparameter setup for Offline-Offline algorithms.**

| Hyperparameter | Value |
|---|---|
| **Shared** | |
| Actor network | $(256, 256)$ |
| Critic network | $(256, 256)$ |
| Learning rate | $3 \times 10^{-4}$ |
| Optimizer | Adam [38] |
| Discount factor | 0.99 |
| Replay buffer size | $10^6$ |
| Nonlinearity | ReLU |
| Target update rate | $5 \times 10^{-3}$ |
| Source domain Batch size | 128 |
| Target domain Batch size | 128 |
| **DARA** | |
| Temperature coefficient | 0.2 |
| Maximum log std | 2 |
| Minimum log std | $-20$ |
| Classifier Network | $(256, 256)$ |
| Reward penalty coefficient $\lambda$ | 0.1 |
| **BOSA** | |
| Temperature coefficient | 0.2 |
| Maximum log std | 2 |
| Minimum log std | $-20$ |
| Policy regularization coefficient $\lambda_{\text{policy}}$ | 0.1 |
| Transition coefficient $\lambda_{\text{transition}}$ | 0.1 |
| Threshold parameter $\epsilon, \epsilon'$ | $\log(0.01)$ |
| Value wight $\omega$ | 0.1 |
| CVAE ensemble size | 1 for the behavior policy, 5 for the dynamics model |
| **IQL** | |
| Temperature coefficient | 0.2 |
| Maximum log std | 2 |
| Minimum log std | $-20$ |
| Inverse temperature parameter $\beta$ | 10.0 (AntMaze), 3.0 (locomotion), 0.5 (Adroit) |
| Expectile parameter $\tau$ | 0.9 for AntMaze tasks and 0.7 otherwise |
| **TD3_BC** | |
| Normalization coefficient $\nu$ | 2.5 |

# D  Benchmark Task Implementation Details

In this section, we offer a comprehensive explanation of the modifications made to the `xml` files to create our benchmark tasks. Additionally, we present a complete list of supported benchmark tasks, along with their visualization results. We restate our naming convention for the benchmark tasks as follows: `[domain]-[shift_type]-[shift_part (optional)]-[shift_level]`, where `domain` can be one of *ant, halfcheetah, hopper, walker2d*, and `shift_type` can be one of *friction, gravity, kinematic, morph* (with morph representing morphology shifts) for locomotion tasks. For AntMaze tasks, the `shift_type` can be one of *small, medium, large*, and for dexterous hand manipulation tasks, the `shift_type` can be *broken-joint* or *shrink-finger*. The optional `shift_part` indicates the specific body part affected by the shift, and `shift_level` can be one of $\{0.1, 0.5, 2.0, 5.0\}$ for friction and gravity shift tasks, or *easy, medium, hard* for the remaining locomotion and manipulation tasks. For AntMaze tasks, the `shift_level` can be *centerblock, empty, lshape, zshape, reversel, reverseu* for the small-size maze, and *1, 2, 3, 4, 5, 6* for medium-size and large-size mazes. For example, *halfcheetah-gravity-2.0* signifies that the target domain has twice the gravity of the source domain, while *ant-kinematic-hipjnt-hard* indicates that the hip joint in the *ant* robot is severely broken (hard level).

Table 7: **Full list of supported task names in the locomotion domain.**

| friction | gravity |
|---|---|
| ant-friction-0.1 | ant-gravity-0.1 |
| ant-friction-0.5 | ant-gravity-0.5 |
| ant-friction-2.0 | ant-gravity-2.0 |
| ant-friction-5.0 | ant-gravity-5.0 |
| halfcheetah-friction-0.1 | halfcheetah-gravity-0.1 |
| halfcheetah-friction-0.5 | halfcheetah-gravity-0.5 |
| halfcheetah-friction-2.0 | halfcheetah-gravity-2.0 |
| halfcheetah-friction-5.0 | halfcheetah-gravity-5.0 |
| hopper-friction-0.1 | hopper-gravity-0.1 |
| hopper-friction-0.5 | hopper-gravity-0.5 |
| hopper-friction-2.0 | hopper-gravity-2.0 |
| hopper-friction-5.0 | hopper-gravity-5.0 |
| walker2d-friction-0.1 | walker2d-gravity-0.1 |
| walker2d-friction-0.5 | walker2d-gravity-0.5 |
| walker2d-friction-2.0 | walker2d-gravity-2.0 |
| walker2d-friction-5.0 | walker2d-gravity-5.0 |

| kinematic | morphology |
|---|---|
| ant-kinematic-hipjnt-easy | ant-morph-halflegs-easy |
| ant-kinematic-hipjnt-medium | ant-morph-halflegs-medium |
| ant-kinematic-hipjnt-hard | ant-morph-halflegs-hard |
| ant-kinematic-anklejnt-easy | ant-morph-alllegs-easy |
| ant-kinematic-anklejnt-medium | ant-morph-alllegs-medium |
| ant-kinematic-anklejnt-hard | ant-morph-alllegs-hard |
| halfcheetah-kinematic-footjnt-easy | halfcheetah-morph-thigh-easy |
| halfcheetah-kinematic-footjnt-medium | halfcheetah-morph-thigh-medium |
| halfcheetah-kinematic-footjnt-hard | halfcheetah-morph-thigh-hard |
| halfcheetah-kinematic-thighjnt-easy | halfcheetah-morph-torso-easy |
| halfcheetah-kinematic-thighjnt-medium | halfcheetah-morph-torso-medium |
| halfcheetah-kinematic-thighjnt-hard | halfcheetah-morph-torso-hard |
| hopper-kinematic-footjnt-easy | hopper-morph-foot-easy |
| hopper-kinematic-footjnt-medium | hopper-morph-foot-medium |
| hopper-kinematic-footjnt-hard | hopper-morph-foot-hard |
| hopper-kinematic-legjnt-easy | hopper-morph-torso-easy |
| hopper-kinematic-legjnt-medium | hopper-morph-torso-medium |
| hopper-kinematic-legjnt-hard | hopper-morph-torso-hard |
| walker2d-kinematic-footjnt-easy | walker2d-morph-leg-easy |
| walker2d-kinematic-footjnt-medium | walker2d-morph-leg-medium |
| walker2d-kinematic-footjnt-hard | walker2d-morph-leg-hard |
| walker2d-kinematic-thighjnt-easy | walker2d-morph-torso-easy |
| walker2d-kinematic-thighjnt-medium | walker2d-morph-torso-medium |
| walker2d-kinematic-thighjnt-hard | walker2d-morph-torso-hard |

## D.1 Locomotion Tasks

Locomotion tasks consist of four dynamic shift categories: friction, gravity, kinematic, and morphology shifts. ODRL supports friction and gravity shift tasks with four shift levels $\{0.1, 0.5, 2.0, 5.0\}$, as well as kinematic and morphology shift tasks with three shift levels (*easy, medium, hard*). For each kinematic and morphology task, we provide two options that involve different body parts being broken or having altered shapes. This results in a total of 80 tasks, with the full list of names available in Table 7.

### D.1.1 Friction shift

The friction shift is implemented by altering the `friction` attribute in the `geom` elements. The frictional components are adjusted to 0.1, 0.5, 2.0, and 5.0 times the frictional components in the source domain, respectively. For instance, to achieve a 5.0x friction for the *hopper* robot, we make the following modifications. Please refer to our provided `xml` files for more details on other tasks. It

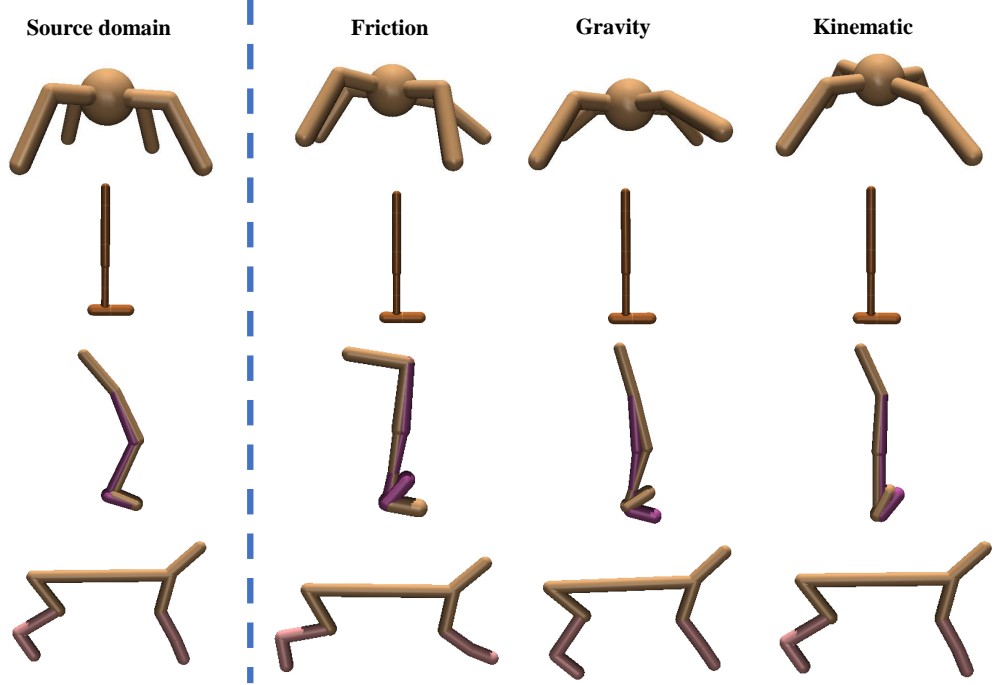

Figure 7: **Visualization results of locomotion tasks.** We present the simulated robot under friction shift, gravity shift and kinematic shift. The simulated robots remain identical with the robots in the source domain in appearance.

is important to note that the simulated robots under the friction shift, as visualized in Figure 7, appear identical to the original robots since only the frictional force components are modified.

```
# torso
<geom friction="4.5" fromto="0 0 1.45 0 0 1.05" name="torso_geom" size="0.05" type="
    capsule"/>
# thigh
<geom friction="4.5" fromto="0 0 1.05 0 0 0.6" name="thigh_geom" size="0.05" type="
    capsule"/>
# leg
<geom friction="4.5" fromto="0 0 0.6 0 0 0.1" name="leg_geom" size="0.04" type="
    capsule"/>
# foot
<geom friction="10.0" fromto="-0.13 0 0.1 0.26 0 0.1" name="foot_geom" size="0.06"
    type="capsule"/>
```

### D.1.2 Gravity shift

We modify the gravity of the simulated robots by revising the `gravity` attribute in the `option` element. Note that some simulated robots do not contain the `<option gravity="0 0 -9.81" />` code, and we manually add it. For example, the gravity of the *halfcheetah* robot in the target domain is revised to 0.5 times the gravity in the source domain with the following code.

```
# gravity
<option gravity="0 0 -4.905" timestep="0.01"/>
```

We also present the visualization results of robots under the gravity shifts in Figure 7.

### D.1.3 Kinematic shift

As for the kinematic shifts, one can see the visualization results of locomotion tasks in Figure 7. The simulated robot remains identical as that in the source domain, and the limbs, foot or some other

body parts are broken. It is worth noting that the rotation range of the specified joint is modified to different values under different shift levels. For each task,

- *ant-kinematic-hipjnt-easy:* the rotation range of the hip joint is modified from $[-30, 30]$ to $[-24, 24]$:

```
<joint axis="0 0 1" name="hip_1" pos="0.0 0.0 0.0" range="-24 24" type="
    hinge"/>
<joint axis="0 0 1" name="hip_2" pos="0.0 0.0 0.0" range="-24 24" type="
    hinge"/>
<joint axis="0 0 1" name="hip_3" pos="0.0 0.0 0.0" range="-24 24" type="
    hinge"/>
<joint axis="0 0 1" name="hip_4" pos="0.0 0.0 0.0" range="-24 24" type="
    hinge"/>
```

- *ant-kinematic-hipjnt-medium:* the rotation range of the hip joint is modified from $[-30, 30]$ to $[-15, 15]$:

```
<joint axis="0 0 1" name="hip_1" pos="0.0 0.0 0.0" range="-15 15" type="
    hinge"/>
<joint axis="0 0 1" name="hip_2" pos="0.0 0.0 0.0" range="-15 15" type="
    hinge"/>
<joint axis="0 0 1" name="hip_3" pos="0.0 0.0 0.0" range="-15 15" type="
    hinge"/>
<joint axis="0 0 1" name="hip_4" pos="0.0 0.0 0.0" range="-15 15" type="
    hinge"/>
```

- *ant-kinematic-hipjnt-hard:* the rotation range of the hip joint is modified from $[-30, 30]$ to $[-6, 6]$:

```
<joint axis="0 0 1" name="hip_1" pos="0.0 0.0 0.0" range="-6 6" type="
    hinge"/>
<joint axis="0 0 1" name="hip_2" pos="0.0 0.0 0.0" range="-6 6" type="
    hinge"/>
<joint axis="0 0 1" name="hip_3" pos="0.0 0.0 0.0" range="-6 6" type="
    hinge"/>
<joint axis="0 0 1" name="hip_4" pos="0.0 0.0 0.0" range="-6 6" type="
    hinge"/>
```

- *ant-kinematic-anklejnt-easy:* the the rotation range of the ankle joint is modified from $[30, 70]$ to $[30, 62]$:

```
<joint axis="-1 1 0" name="ankle_1" pos="0.0 0.0 0.0" range="30 62" type
    ="hinge"/>
<joint axis="1 1 0" name="ankle_2" pos="0.0 0.0 0.0" range="-62 -30"
    type="hinge"/>
<joint axis="-1 1 0" name="ankle_3" pos="0.0 0.0 0.0" range="-62 -30"
    type="hinge"/>
<joint axis="1 1 0" name="ankle_4" pos="0.0 0.0 0.0" range="30 62" type=
    "hinge"/>
```

- *ant-kinematic-anklejnt-medium:* the the rotation range of the ankle joint is modified from $[30, 70]$ to $[30, 50]$:

```
<joint axis="-1 1 0" name="ankle_1" pos="0.0 0.0 0.0" range="30 50" type
    ="hinge"/>
<joint axis="1 1 0" name="ankle_2" pos="0.0 0.0 0.0" range="-50 -30"
    type="hinge"/>
<joint axis="-1 1 0" name="ankle_3" pos="0.0 0.0 0.0" range="-50 -30"
    type="hinge"/>
<joint axis="1 1 0" name="ankle_4" pos="0.0 0.0 0.0" range="30 50" type=
    "hinge"/>
```

- *ant-kinematic-anklejnt-hard:* the the rotation range of the ankle joint is modified from $[30, 70]$ to $[30, 38]$:

```
<joint axis="-1 1 0" name="ankle_1" pos="0.0 0.0 0.0" range="30 38" type
    ="hinge"/>
<joint axis="1 1 0" name="ankle_2" pos="0.0 0.0 0.0" range="-38 -30"
    type="hinge"/>
<joint axis="-1 1 0" name="ankle_3" pos="0.0 0.0 0.0" range="-38 -30"
    type="hinge"/>
<joint axis="1 1 0" name="ankle_4" pos="0.0 0.0 0.0" range="30 38" type=
    "hinge"/>
```

- *halfcheetah-kinematic-foojnt-easy*: the rotation range of the foot joint is modified to be 0.8 times of foot joint's rotation range in the source domain:

```
<joint axis="0 1 0" damping="3" name="bfoot" pos="0 0 0" range="-.32
    .628" stiffness="120" type="hinge"/>
<joint axis="0 1 0" damping="1.5" name="ffoot" pos="0 0 0" range="-.4 .4
    " stiffness="60" type="hinge"/>
```

- *halfcheetah-kinematic-foojnt-medium*: the rotation range of the foot joint is modified to be 0.5 times of foot joint's rotation range in the source domain:

```
<joint axis="0 1 0" damping="3" name="bfoot" pos="0 0 0" range="-.2
    .3925" stiffness="120" type="hinge"/>
<joint axis="0 1 0" damping="1.5" name="ffoot" pos="0 0 0" range="-.25
    .25" stiffness="60" type="hinge"/>
```

- *halfcheetah-kinematic-foojnt-hard*: the rotation range of the foot joint is modified to be:

```
<joint axis="0 1 0" damping="3" name="bfoot" pos="0 0 0" range="-.08
    .157" stiffness="120" type="hinge"/>
<joint axis="0 1 0" damping="1.5" name="ffoot" pos="0 0 0" range="-.1 .1
    " stiffness="60" type="hinge"/>
```

- *halfcheetah-kinematic-thighjnt-easy:* the rotation range of the thigh joint is modified to be 0.8 times of that in the source domain:

```
<joint axis="0 1 0" damping="6" name="bthigh" pos="0 0 0" range="-.416
    .84" stiffness="240" type="hinge"/>
<joint axis="0 1 0" damping="4.5" name="fthigh" pos="0 0 0" range="-.8
    .56" stiffness="180" type="hinge"/>
```

- *halfcheetah-kinematic-thighjnt-medium:* the rotation range of the thigh joint is modified to be 0.5 times of that in the source domain:

```
<joint axis="0 1 0" damping="6" name="bthigh" pos="0 0 0" range="-.26
    .525" stiffness="240" type="hinge"/>
<joint axis="0 1 0" damping="4.5" name="fthigh" pos="0 0 0" range="-.5
    .35" stiffness="180" type="hinge"/>
```

- *halfcheetah-kinematic-thighjnt-hard:* the rotation range of the thigh joint is modified to be 0.2 times of that in the source domain:

```
<joint axis="0 1 0" damping="6" name="bthigh" pos="0 0 0" range="-.104
    .21" stiffness="240" type="hinge"/>
<joint axis="0 1 0" damping="4.5" name="fthigh" pos="0 0 0" range="-.2
    .14" stiffness="180" type="hinge"/>
```

- *hopper-kinematic-footjnt-easy:* the rotation range of the foot joint is modified from $[-45, 45]$ to $[-36, 36]$:

```
<joint axis="0 -1 0" name="foot_joint" pos="0 0 0.1" range="-36 36" type
    ="hinge"/>
```

- *hopper-kinematic-footjnt-medium:* the rotation range of the foot joint is modified from $[-45, 45]$ to $[-22.5, 22.5]$:

```xml
<joint axis="0 -1 0" name="foot_joint" pos="0 0 0.1" range="-22.5 22.5"
    type="hinge"/>
```

- *hopper-kinematic-footjnt-hard:* the rotation range of the foot joint is modified from $[-45, 45]$ to $[-9, 9]$:

```xml
<joint axis="0 -1 0" name="foot_joint" pos="0 0 0.1" range="-9 9" type="
    hinge"/>
```

- *hopper-kinematic-legjnt-easy:* the rotation range of the leg joint is modified from $[-150, 0]$ to $[-120, 0]$:

```xml
<joint axis="0 -1 0" name="leg_joint" pos="0 0 0.6" range="-120 0" type=
    "hinge"/>
```

- *hopper-kinematic-legjnt-medium:* the rotation range of the leg joint is modified from $[-150, 0]$ to $[-75, 0]$:

```xml
<joint axis="0 -1 0" name="leg_joint" pos="0 0 0.6" range="-75 0" type="
    hinge"/>
```

- *hopper-kinematic-legjnt-hard:* the rotation range of the leg joint is modified from $[-150, 0]$ to $[-30, 0]$:

```xml
<joint axis="0 -1 0" name="leg_joint" pos="0 0 0.6" range="-30 0" type="
    hinge"/>
```

- *walker2d-kinematic-footjnt-easy:* the rotation range of the foot joint is modified from $[-45, 45]$ to $[-36, 36]$:

```xml
<joint axis="0 -1 0" name="foot_joint" pos="0 0 0.1" range="-36 36" type
    ="hinge"/>
<joint axis="0 -1 0" name="foot_left_joint" pos="0 0 0.1" range="-36 36"
    type="hinge"/>
```

- *walker2d-kinematic-footjnt-medium:* the rotation range of the foot joint is modified from $[-45, 45]$ to $[-22.5, 22.5]$:

```xml
<joint axis="0 -1 0" name="foot_joint" pos="0 0 0.1" range="-22.5 22.5"
    type="hinge"/>
<joint axis="0 -1 0" name="foot_left_joint" pos="0 0 0.1" range="-22.5
    22.5" type="hinge"/>
```

- *walker2d-kinematic-footjnt-hard:* the rotation range of the foot joint is modified from $[-45, 45]$ to $[-9, 9]$:

```xml
<joint axis="0 -1 0" name="foot_joint" pos="0 0 0.1" range="-9 9" type="
    hinge"/>
<joint axis="0 -1 0" name="foot_left_joint" pos="0 0 0.1" range="-9 9"
    type="hinge"/>
```

- *walker2d-kinematic-thighjnt-easy:* the rotation range of the thigh joint is modified from $[-150, 0]$ to $[-120, 0]$:

```xml
<joint axis="0 -1 0" name="thigh_joint" pos="0 0 1.05" range="-120 0"
    type="hinge"/>
<joint axis="0 -1 0" name="thigh_left_joint" pos="0 0 1.05" range="-120
    0" type="hinge"/>
```

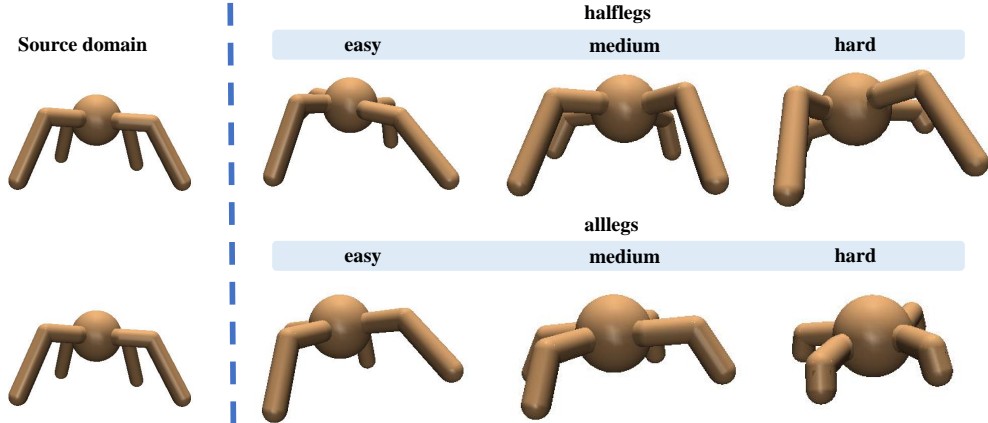

Figure 8: **Visualization results of the morphology shifts in the *ant* task.** We consider morphology shifts in the legs, where the front two legs or all legs become shorter.

- *walker2d-kinematic-thighjnt-medium:* the rotation range of the thigh joint is modified from $[-150, 0]$ to $[-75, 0]$:

```
<joint axis="0 -1 0" name="thigh_joint" pos="0 0 1.05" range="-75 0"
    type="hinge"/>
<joint axis="0 -1 0" name="thigh_left_joint" pos="0 0 1.05" range="-75 0
    " type="hinge"/>
```

- *walker2d-kinematic-thighjnt-hard:* the rotation range of the thigh joint is modified from $[-150, 0]$ to $[-30, 0]$:

```
<joint axis="0 -1 0" name="thigh_joint" pos="0 0 1.05" range="-30 0"
    type="hinge"/>
<joint axis="0 -1 0" name="thigh_left_joint" pos="0 0 1.05" range="-30 0
    " type="hinge"/>
```

### D.1.4 Morphology shift

The morphology shift involves directly altering the visual appearance of the robot. We consider various morphological changes for different simulated robots. The visualization results of these changes are presented as follows: the morphology change of the *ant* task in Figure 8, the morphology change of the *halfcheetah* task in Figure 9, the morphology change of the *hopper* task in Figure 10, and the morphology change of the *walker2d* task in Figure 11. We provide a summary of the specific modifications made for each morphology shift task below.

- *ant-morph-halflegs-easy:* the leg sizes of the front two legs are revised to be 0.8 times of those in the source domain:

```
<geom fromto="0.0 0.0 0.0 0.32 0.32 0.0" name="left_ankle_geom" size="
    0.08" type="capsule"/>
<geom fromto="0.0 0.0 0.0 -0.32 0.32 0.0" name="right_ankle_geom" size="
    0.08" type="capsule"/>
```

- *ant-morph-halflegs-medium:* the leg sizes of the front two legs are revised to be 0.5 times of those in the source domain:

```
<geom fromto="0.0 0.0 0.0 0.2 0.2 0.0" name="left_ankle_geom" size="0.08
    " type="capsule"/>
<geom fromto="0.0 0.0 0.0 -0.2 0.2 0.0" name="right_ankle_geom" size="
    0.08" type="capsule"/>
```

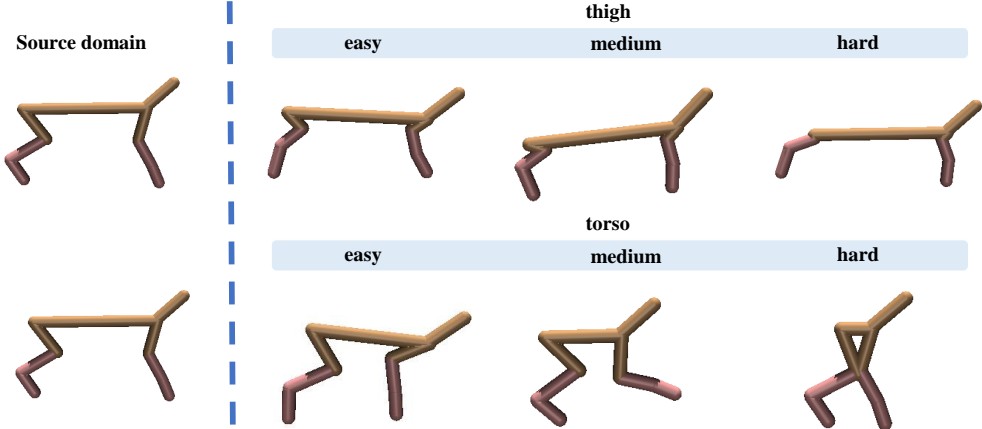

Figure 9: **Visualization results of the morphology shifts in the *halfcheetah* task.** We consider shorter thigh or torso part for the robot.

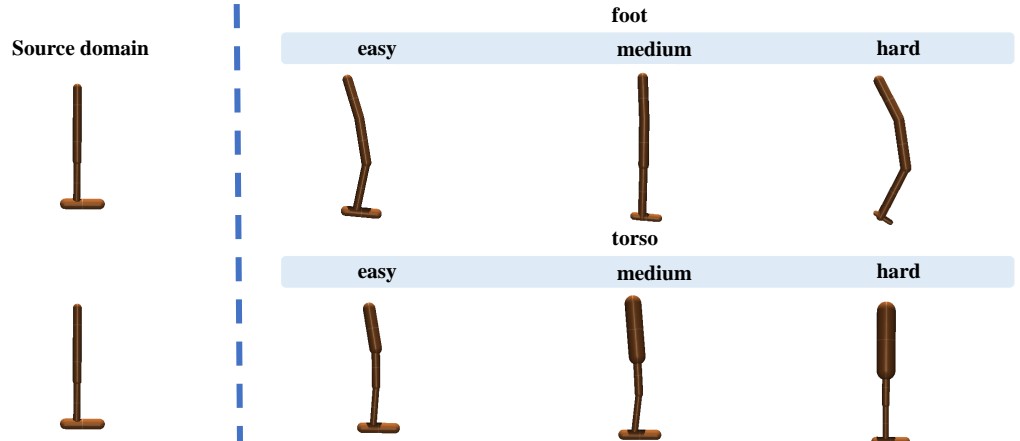

Figure 10: **Visualization results of the morphology shifts in the *hopper* task.** The morphology shift occurs in the foot part or the torso part.

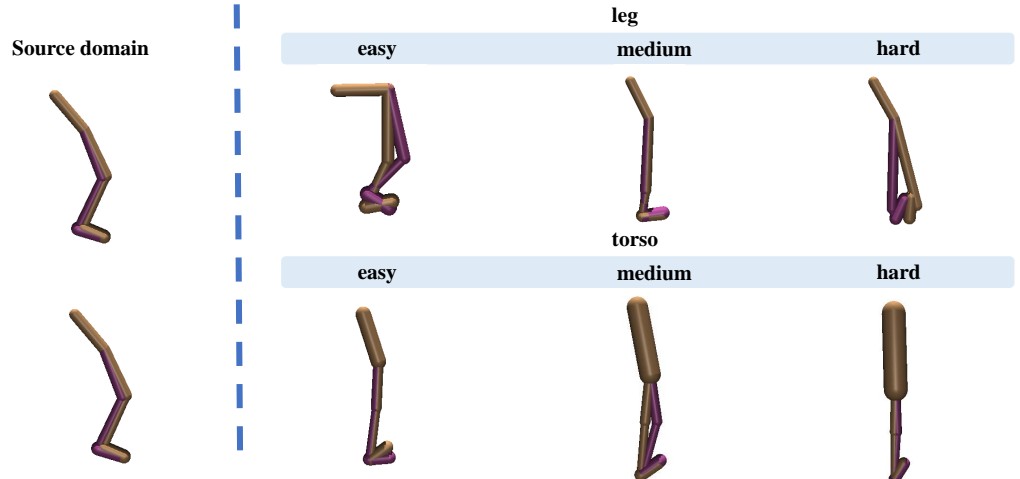

Figure 11: **Visualization results of the morphology shifts in the *walker2d* task.** The morphology mismatch happens at the leg or the torso part of the robot.

- *ant-morph-halflegs-hard:* the leg sizes of the front two legs are revised to be 0.2 times of those in the source domain:

```
<geom fromto="0.0 0.0 0.0 0.08 0.08 0.0" name="left_ankle_geom" size="
    0.08" type="capsule"/>
<geom fromto="0.0 0.0 0.0 -0.08 0.08 0.0" name="right_ankle_geom" size="
    0.08" type="capsule"/>
```

- *ant-morph-alllegs-easy:* the leg sizes of all legs are revised to be 0.8 times of those in the source domain:

```
<geom fromto="0.0 0.0 0.0 0.32 0.32 0.0" name="left_ankle_geom" size="
    0.08" type="capsule"/>
<geom fromto="0.0 0.0 0.0 -0.32 0.32 0.0" name="right_ankle_geom" size="
    0.08" type="capsule"/>
<geom fromto="0.0 0.0 0.0 -0.32 -0.32 0.0" name="third_ankle_geom" size=
    "0.08" type="capsule"/>
<geom fromto="0.0 0.0 0.0 0.32 -0.32 0.0" name="fourth_ankle_geom" size=
    "0.08" type="capsule"/>
```

- *ant-morph-alllegs-medium:* the leg sizes of all legs are revised to be 0.5 times of those in the source domain:

```
<geom fromto="0.0 0.0 0.0 0.2 0.2 0.0" name="left_ankle_geom" size="0.08
    " type="capsule"/>
<geom fromto="0.0 0.0 0.0 -0.2 0.2 0.0" name="right_ankle_geom" size="
    0.08" type="capsule"/>
<geom fromto="0.0 0.0 0.0 -0.2 -0.2 0.0" name="third_ankle_geom" size="
    0.08" type="capsule"/>
<geom fromto="0.0 0.0 0.0 0.2 -0.2 0.0" name="fourth_ankle_geom" size="
    0.08" type="capsule"/>
```

- *ant-morph-alllegs-hard:* the leg sizes of all legs are revised to be 0.2 times of those in the source domain:

```
<geom fromto="0.0 0.0 0.0 0.08 0.08 0.0" name="left_ankle_geom" size="
    0.08" type="capsule"/>
<geom fromto="0.0 0.0 0.0 -0.08 0.08 0.0" name="right_ankle_geom" size="
    0.08" type="capsule"/>
<geom fromto="0.0 0.0 0.0 -0.08 -0.08 0.0" name="third_ankle_geom" size=
    "0.08" type="capsule"/>
<geom fromto="0.0 0.0 0.0 0.08 -0.08 0.0" name="fourth_ankle_geom" size=
    "0.08" type="capsule"/>
```

- *halfcheetah-morph-thigh-easy:* the front thigh size and the back thigh size are modified to be:

```
<geom fromto="0 0 0 0.11 0 -0.11" name="bthigh" size="0.046" type="
    capsule"/>
<body name="bshin" pos="0.11 0 -0.11">
<geom fromto="0 0 0 -.13 0 -.15" name="bshin" rgba="0.9 0.6 0.6 1" size=
    "0.046" type="capsule"/>
<body name="bfoot" pos="-.13 0 -.15">
<geom fromto="0 0 0 -0.09 0 -0.1" name="fthigh" size="0.046" type="
    capsule"/>
<body name="fshin" pos="-0.09 0 -0.1">
<geom fromto="0 0 0 .11 0 -.13" name="fshin" rgba="0.9 0.6 0.6 1" size="
    0.046" type="capsule"/>
<body name="ffoot" pos=".11 0 -.13">
```

- *halfcheetah-morph-thigh-medium:* the front thigh size and the back thigh size are modified to be:

```
    <geom fromto="0 0 0 0.08 0 -0.08" name="bthigh" size="0.046" type="
        capsule"/>
    <body name="bshin" pos="0.08 0 -0.08">
    <geom fromto="0 0 0 -.13 0 -.15" name="bshin" rgba="0.9 0.6 0.6 1" size=
        "0.046" type="capsule"/>
    <body name="bfoot" pos="-.13 0 -.15">
    <geom fromto="0 0 0 -0.07 0 -0.08" name="fthigh" size="0.046" type="
        capsule"/>
    <body name="fshin" pos="-0.07 0 -0.08">
    <geom fromto="0 0 0 .11 0 -.13" name="fshin" rgba="0.9 0.6 0.6 1" size="
        0.046" type="capsule"/>
    <body name="ffoot" pos=".11 0 -.13">
```

- *halfcheetah-morph-thigh-hard:* the front thigh size and the back thigh size are modified to be:

```
    <geom fromto="0 0 0 0.02 0 -0.02" name="bthigh" size="0.046" type="
        capsule"/>
    <body name="bshin" pos="0.02 0 -0.02">
    <geom fromto="0 0 0 -.13 0 -.15" name="bshin" rgba="0.9 0.6 0.6 1" size=
        "0.046" type="capsule"/>
    <body name="bfoot" pos="-.13 0 -.15">
    <geom fromto="0 0 0 -0.04 0 -0.05" name="fthigh" size="0.046" type="
        capsule"/>
    <body name="fshin" pos="-0.04 0 -0.05">
    <geom fromto="0 0 0 .11 0 -.13" name="fshin" rgba="0.9 0.6 0.6 1" size="
        0.046" type="capsule"/>
    <body name="ffoot" pos=".11 0 -.13">
```

- *halfcheetah-morph-torso-easy:* the torso size of the robot is revised to be 0.8 times of that in the source domain:

```
    <geom fromto="-.4 0 0 .4 0 0" name="torso" size="0.046" type="capsule"/>
    <geom axisangle="0 1 0 .87" name="head" pos=".5 0 .1" size="0.046 .15"
        type="capsule"/>
    <body name="bthigh" pos="-.4 0 0">
    <body name="fthigh" pos=".4 0 0">
```

- *halfcheetah-morph-torso-medium:* the torso size of the robot is revised to be 0.5 times of that in the source domain:

```
    <geom fromto="-.25 0 0 .25 0 0" name="torso" size="0.046" type="capsule"
        />
    <geom axisangle="0 1 0 .87" name="head" pos=".35 0 .1" size="0.046 .15"
        type="capsule"/>
    <body name="bthigh" pos="-.25 0 0">
    <body name="fthigh" pos=".25 0 0">
```

- *halfcheetah-morph-torso-hard:* the torso size of the robot is revised to be 0.2 times of that in the source domain:

```
    <geom fromto="-.1 0 0 .1 0 0" name="torso" size="0.046" type="capsule"/>
    <geom axisangle="0 1 0 .87" name="head" pos=".2 0 .1" size="0.046 .15"
        type="capsule"/>
    <body name="bthigh" pos="-.1 0 0">
    <body name="fthigh" pos=".1 0 0">
```

- *hopper-morph-foot-easy:* the foot size is revised to be 0.8 times of that within the source domain:

```
    <geom friction="2.0" fromto="-0.104 0 0.1 0.208 0 0.1" name="foot_geom"
        size="0.048" type="capsule"/>
```

- *hopper-morph-foot-medium:* the foot size is revised to be 0.6 times of that within the source domain:

```
<geom friction="2.0" fromto="-0.078 0 0.1 0.156 0 0.1" name="foot_geom"
    size="0.036" type="capsule"/>
```

- *hopper-morph-foot-hard:* the foot size is revised to be 0.4 times of that within the source domain:

```
<geom friction="2.0" fromto="-0.052 0 0.1 0.104 0 0.1" name="foot_geom"
    size="0.024" type="capsule"/>
```

- *hopper-morph-torso-easy:* the torso size is revised to be 1.5 times of that within the source domain, and the length of the torso becomes 0.48:

```
<geom friction="0.9" fromto="0 0 1.53 0 0 1.05" name="torso_geom" size="
    0.075" type="capsule"/>
```

- *hopper-morph-torso-medium:* the torso size is revised to be 2.0 times of that within the source domain, and the length of the torso becomes 0.64:

```
<geom friction="0.9" fromto="0 0 1.69 0 0 1.05" name="torso_geom" size="
    0.1" type="capsule"/>
```

- *hopper-morph-torso-hard:* the torso size is revised to be 2.5 times of that within the source domain, and the length of the torso becomes 0.8:

```
<geom friction="0.9" fromto="0 0 1.85 0 0 1.05" name="torso_geom" size="
    0.125" type="capsule"/>
```

- *walker2d-morph-leg-easy:* the leg size of the robot is revised to be 0.8 times of that in the source domain (the thigh size becomes larger accordingly):

```
<geom friction="0.9" fromto="0 0 1.05 0 0 0.5" name="thigh_geom" size="
    0.05" type="capsule"/>
<joint axis="0 -1 0" name="leg_joint" pos="0 0 0.5" range="-150 0" type=
    "hinge"/>
<geom friction="0.9" fromto="0 0 0.5 0 0 0.1" name="leg_geom" size="0.04
    " type="capsule"/>
<geom friction="0.9" fromto="0 0 1.05 0 0 0.5" name="thigh_left_geom"
    rgba=".7 .3 .6 1" size="0.05" type="capsule"/>
<joint axis="0 -1 0" name="leg_left_joint" pos="0 0 0.5" range="-150 0"
    type="hinge"/>
<geom friction="0.9" fromto="0 0 0.5 0 0 0.1" name="leg_left_geom" rgba=
    ".7 .3 .6 1" size="0.04" type="capsule"/>
```

- *walker2d-morph-leg-medium:* the leg size of the robot is revised to be 0.5 times of that in the source domain (the thigh size becomes larger accordingly):

```
<geom friction="0.9" fromto="0 0 1.05 0 0 0.35" name="thigh_geom" size="
    0.05" type="capsule"/>
<joint axis="0 -1 0" name="leg_joint" pos="0 0 0.35" range="-150 0" type
    ="hinge"/>
<geom friction="0.9" fromto="0 0 0.35 0 0 0.1" name="leg_geom" size="
    0.04" type="capsule"/>
<geom friction="0.9" fromto="0 0 1.05 0 0 0.35" name="thigh_left_geom"
    rgba=".7 .3 .6 1" size="0.05" type="capsule"/>
<joint axis="0 -1 0" name="leg_left_joint" pos="0 0 0.35" range="-150 0"
    type="hinge"/>
<geom friction="0.9" fromto="0 0 0.35 0 0 0.1" name="leg_left_geom" rgba
    =".7 .3 .6 1" size="0.04" type="capsule"/>
```

- *walker2d-morph-leg-easy:* the leg size of the robot is revised to be 0.2 times of that in the source domain (the thigh size becomes larger accordingly):

```
<geom friction="0.9" fromto="0 0 1.05 0 0 0.2" name="thigh_geom" size="
    0.05" type="capsule"/>
<joint axis="0 -1 0" name="leg_joint" pos="0 0 0.2" range="-150 0" type=
    "hinge"/>
<geom friction="0.9" fromto="0 0 0.2 0 0 0.1" name="leg_geom" size="0.04
    " type="capsule"/>
<geom friction="0.9" fromto="0 0 1.05 0 0 0.2" name="thigh_left_geom"
    rgba=".7 .3 .6 1" size="0.05" type="capsule"/>
<joint axis="0 -1 0" name="leg_left_joint" pos="0 0 0.2" range="-150 0"
    type="hinge"/>
<geom friction="0.9" fromto="0 0 0.2 0 0 0.1" name="leg_left_geom" rgba=
    ".7 .3 .6 1" size="0.04" type="capsule"/>
```

- *walker2d-morph-torso-easy:* the torso size is revised to be 1.5 times of that within the source domain, and the length of the torso becomes 0.48:

```
<geom friction="0.9" fromto="0 0 1.53 0 0 1.05" name="torso_geom" size="
    0.075" type="capsule"/>
```

- *walker2d-morph-torso-medium:* the torso size is revised to be 2.0 times of that within the source domain, and the length of the torso becomes 0.64:

```
<geom friction="0.9" fromto="0 0 1.69 0 0 1.05" name="torso_geom" size="
    0.1" type="capsule"/>
```

- *walker2d-morph-torso-hard:* the torso size is revised to be 2.5 times of that within the source domain, and the length of the torso becomes 0.8:

```
<geom friction="0.9" fromto="0 0 1.85 0 0 1.05" name="torso_geom" size="
    0.125" type="capsule"/>
```

## D.2   AntMaze Task

AntMaze navigation tasks are composed of three different map sizes (*small, medium, large*), with each size featuring six unique map layouts, resulting in a total of 18 tasks. The navigation domain is used to assess the agent's ability to transfer the learned policy to diverse maps and obstacles. For the small maze, the shift level can be *centerblock* (a block in the center of the maze), *empty* (no blocks in the maze), *lshape*, *zshape*, *reverseu*, or *reversel*. For the medium and large size mazes, the shift level can be $\{1, 2, 3, 4, 5, 6\}$, each representing a different type of map layout. The full list of AntMaze task names can be found in Table 8. It's important to note that only the map layouts are changed, with the embodied ant robot remaining unmodified. The AntMaze domain here has a single starting point and a single goal position. The objective is to navigate the ant robot from the bottom left corner of the map to the top right corner. Visualization results for all supported maps are shown in Figure 12.

## D.3   Dexterous Manipulation

The dexterous manipulation tasks are constructed based on Adroit. We adopt four tasks *pen, door, relocate, hammer*, where we need to control a 24-DoF shadow hand robot hand to hammer a nail, open a door, twirls a pen or pick up and move a ball. We consider two kinds of dynamics shifts, kinematic shift and morphology shift. Each dynamics shift is equipped with three shift levels: *easy, medium, hard*. We modify the embodied dexterous hand instead of changing the objects like the pen, indicating that the four tasks share the identical robot hand given the dynamics shift type and the shift level. We summarize the detailed modifications to the robot hand below.

### D.3.1   Kinematic shift

The kinematic shift occurs in all finger joints of the index finger and the thumb. The simulated robot hand remains the same as that in the source domain in visual appearance. The visualization results can be found in Figure 13. We provide three types of shift levels (*easy, medium, hard*) for the kinematic shift. For each shift level, we have

Table 8: **Full list of supported task names in AntMaze.**

|  |  |  |
|---|---|---|
| Map layout | small maze | antmaze-small-empty
antmaze-small-centerblock
antmaze-small-lshape
antmaze-small-zshape
antmaze-small-reversel
antmaze-small-reverseu |
|  | medium maze | antmaze-medium-1
antmaze-medium-2
antmaze-medium-3
antmaze-medium-4
antmaze-medium-5
antmaze-medium-6 |
|  | large maze | antmaze-large-1
antmaze-large-2
antmaze-large-3
antmaze-large-4
antmaze-large-5
antmaze-large-6 |

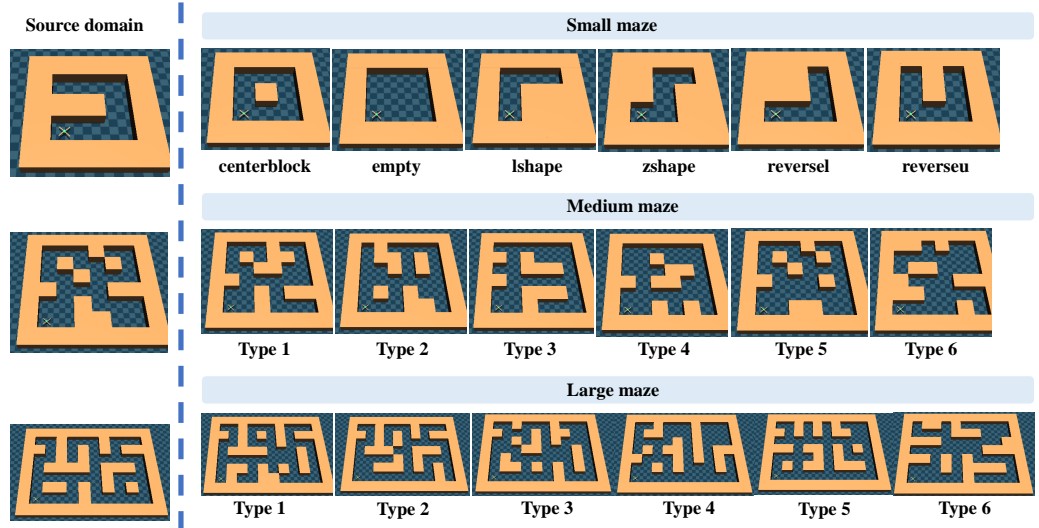

Figure 12: **Visualization of AntMaze tasks.** We consider three different map sizes and each map size contains 6 different map layouts. The robot starts from the bottom left corner of the map and aim at reaching the top left position in the map.

Table 9: **Full list of supported task names in the Adroit domain.**

| kinematic | | morphology | |
|---|---|---|---|
| pen-broken-joint-easy | | pen-shrink-finger-easy | |
| pen-broken-joint-medium | | pen-shrink-finger-medium | |
| pen-broken-joint-hard | | pen-shrink-finger-hard | |
| door-broken-joint-easy | | door-shrink-finger-easy | |
| door-broken-joint-medium | | door-shrink-finger-medium | |
| door-broken-joint-hard | | door-shrink-finger-hard | |
| relocate-broken-joint-easy | | relocate-shrink-finger-easy | |
| relocate-broken-joint-medium | | relocate-shrink-finger-medium | |
| relocate-broken-joint-hard | | relocate-shrink-finger-hard | |
| hammer-broken-joint-easy | | hammer-shrink-finger-easy | |
| hammer-broken-joint-medium | | hammer-shrink-finger-medium | |
| hammer-broken-joint-hard | | hammer-shrink-finger-hard | |

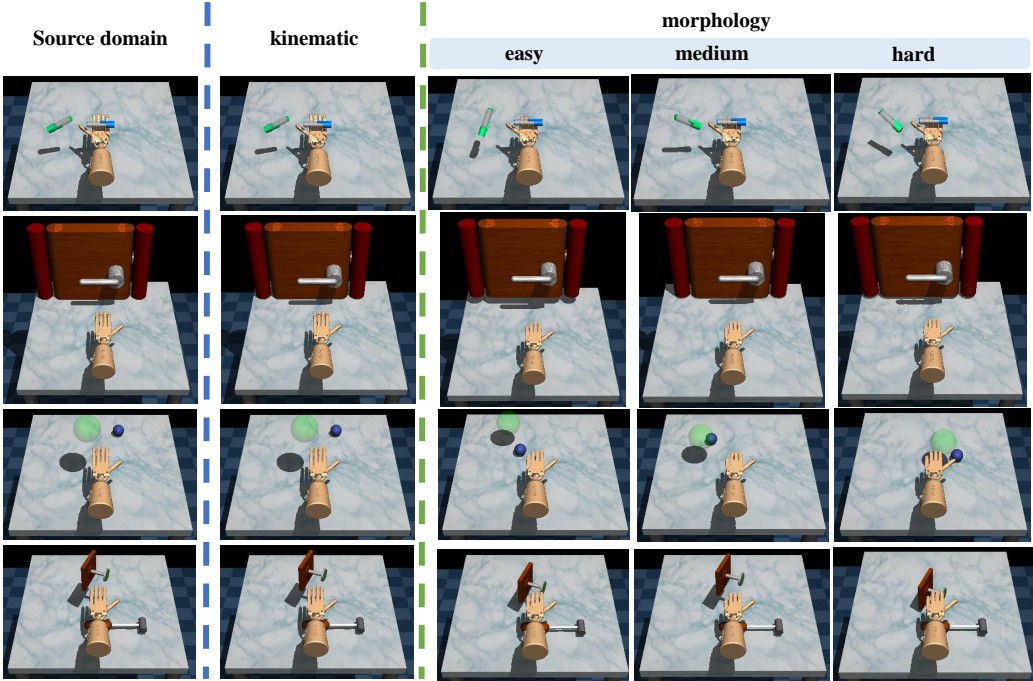

Figure 13: **Visualization of dexterous hand manipulation tasks**. The robot hand under kinematic shifts has the same appearance as the source domain, while the fingers are contracted for morphology shifts.

- *pen / door / relocate / hammer-broken-joint-easy:* the rotation ranges of all finger joints in the index finger and the thumb are modified to be 0.5 times of those in the source domain:

```
# index finger
<joint name="FFJ3" pos="0 0 0" axis="0 1 0" range="-0.218 0.218" user="
    1103" />
<joint name="FFJ2" pos="0 0 0" axis="1 0 0" range="0 0.7855" user="1102"
     />
<joint name="FFJ1" pos="0 0 0" axis="1 0 0" range="0 0.7855" user="1101"
     />
<joint name="FFJ0" pos="0 0 0" axis="1 0 0" range="0 0.7855" user="1100"
     />
# thumb
<joint name="THJ4" pos="0 0 0" axis="0 0 -1" range="-0.5235 0.5235" user
    ="1121" />
<joint name="THJ3" pos="0 0 0" axis="1 0 0" range="0 0.6545" user="1120"
     />
<joint name="THJ2" pos="0 0 0" axis="1 0 0" range="-0.131 0.131" user="
    1119" />
<joint name="THJ1" pos="0 0 0" axis="0 1 0" range="-0.262 0.262" user="
    1118" />
<joint name="THJ0" pos="0 0 0" axis="0 1 0" range="-0.785 0" user="1117"
     />
```

- *pen / door / relocate / hammer-broken-joint-medium:* the rotation ranges of all finger joints in the index finger and the thumb are modified to be 0.25 times of those in the source domain:

```
# index finger
<joint name="FFJ3" pos="0 0 0" axis="0 1 0" range="-0.109 0.109" user="
    1103" />
<joint name="FFJ2" pos="0 0 0" axis="1 0 0" range="0 0.39275" user="1102
    " />
```

```
      <joint name="FFJ1" pos="0 0 0" axis="1 0 0" range="0 0.39275" user="1101
          " />
      <joint name="FFJ0" pos="0 0 0" axis="1 0 0" range="0 0.39275" user="1100
          " />
      # thumb
      <joint name="THJ4" pos="0 0 0" axis="0 0 -1" range="-0.26175 0.26175"
          user="1121" />
      <joint name="THJ3" pos="0 0 0" axis="1 0 0" range="0 0.32725" user="1120
          " />
      <joint name="THJ2" pos="0 0 0" axis="1 0 0" range="-0.0655 0.0655" user=
          "1119" />
      <joint name="THJ1" pos="0 0 0" axis="0 1 0" range="-0.131 0.131" user="
          1118" />
      <joint name="THJ0" pos="0 0 0" axis="0 1 0" range="-0.3925 0" user="1117
          " />
```

- *pen / door / relocate / hammer-broken-joint-hard:* the rotation ranges of all finger joints in the index finger and the thumb are modified to be 0.125 times of those in the source domain:

```
      # index finger
      <joint name="FFJ3" pos="0 0 0" axis="0 1 0" range="-0.0545 0.0545" user=
          "1103" />
      <joint name="FFJ2" pos="0 0 0" axis="1 0 0" range="0 0.196375" user="
          1102" />
      <joint name="FFJ1" pos="0 0 0" axis="1 0 0" range="0 0.196375" user="
          1101" />
      <joint name="FFJ0" pos="0 0 0" axis="1 0 0" range="0 0.196375" user="
          1100" />
      # thumb
      <joint name="THJ4" pos="0 0 0" axis="0 0 -1" range="-0.130875 0.130875"
          user="1121" />
      <joint name="THJ3" pos="0 0 0" axis="1 0 0" range="0 0.163625" user="
          1120" />
      <joint name="THJ2" pos="0 0 0" axis="1 0 0" range="-0.03275 0.03275"
          user="1119" />
      <joint name="THJ1" pos="0 0 0" axis="0 1 0" range="-0.0655 0.0655" user=
          "1118" />
      <joint name="THJ0" pos="0 0 0" axis="0 1 0" range="-0.19625 0" user="
          1117" />
```

### D.3.2  Morphology shift

The morphology shift happens at the proximal, intermediate, and distal phalanges in the index, middle, ring, and little fingers. The thumb is unchanged. We shrink the phalanges sizes of these fingers with three different shift levels (*easy, medium, hard*). Please check the visualization results of dexterous hand under morphology shifts in Figure 13. For each shift level, we have

- *pen / door / relocate / hammer-shrink-finger-easy:* the phalanges sizes are modified to be 0.5 times of those in the source domain:

```
      # index finger
      <inertial pos="0 0 0.0115" quat="0.707095 -0.00400054 0.00400054
          0.707095" mass="0.014" diaginertia="1e-05 1e-05 1e-05" />
      <geom name="C_ffproximal" class="DC_Hand" size="0.01 0.01125" pos="0 0
          0.01125" type="capsule" />
      <site class="D_Touch" name="Tch_ffproximal" size="0.009 0.004 0.006" pos
          ="0 -.007 .011"/>
      <body name="ffmiddle" pos="0 0 0.0225">
      <inertial pos="0 0 0.0055" quat="0.707107 0 0 0.707107" mass="0.012"
          diaginertia="1e-05 1e-05 1e-05" />
      <geom name="C_ffmiddle" class="DC_Hand" size="0.00805 0.00625" pos="0 0
          0.00625" type="capsule" />
      <site class="D_Touch" name="Tch_ffmiddle" size="0.009 0.002 0.0036" pos=
          "0 -.007 .0065"/>
```

```xml
<body name="ffdistal" pos="0 0 0.0125">
<inertial pos="0 0 0.0075" quat="0.7071 -0.00300043 0.00300043 0.7071"
    mass="0.01" diaginertia="1e-05 1e-05 1e-05" />
<geom name="C_ffdistal" class="DC_Hand" size="0.00705 0.006" pos="0 0
    0.006" type="capsule" condim="4" />
<site name="S_fftip" pos="0 0 0.013" group="3" />
<site name="Tch_fftip" class="D_Touch" pos="0 -0.004 0.009" />
# middle finger
<inertial pos="0 0 0.0115" quat="0.707095 -0.00400054 0.00400054
    0.707095" mass="0.014" diaginertia="1e-05 1e-05 1e-05" />
<geom name="C_mfproximal" class="DC_Hand" size="0.01 0.01125" pos="0 0
    0.01125" type="capsule" />
<site class="D_Touch" name="Tch_mfproximal" size="0.009 0.004 0.006" pos
    ="0 -.007 .022"/>
<body name="mfmiddle" pos="0 0 0.0225">
<inertial pos="0 0 0.006" quat="0.707107 0 0 0.707107" mass="0.012"
    diaginertia="1e-05 1e-05 1e-05" />
<geom name="C_mfmiddle" class="DC_Hand" size="0.00805 0.00625" pos="0 0
    0.00625" type="capsule" />
<site class="D_Touch" name="Tch_mfmiddle" size="0.009 0.002 0.0035" pos=
    "0 -.007 .0065"/>
<body name="mfdistal" pos="0 0 0.0125">
<inertial pos="0 0 0.0075" quat="0.7071 -0.00300043 0.00300043 0.7071"
    mass="0.01" diaginertia="1e-05 1e-05 1e-05" />
<geom name="C_mfdistal" class="DC_Hand" size="0.00705 0.006" pos="0 0
    0.006" type="capsule" condim="4" />
<site name="S_mftip" pos="0 0 0.013" group="3" />
<site name="Tch_mftip" class="D_Touch" pos="0 -0.004 0.009" />
# ring finger
<inertial pos="0 0 0.0115" quat="0.707095 -0.00400054 0.00400054
    0.707095" mass="0.014" diaginertia="1e-05 1e-05 1e-05" />
<geom name="C_rfproximal" class="DC_Hand" size="0.01 0.01125" pos="0 0
    0.01125" type="capsule" />
<site class="D_Touch" name="Tch_rfproximal" size="0.009 0.004 0.006" pos
    ="0 -.007 .011"/>
<body name="rfmiddle" pos="0 0 0.0225">
<inertial pos="0 0 0.006" quat="0.707107 0 0 0.707107" mass="0.012"
    diaginertia="1e-05 1e-05 1e-05" />
<geom name="C_rfmiddle" class="DC_Hand" size="0.00805 0.00625" pos="0 0
    0.00625" type="capsule" />
<site class="D_Touch" name="Tch_rfmiddle" size="0.009 0.002 0.0035" pos=
    "0 -.007 .0065"/>
<body name="rfdistal" pos="0 0 0.0125">
<inertial pos="0 0 0.0075" quat="0.7071 -0.00300043 0.00300043 0.7071"
    mass="0.01" diaginertia="1e-05 1e-05 1e-05" />
<geom name="V_rfdistal" class="D_Vizual" pos="0 0 0.0005" mesh="F1" />
<geom name="C_rfdistal" class="DC_Hand" size="0.00705 0.006" pos="0 0
    0.006" type="capsule" condim="4" />
<site name="S_rftip" pos="0 0 0.013" group="3" />
<site name="Tch_rftip" class="D_Touch" pos="0 -0.004 0.009" />
# little finger
<body name="lfknuckle" pos="-0.017 0 0.022">
<inertial pos="0 0 0.0115" quat="0.707095 -0.00400054 0.00400054
    0.707095" mass="0.014" diaginertia="1e-05 1e-05 1e-05" />
<geom name="C_lfproximal" class="DC_Hand" size="0.01 0.01125" pos="0 0
    0.01125" type="capsule" />
<site class="D_Touch" name="Tch_lfproximal" size="0.009 0.004 0.006" pos
    ="0 -.007 .011"/>
<body name="lfmiddle" pos="0 0 0.0225">
<inertial pos="0 0 0.006" quat="0.707107 0 0 0.707107" mass="0.012"
    diaginertia="1e-05 1e-05 1e-05" />
<geom name="C_lfmiddle" class="DC_Hand" size="0.00805 0.00625" pos="0 0
    0.00625" type="capsule" />
<site class="D_Touch" name="Tch_lfmiddle" size="0.009 0.002 0.0035" pos=
    "0 -.007 .0065"/>
```

```
<body name="lfdistal" pos="0 0 0.0125">
<inertial pos="0 0 0.0075" quat="0.7071 -0.00300043 0.00300043 0.7071"
    mass="0.01" diaginertia="1e-05 1e-05 1e-05" />
<geom name="V_lfdistal" class="D_Vizual" pos="0 0 0.0005" mesh="F1" />
<geom name="C_lfdistal" class="DC_Hand" size="0.00705 0.006" pos="0 0
    0.006" type="capsule" condim="4" />
<site name="S_lftip" pos="0 0 0.013" group="3" />
<site name="Tch_lftip" class="D_Touch" pos="0 -0.004 0.009" />
```

- *pen / door / relocate / hammer-shrink-finger-medium:* the phalanges sizes are modified to be 0.25 times of those in the source domain:

```
# index finger
<inertial pos="0 0 0.00575" quat="0.707095 -0.00400054 0.00400054
    0.707095" mass="0.014" diaginertia="1e-05 1e-05 1e-05" />
<geom name="C_ffproximal" class="DC_Hand" size="0.01 0.005625" pos="0 0
    0.005625" type="capsule" />
<site class="D_Touch" name="Tch_ffproximal" size="0.009 0.004 0.003" pos
    ="0 -.007 .0055"/>
<body name="ffmiddle" pos="0 0 0.01125">
<inertial pos="0 0 0.00275" quat="0.707107 0 0 0.707107" mass="0.012"
    diaginertia="1e-05 1e-05 1e-05" />
<geom name="C_ffmiddle" class="DC_Hand" size="0.00805 0.003125" pos="0 0
     0.003125" type="capsule" />
<site class="D_Touch" name="Tch_ffmiddle" size="0.009 0.002 0.0018" pos=
    "0 -.007 .00325"/>
<body name="ffdistal" pos="0 0 0.00625">
<inertial pos="0 0 0.00375" quat="0.7071 -0.00300043 0.00300043 0.7071"
    mass="0.01" diaginertia="1e-05 1e-05 1e-05" />
<geom name="C_ffdistal" class="DC_Hand" size="0.00705 0.003" pos="0 0
    0.003" type="capsule" condim="4" />
<site name="S_fftip" pos="0 0 0.0065" group="3" />
<site name="Tch_fftip" class="D_Touch" pos="0 -0.004 0.0045" />
# middle finger
<inertial pos="0 0 0.00575" quat="0.707095 -0.00400054 0.00400054
    0.707095" mass="0.014" diaginertia="1e-05 1e-05 1e-05" />
<geom name="C_mfproximal" class="DC_Hand" size="0.01 0.005625" pos="0 0
    0.005625" type="capsule" />
<site class="D_Touch" name="Tch_mfproximal" size="0.009 0.004 0.003" pos
    ="0 -.007 .022"/>
<body name="mfmiddle" pos="0 0 0.01125">
<inertial pos="0 0 0.003" quat="0.707107 0 0 0.707107" mass="0.012"
    diaginertia="1e-05 1e-05 1e-05" />
<geom name="C_mfmiddle" class="DC_Hand" size="0.00805 0.003125" pos="0 0
     0.003125" type="capsule" />
<site class="D_Touch" name="Tch_mfmiddle" size="0.009 0.002 0.00175" pos
    ="0 -.007 .00325"/>
<body name="mfdistal" pos="0 0 0.00625">
<inertial pos="0 0 0.00375" quat="0.7071 -0.00300043 0.00300043 0.7071"
    mass="0.01" diaginertia="1e-05 1e-05 1e-05" />
<geom name="C_mfdistal" class="DC_Hand" size="0.00705 0.003" pos="0 0
    0.003" type="capsule" condim="4" />
<site name="S_mftip" pos="0 0 0.0065" group="3" />
<site name="Tch_mftip" class="D_Touch" pos="0 -0.004 0.0045" />
# ring finger
<inertial pos="0 0 0.00575" quat="0.707095 -0.00400054 0.00400054
    0.707095" mass="0.014" diaginertia="1e-05 1e-05 1e-05" />
<geom name="C_rfproximal" class="DC_Hand" size="0.01 0.005625" pos="0 0
    0.005625" type="capsule" />
<site class="D_Touch" name="Tch_rfproximal" size="0.009 0.004 0.003" pos
    ="0 -.007 .0055"/>
<body name="rfmiddle" pos="0 0 0.01125">
<inertial pos="0 0 0.003" quat="0.707107 0 0 0.707107" mass="0.012"
    diaginertia="1e-05 1e-05 1e-05" />
```

```
<geom name="C_rfmiddle" class="DC_Hand" size="0.00805 0.003125" pos="0 0
    0.003125" type="capsule" />
<site class="D_Touch" name="Tch_rfmiddle" size="0.009 0.002 0.00175" pos
    ="0 -.007 .00325"/>
<body name="rfdistal" pos="0 0 0.00625">
<inertial pos="0 0 0.00375" quat="0.7071 -0.00300043 0.00300043 0.7071"
    mass="0.01" diaginertia="1e-05 1e-05 1e-05" />
<geom name="V_rfdistal" class="D_Vizual" pos="0 0 0.00025" mesh="F1" />
<geom name="C_rfdistal" class="DC_Hand" size="0.00705 0.003" pos="0 0
    0.003" type="capsule" condim="4" />
<site name="S_rftip" pos="0 0 0.0065" group="3" />
<site name="Tch_rftip" class="D_Touch" pos="0 -0.004 0.0045" />
# little finger
<body name="lfknuckle" pos="-0.017 0 0.011">
<inertial pos="0 0 0.00575" quat="0.707095 -0.00400054 0.00400054
    0.707095" mass="0.014" diaginertia="1e-05 1e-05 1e-05" />
<geom name="C_lfproximal" class="DC_Hand" size="0.01 0.005625" pos="0 0
    0.005625" type="capsule" />
<site class="D_Touch" name="Tch_lfproximal" size="0.009 0.004 0.003" pos
    ="0 -.007 .0055"/>
<body name="lfmiddle" pos="0 0 0.01125">
<inertial pos="0 0 0.003" quat="0.707107 0 0 0.707107" mass="0.012"
    diaginertia="1e-05 1e-05 1e-05" />
<geom name="C_lfmiddle" class="DC_Hand" size="0.00805 0.003125" pos="0 0
    0.003125" type="capsule" />
<site class="D_Touch" name="Tch_lfmiddle" size="0.009 0.002 0.00175" pos
    ="0 -.007 .00325"/>
<body name="lfdistal" pos="0 0 0.00625">
<inertial pos="0 0 0.00375" quat="0.7071 -0.00300043 0.00300043 0.7071"
    mass="0.01" diaginertia="1e-05 1e-05 1e-05" />
<geom name="V_lfdistal" class="D_Vizual" pos="0 0 0.00025" mesh="F1" />
<geom name="C_lfdistal" class="DC_Hand" size="0.00705 0.003" pos="0 0
    0.003" type="capsule" condim="4" />
<site name="S_lftip" pos="0 0 0.0065" group="3" />
<site name="Tch_lftip" class="D_Touch" pos="0 -0.004 0.0045" />
```

- *pen / door / relocate / hammer-shrink-finger-hard:* the phalanges sizes are modified to be 0.125 times of those in the source domain:

```
# index finger
<inertial pos="0 0 0.002875" quat="0.707095 -0.00400054 0.00400054
    0.707095" mass="0.014" diaginertia="1e-05 1e-05 1e-05" />
<geom name="C_ffproximal" class="DC_Hand" size="0.01 0.0028125" pos="0 0
    0.0028125" type="capsule" />
<site class="D_Touch" name="Tch_ffproximal" size="0.009 0.004 0.0015"
    pos="0 -.007 .00275"/>
<body name="ffmiddle" pos="0 0 0.005625">
<inertial pos="0 0 0.001375" quat="0.707107 0 0 0.707107" mass="0.012"
    diaginertia="1e-05 1e-05 1e-05" />
<geom name="C_ffmiddle" class="DC_Hand" size="0.00805 0.0015625" pos="0
    0 0.0015625" type="capsule" />
<site class="D_Touch" name="Tch_ffmiddle" size="0.009 0.002 0.0009" pos=
    "0 -.007 .001625"/>
<body name="ffdistal" pos="0 0 0.003125">
<inertial pos="0 0 0.001875" quat="0.7071 -0.00300043 0.00300043 0.7071"
    mass="0.01" diaginertia="1e-05 1e-05 1e-05" />
<geom name="C_ffdistal" class="DC_Hand" size="0.00705 0.0015" pos="0 0
    0.0015" type="capsule" condim="4" />
<site name="S_fftip" pos="0 0 0.00325" group="3" />
<site name="Tch_fftip" class="D_Touch" pos="0 -0.004 0.00225" />
# middle finger
<inertial pos="0 0 0.002875" quat="0.707095 -0.00400054 0.00400054
    0.707095" mass="0.014" diaginertia="1e-05 1e-05 1e-05" />
<geom name="C_mfproximal" class="DC_Hand" size="0.01 0.0028125" pos="0 0
    0.0028125" type="capsule" />
```

```xml
<site class="D_Touch" name="Tch_mfproximal" size="0.009 0.004 0.0015"
    pos="0 -.007 .022"/>
<body name="mfmiddle" pos="0 0 0.005625">
<inertial pos="0 0 0.0015" quat="0.707107 0 0 0.707107" mass="0.012"
    diaginertia="1e-05 1e-05 1e-05" />
<geom name="C_mfmiddle" class="DC_Hand" size="0.00805 0.0015625" pos="0
    0 0.0015625" type="capsule" />
<site class="D_Touch" name="Tch_mfmiddle" size="0.009 0.002 0.000875"
    pos="0 -.007 .001625"/>
<body name="mfdistal" pos="0 0 0.003125">
<inertial pos="0 0 0.001875" quat="0.7071 -0.00300043 0.00300043 0.7071"
    mass="0.01" diaginertia="1e-05 1e-05 1e-05" />
<geom name="C_mfdistal" class="DC_Hand" size="0.00705 0.0015" pos="0 0
    0.0015" type="capsule" condim="4" />
<site name="S_mftip" pos="0 0 0.00325" group="3" />
<site name="Tch_mftip" class="D_Touch" pos="0 -0.004 0.00225" />
# ring finger
<inertial pos="0 0 0.002875" quat="0.707095 -0.00400054 0.00400054
    0.707095" mass="0.014" diaginertia="1e-05 1e-05 1e-05" />
<geom name="C_rfproximal" class="DC_Hand" size="0.01 0.0028125" pos="0 0
    0.0028125" type="capsule" />
<site class="D_Touch" name="Tch_rfproximal" size="0.009 0.004 0.0015"
    pos="0 -.007 .00275"/>
<body name="rfmiddle" pos="0 0 0.005625">
<inertial pos="0 0 0.0015" quat="0.707107 0 0 0.707107" mass="0.012"
    diaginertia="1e-05 1e-05 1e-05" />
<geom name="C_rfmiddle" class="DC_Hand" size="0.00805 0.0015625" pos="0
    0 0.0015625" type="capsule" />
<site class="D_Touch" name="Tch_rfmiddle" size="0.009 0.002 0.000875"
    pos="0 -.007 .001625"/>
<body name="rfdistal" pos="0 0 0.003125">
<inertial pos="0 0 0.001875" quat="0.7071 -0.00300043 0.00300043 0.7071"
    mass="0.01" diaginertia="1e-05 1e-05 1e-05" />
<geom name="V_rfdistal" class="D_Vizual" pos="0 0 0.000125" mesh="F1" />
<geom name="C_rfdistal" class="DC_Hand" size="0.00705 0.0015" pos="0 0
    0.0015" type="capsule" condim="4" />
<site name="S_rftip" pos="0 0 0.00325" group="3" />
<site name="Tch_rftip" class="D_Touch" pos="0 -0.004 0.00225" />
# little finger
<body name="lfknuckle" pos="-0.017 0 0.0055">
<inertial pos="0 0 0.002875" quat="0.707095 -0.00400054 0.00400054
    0.707095" mass="0.014" diaginertia="1e-05 1e-05 1e-05" />
<geom name="C_lfproximal" class="DC_Hand" size="0.01 0.0028125" pos="0 0
    0.0028125" type="capsule" />
<site class="D_Touch" name="Tch_lfproximal" size="0.009 0.004 0.0015"
    pos="0 -.007 .00275"/>
<body name="lfmiddle" pos="0 0 0.005625">
<inertial pos="0 0 0.0015" quat="0.707107 0 0 0.707107" mass="0.012"
    diaginertia="1e-05 1e-05 1e-05" />
<geom name="C_lfmiddle" class="DC_Hand" size="0.00805 0.0015625" pos="0
    0 0.0015625" type="capsule" />
<site class="D_Touch" name="Tch_lfmiddle" size="0.009 0.002 0.000875"
    pos="0 -.007 .001625"/>
<body name="lfdistal" pos="0 0 0.03125">
<inertial pos="0 0 0.001875" quat="0.7071 -0.00300043 0.00300043 0.7071"
    mass="0.01" diaginertia="1e-05 1e-05 1e-05" />
<geom name="V_lfdistal" class="D_Vizual" pos="0 0 0.000125" mesh="F1" />
<geom name="C_lfdistal" class="DC_Hand" size="0.00705 0.0015" pos="0 0
    0.0015" type="capsule" condim="4" />
<site name="S_lftip" pos="0 0 0.00325" group="3" />
<site name="Tch_lftip" class="D_Touch" pos="0 -0.004 0.00225" />
```

# E  Additional Results

In this section, we provide additional experimental results that were omitted from the main text due to space constraints. These include experiments conducted under the **Online-Online** setting, performance comparisons across varying dataset qualities on locomotion tasks in the **Offline-Online**, **Online-Offline**, and **Offline-Offline** settings, and some results on the AntMaze and Adroit domain.

## E.1  Wider Results in the Online-Online Setting

We present a broader range of empirical results for the implemented algorithms in the context of both the online source domain and the online target domain in Figure 14. It is evident that different algorithms excel in various types of dynamic shift tasks. For instance, DARC struggles with the *ant-morph-alllegs-medium* task but delivers impressive performance on dexterous manipulation tasks. We observe that **Obs 2** and **Obs 4** still hold, i.e., PAR fails to achieve meaningful performance in dexterous manipulation tasks but achieves good results in locomotion tasks, while DARC and SAC_IW rank among the best algorithms for dexterous manipulation tasks. We reemphasize that the experimental results demonstrate that no single algorithm can master all types of dynamic shifts across all domains, thereby validating **Obs 1**.

## E.2  Wider Results in the Offline-Online Setting and the Online-Offline Setting

**Offline-Online setting.** We follow the same experimental setup as in the main text (interacting with the target domain for only 0.1M steps) and present a wider range of experimental results in Figure 15. Here, we find that our observations (**Obs 6**, **Obs 7**, **Obs 8**) from the main text still hold. Firstly, even when provided with expert source domain datasets, existing methods do not necessarily achieve good performance in the target domain. In fact, on many tasks, the performance with expert source domain datasets is even worse than with medium-replay or medium datasets. Furthermore, methods that treat both domains as a single mixed domain can outperform off-dynamics RL algorithms, particularly RLPD, which exhibits superior performance on numerous tasks. Moreover, on tasks like *walker2d-kinematic-footjnt-hard*, methods that involve conservative value estimation (e.g., MCQ_SAC) outperform those that utilize the BC term when given expert source domain datasets.

Furthermore, we carry out experiments on dexterous hand manipulation tasks. D4RL provides three types of source domain datasets for these tasks: *human, cloned, expert*. For our experiments, we use the expert-level source domain datasets. The results are shown in Figure 16, where we observe that existing off-dynamics RL algorithms typically struggle with high-dimensional, sparse reward tasks like Adroit, even when expert source domain datasets are available, as highlighted in **Obs 6**. RLPD is the only method that consistently achieves good performance across numerous tasks. This suggests that complex manipulation tasks pose a significant challenge for existing off-dynamics RL methods, and specific designs may be required when training on the Adroit domain, such as building off-dynamics RL algorithms on top of RLPD instead of SAC. These results indicate that there is still substantial work to be done towards realizing general dynamics-aware RL algorithms.

**Online-Offline setting.** We train algorithms in this category for 500K gradient steps, with 500K interactions with the source domain. We present the wider empirical results given the offline target domain and the online source domain in Figure 17, where it is evident that **Obs 9** still holds, i.e., it is difficult to achieve a good performance given the random target domain datasets, and policy performance may not improve even when expert target domain datasets are provided. **Obs 10** also holds as PAR_BC can achieve quite good performance in *walker2d-gravity-5.0*, and numerous algorithms can acquire good policies in *ant-kinematic-anklejnt-medium* and *ant-friction-0.5*.

Moreover, we include experiments on AntMaze tasks to investigate whether existing methods can achieve efficient policy adaptation using limited offline data from the target domain and sufficient online interactions with the source domain. We use the mixed target domain datasets for the small, medium, and large size mazes, respectively. The results are depicted in Figure 18. It appears that most existing methods struggle with AntMaze tasks, and adapting policies across varied landscapes or obstacles remains a challenge. This observation aligns with **Obs 3** in the main text.

We believe the additional evidence further strengthens the observations presented in the main text.

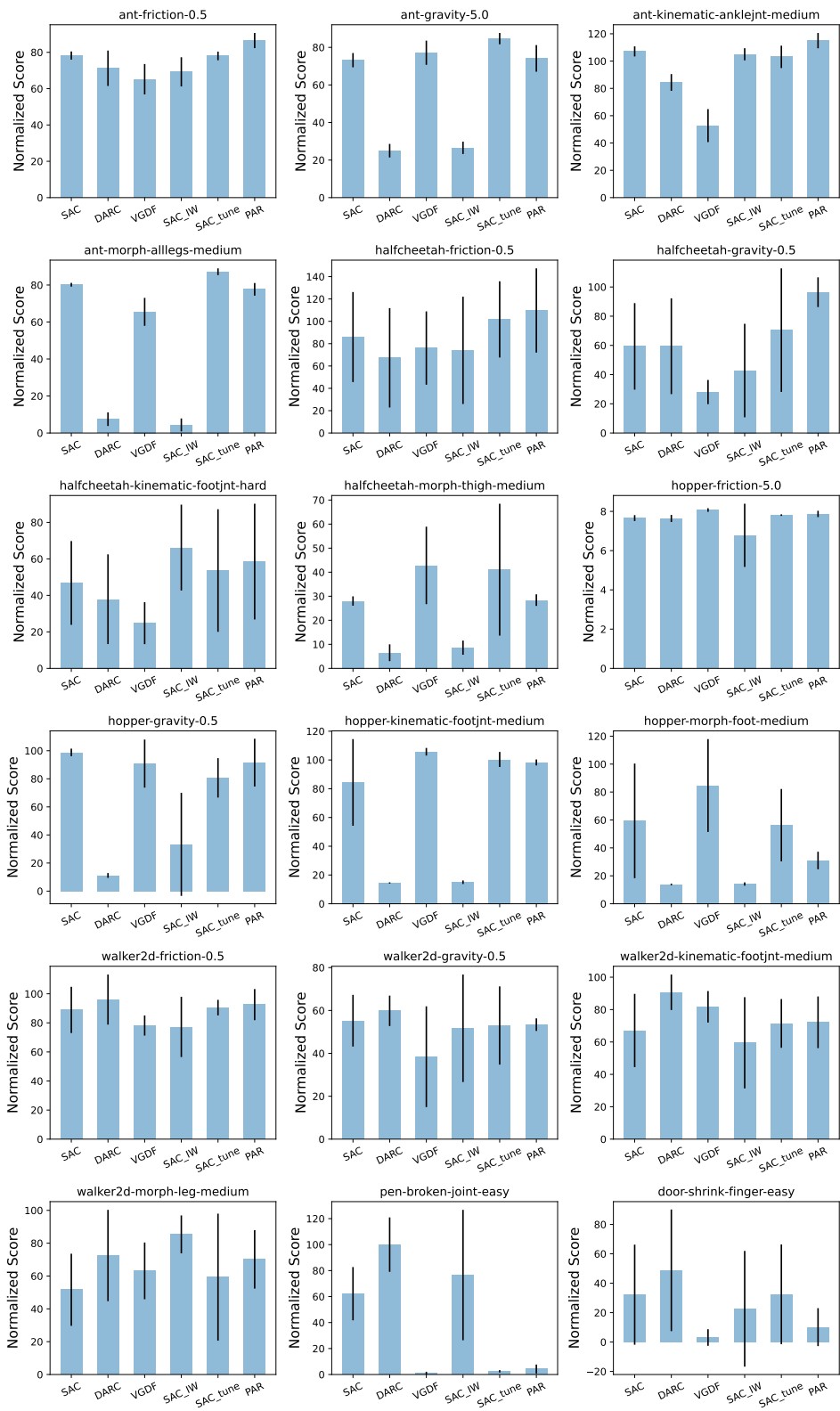

Figure 14: **Wider normalized score comparison in the Online-Online setting.** We select 4 kinds of dynamics shift tasks from each single locomotion task, along with two dexterous hand manipulation tasks. We do not include results on AntMaze tasks because all algorithms fail on those tasks.

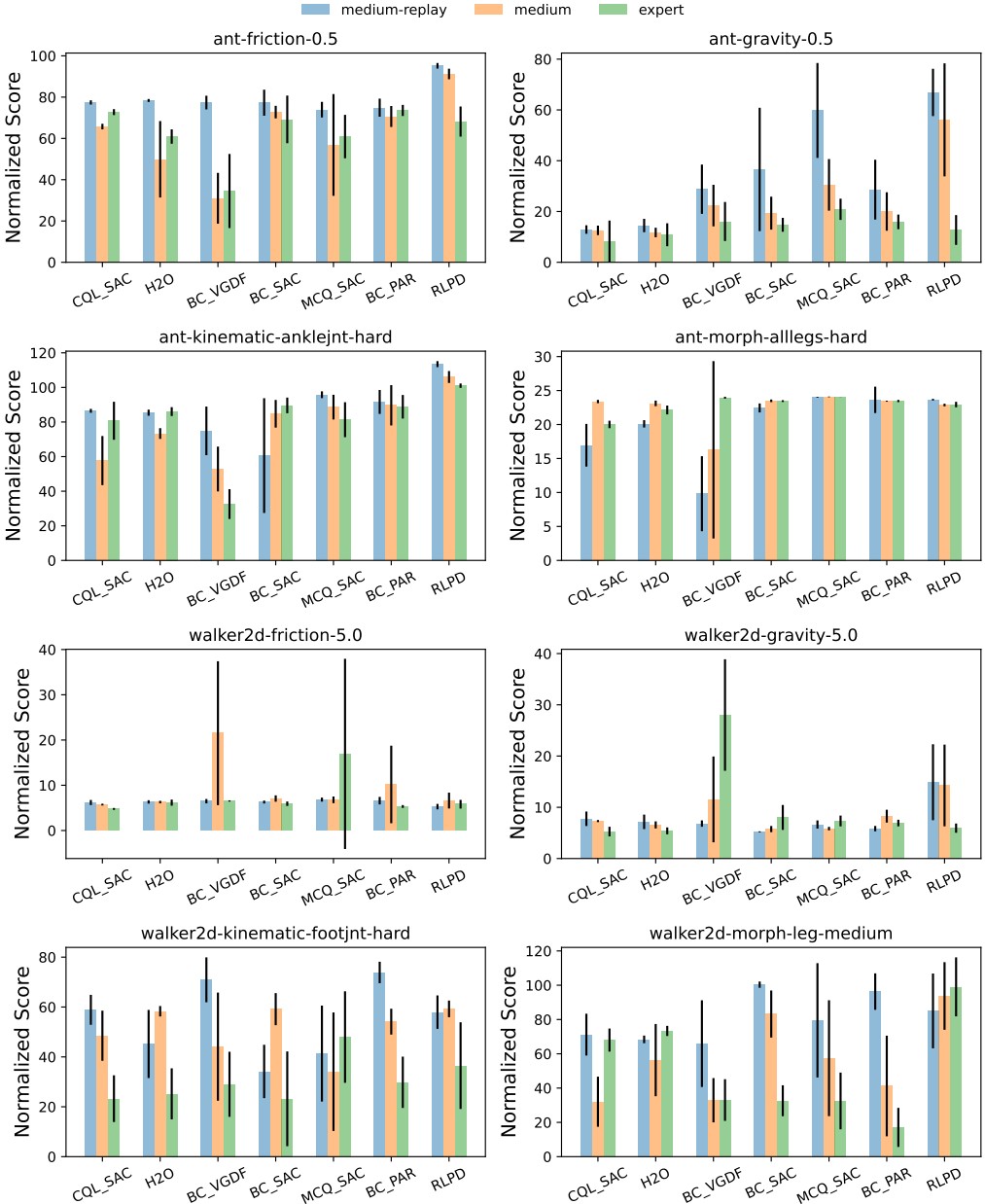

Figure 15: **Wider normalized score comparison under varied source domain dataset qualities under Offline-Online setting.** We choose *ant* and *walker2d* tasks with 4 varied dynamics shift tasks. The source domain datasets can be *medium-replay*, *medium*, or *expert*, which are directly taken from the D4RL benchmark. We report the average performance and the standard deviations at the final gradient step.

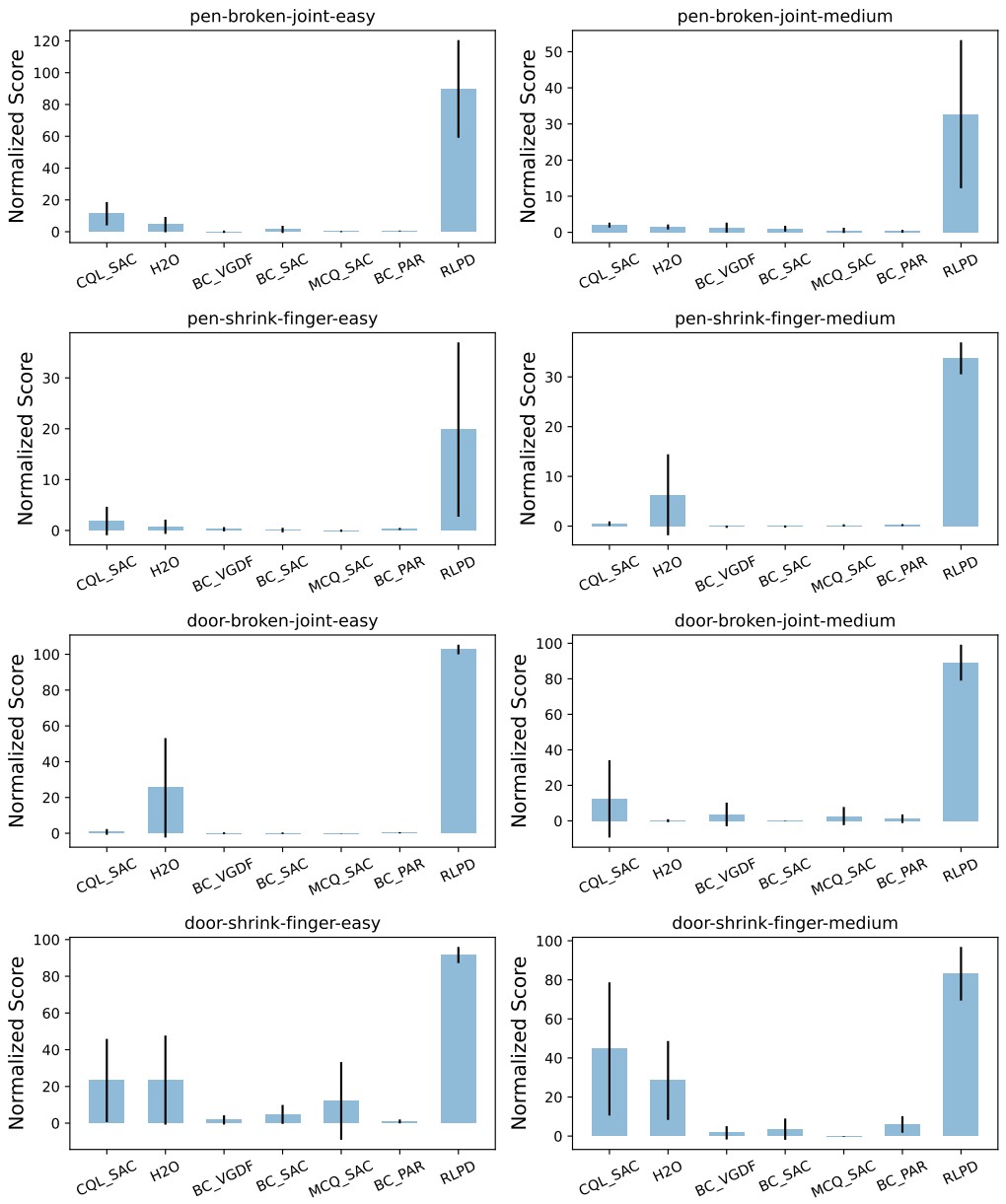

Figure 16: **Empirical results on selected Adroit tasks under the Offline-Online setting.** We adopt the expert-level source domain datasets for experiments.

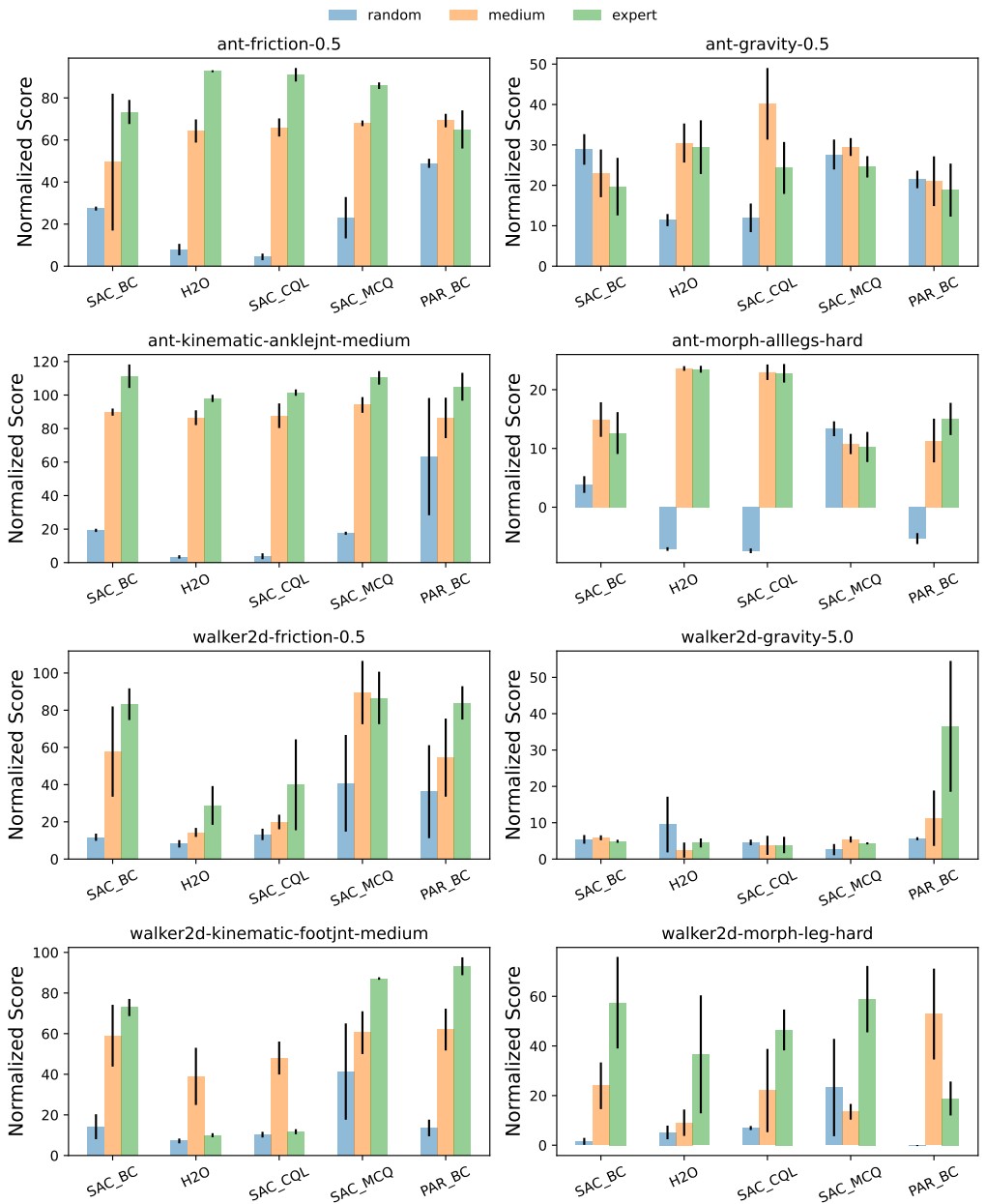

Figure 17: **Normalized score comparison of baseline methods on wider environments.** We choose 8 tasks from the *ant* and *walker2d* robots with varied dynamics shifts. We consider the following qualities of the target domain datasets: *random, medium, expert*. The final mean performance as well as the standard deviations are reported.

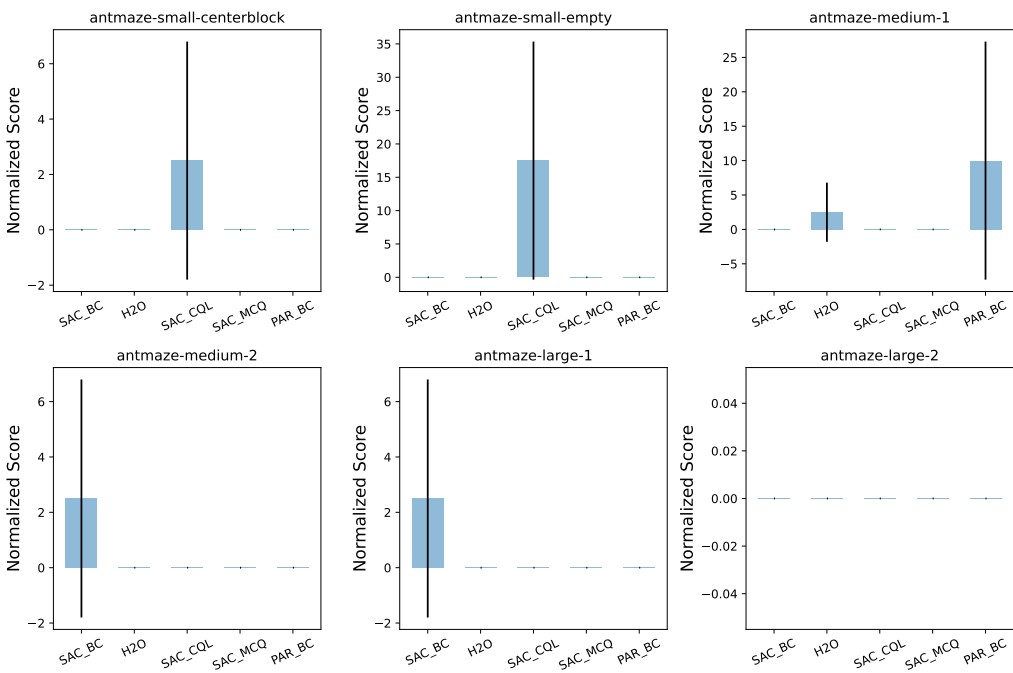

Figure 18: **Normalized score comparison on some AntMaze tasks under the Online-Offline setting.** It can be found that most of the methods fail on these tasks (with a normalized score 0).

### E.3 Empirical Results in the Offline-Offline Setting

In the main text, we primarily focus on the **Online-Online**, **Offline-Online**, and **Online-Offline** settings. Here, we provide some experimental results in the **Offline-Offline** setting. We train the baseline methods for 500K gradient steps, without any interactions with either the source or the target domain. We use IQL as the base offline RL algorithm for DARA. We select four tasks (*ant-friction-0.5, ant-friction-5.0, walker2d-kinematic-footjnt-medium, walker2d-kinematic-footjnt-hard*) and examine how the baselines perform under a pure offline setting, and how the performance changes if the shift level changes. For source domain offline datasets, we consider *medium-replay, medium, expert* datasets from D4RL, while for the target domain datasets, we consider *random, medium, expert* datasets. We first fix the dataset quality in the target domain to be medium, and examine how the baselines perform under different qualities of source domain datasets. Then, we fix the source domain dataset quality to be medium, and sweep the target domain offline datasets across *random, medium, expert* qualities. The results are presented in Figure 19 and Figure 20, respectively.

Based on Figure 19, we observe that the agent's performance may not improve even when expert source domain datasets or target domain datasets are provided (e.g., the performance of IQL and BOSA drop with expert-level datasets in the *ant-friction-5.0* task), and the agent's performance is unsatisfactory if only random target domain datasets are available. This further validates **Obs 6** and **Obs 9**. We also observe that IQL and TD3_BC can beat BOSA and DARA on some tasks, which validates **Obs 7**. Note that all of our experiments generally adopt one set of hyperparameters for implemented methods, which should explain in part why some approaches exhibit poor performance on some tasks.

## F Compute Infrastructure

Our experiments are conducted via the following dependencies:

- gym == 0.18.3
- torch == 1.11.0
- dm-control == 1.0.8

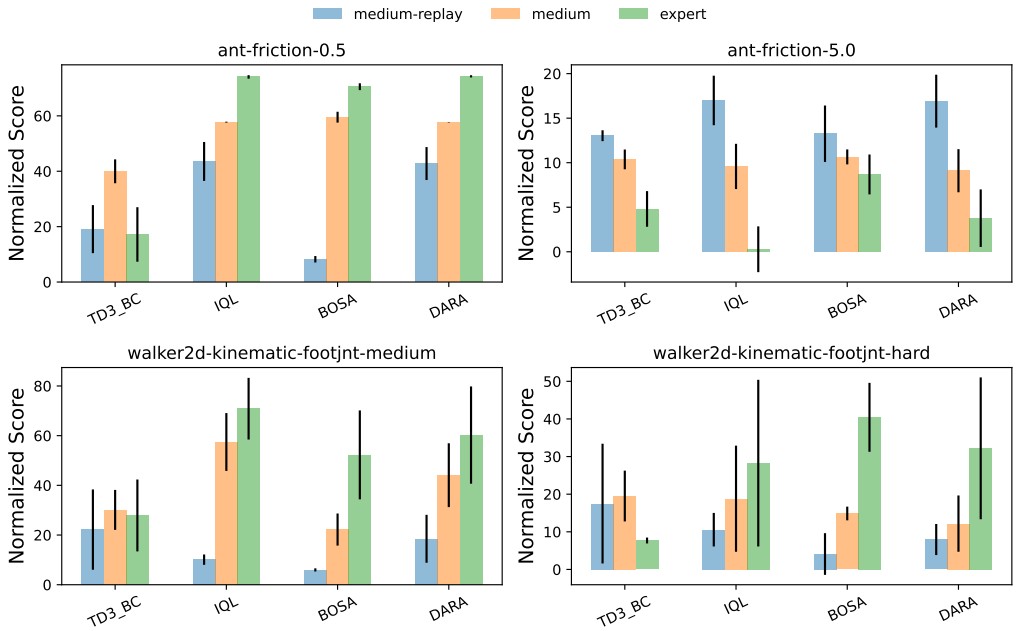

Figure 19: **Normalized score comparison of baselines given medium-level target domain datasets and varied qualities of source domain datasets.** We report the final average performance and the corresponding standard deviation.

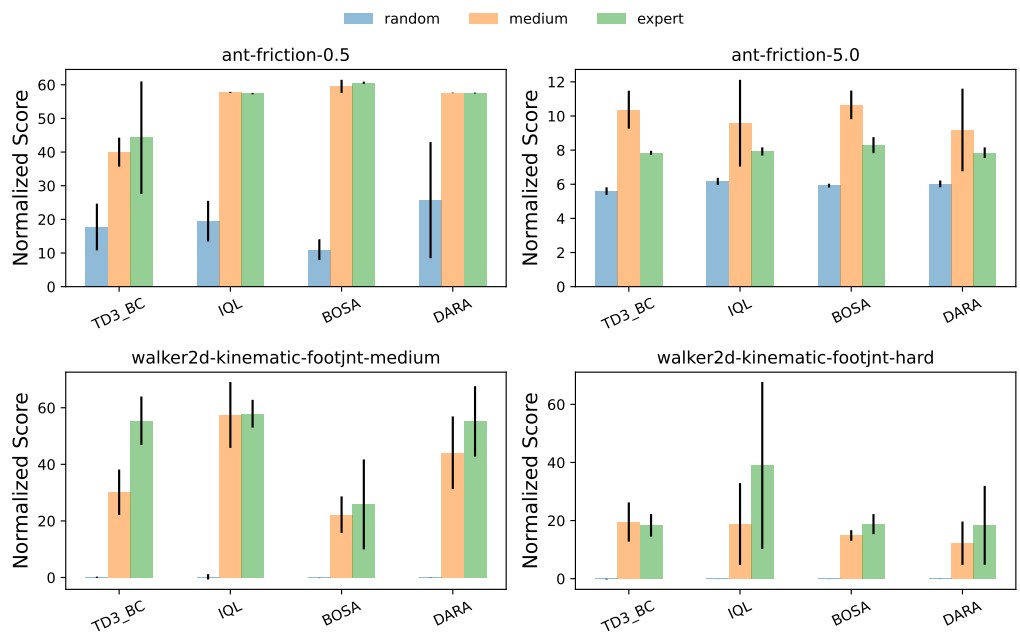

Figure 20: **Normalized score comparison of baselines given medium-level source domain datasets and varied qualities of target domain datasets.** We report the final average performance and the corresponding standard deviation.

- numpy == 1.23.5
- d4rl == 1.1
- mujoco-py == 2.1.2.14
- python == 3.8.13

In Table 10, we list the compute infrastructure that we use to run all of the algorithms.

Table 10: **Compute infrastructure.**

| CPU | GPU | Memory |
|---|---|---|
| AMD EPYC 7452 | RTX3090×20 | 720GB |

We adopt the Gym environments and mujoco-py under the MIT License. For the D4RL library (including the Antmaze domain and the Adroit domain, and offline datasets), all datasets are licensed under the Creative Commons Attribution 4.0 License (CC BY), and code is licensed under the Apache 2.0 License.

## G   Broader Impacts

In this work, we propose the first benchmark for off-dynamics RL. The goal of our benchmark is to facilitate the development of more general and advanced dynamics-aware RL algorithms that can quickly adapt to structurally similar environments or tasks with varied transition dynamics. Our benchmark primarily focuses on policy adaptation within a single task, i.e., only the transition dynamics vary, and the state space and action space remain unchanged. We also provide only one single source domain and one single target domain. However, it should not be difficult to modify our benchmark code to support policy adaptation from *multiple source domains* to a single target domain. It is also possible to revise the code to evaluate the *zero-shot* transfer capability of the agent. Furthermore, one could modify our benchmark to support cross-domain policy adaptation, where the source and target domains can have varied state spaces and action spaces.

Regarding potential social impacts, we believe that our benchmark is beneficial for advancing dynamics-aware RL algorithms, which can be helpful for humans since robots can quickly adapt to new environments or tasks and assist humans in completing more complex and dangerous work. On the other hand, as robots become more capable and adaptable, they may replace human labor in various industries, leading to job losses and increased unemployment. This can exacerbate economic inequality. Meanwhile, it may become more difficult for humans to understand how off-dynamics RL algorithms work and make decisions. This lack of transparency can lead to a loss of trust in robotic systems and hinder accountability. Finally, the developed off-dynamics RL algorithms can potentially be used in military applications, including wars. These algorithms, which account for the dynamics of the environment and robotic systems, could be employed to improve the performance of autonomous weapons, drones, and other military robots.

