# OpenReview forum: "ODRL: A Benchmark for Off-Dynamics Reinforcement Learning"
_NeurIPS.cc/2024/Datasets_and_Benchmarks_Track — NeurIPS 2024 Track Datasets and Benchmarks Poster_

### Official Review · Reviewer_5MsE · 2024-06-14
**Interesting benchmark with room for improvements**

**Rating:** 6
**Confidence:** 4
**Correctness:** See the review above.

**Review:**

Strengths:
- The benchmark includes a diverse range of tasks and experimental settings to evaluate off-dynamics algorithms.
- The benchmark also includes many algorithm implementations, which facilitates building upon the state-of-the-art.

Weaknesses:
- To run the benchmark, it is necessary to use a Python script (https://github.com/OffDynamicsRL/off-dynamics-rl/blob/main/train.py) with 438 lines of code. Moreover, it is not possible to employ an implementation of an RL algorithm and directly test it in the benchmark, as it seems it is necessary to have the RL algorithm implementation compliant with the 438-line train script. I suggest having the benchmark separate from the algorithm implementations to allow people to use the benchmark with other RL algorithm libraries.
- The problem of Off-Dynamics RL is strongly related to other RL problem formulations, which were not discussed (see below). Moreover, there are other benchmarks that tackle this problem, although they have different features.
- Given that the experiments were performed with only 5 random seeds, and the standard deviations are very high, it is not possible to draw significant conclusions (e.g., see Figure 4).

Questions:

1) “Despite the success, existing papers often conduct experiments within self-proposed environments, which is unhealthy for the advances of this field because it fails to truly reveal the merits of the proposed method.”
I agree with the motivation of having a centralized benchmark so that the results across different papers are comparable. However, I do not fully agree with this sentence since the environments introduced in the paper are also “self-proposed”. Instead, the authors should motivate why someone would use your environments instead of the environments proposed in another work.

2) “P(s′|s, a) : S×A×S → [0, 1]”. This notation only applies to discrete domains.

3) The benchmark includes Ant, Hopper, HalfCheetah, and Walker2d robots. Why did you not include the Humanoid robot, which is more complex? The others are now relatively easy for state-of-the-art single-task methods.

4) “Obs 2. PAR achieves the best performance on locomotion tasks but fails on other domains.” It achieves the best performance on all tasks except dexterous manipulation compared to the evaluated methods.

5) “Obs 3. AntMaze tasks are extremely challenging and no algorithm can achieve meaningful returns, indicating that adapting policies across barriers is hard for state-based methods.”
Are the current methods insufficient, or is the task actually not possible to solve with the given budget for the target domain? For instance, if there are no transitions in regions where walls were introduced, it is possible to learn a policy even assuming a perfect algorithm.

6) “The lack of success with PAR in dexterous manipulation tasks can be credited to the poor reward penalty terms on the source domain data.”
This observation is unclear. Please elaborate on the relationship between reward penalty terms in the source domain and the lack of success in dexterous manipulation tasks.

7) Regarding the results in Fig. 6 in ant-friction-5.0: What explains, for instance, the performance of DARC decreasing so much? The authors should explain how the algorithms transfer to the target domain. Do they reuse the replay buffer from the source domain while training for the target domain, for instance? This would hurt performance if the transitions are too different (negative transfer).

**Strengths:**

See the review above.

**Additional Feedback:**

See the review above.

**Clarity:**

The paper is, in general, well written, with some points that require clarification (see review above).

Minor: The green color used in the citations is too bright, and it is difficult to read it.

**Documentation:**

Although the README of the GitHub repository is detailed, there is no website with documentation for the benchmark. This would be very valuable for the library users.

**Ethics:**

There are no ethical concerns that warrant further discussion in this submission.

**Limitations:**

The authors properly discussed a few limitations of their work.

**Opportunities For Improvement:**

See the review above.

**Relation To Prior Work:**

The authors refer to the setting tackled as “off-dynamics RL.” However, the problem of dealing with varying dynamics in RL has been tackled in the literature with a variety of names. For instance, multi-task RL, non-stationary RL, contextual RL, etc. The authors should discuss these other problem settings, which are very related to off-dynamics RL.

Moreover, the authors should discuss the related benchmark CARL (context adaptive RL) (https://github.com/automl/CARL), which also includes many tasks where it is possible to modify the dynamics function.

**Summary And Contributions:**

This paper introduces a benchmark for off-dynamics reinforcement learning, i.e., reinforcement learning in which the agent is tasked with solving a “source” task and then has to solve a “target” task whose only the dynamics (state-transition function) is different from the source task. The proposed benchmark, ODRL, introduces locomotion and dexterous manipulation tasks, which are based on known Mujoco RL tasks, with different levels of changes in the dynamics (e.g., gravity, friction, morphology). The benchmark supports 4 different settings (online-online, offline-online, online-offline, offline-offline) depending on whether the source or target tasks need to be solved online or offline. The library also includes the implementation of various off-dynamics RL algorithms. Finally, the authors report the empirical results of the implemented methods in different settings and provide key observations regarding their findings.

---

> ### Author Rebuttal · Authors · 2024-08-18
>
> **[Rebuttal Part 1/3]**
>
> We thank the reviewer for the insightful review. We appreciate that the reviewer thinks that our paper is well-written, ODRL covers diverse tasks, and implements many off-dynamics RL algorithms. Please find our clarifications below. We hope the reviewer will be willing to raise the score if the concerns are addressed.
>
> **Concern 1: Have the benchmark separate from the algorithm implementations**
>
> Thanks for the comment. Actually, one does not need to have the RL algorithm implementation compliant with the `train.py` script to run their own algorithms in our benchmark. One can easily get environments from our benchmark without using the `train` script:
> ```
> from dataset.call_dataset                 import call_tar_dataset
> from envs.mujoco.call_mujoco_env          import call_mujoco_env
> from envs.adroit.call_adroit_env          import call_adroit_env
> from envs.antmaze.call_antmaze_env        import call_antmaze_env
> from envs.infos                           import get_normalized_score
> ```
> It is quite easy to use these functions, all one needs is a `dict` that contains `env_name` and `shift_level`, e.g.,
> ```
> tar_env_config = {
>             'env_name': 'ant-friction',
>             'shift_level': 0.5,
>         }
> ```
> Then the target environment with dynamics shift can be acquired:
> ```
> tar_env = call_mujoco_env(tar_env_config)
> ```
> Having the environment, one can test their algorithm with their own `train` script. **We emphasize that our benchmark is indeed separated from the algorithm implementations**. All algorithmic details are implemented in a single file instead of being involved in our `train.py` script.
>
> The `train` script is long since it includes four different experimental settings and handles different domains in ODRL (e.g., MuJoCo, Antmaze, Adroit). We would like to keep all details in a single file such that the newcomers can easily get familiar with the benchmark and can make some modifications quickly. The `train` script should not be difficult to read. However, we can understand that it would be better to move some codes to another file to keep the `train` script clean and concise. We could move the codes on setting seeds, creating environments, and loading datasets to another file if the reviewer deems it necessary.
>
> **Concern 2: Discussion of ODRL against other related benchmarks**
>
> Thanks for catching that. We agree that it counts to discuss other related RL problem formulations and include explicit comparison against existing RL benchmarks. We include the following benchmarks: D4RL [1], DMC suite [2], Meta-World [3], RLBench [4], CARL [5], and summarize the comparison below
>
> | Benchmark | Offline datasets | Diverse Domains | Multi-task | Single-task Dynamics Shift |
> | ----    | :---: | :---: | :---: | :---: |
> | D4RL | $\checkmark$ | $\checkmark$ | x | x |
> | DMC suite | x | $\checkmark$ | x | x |
> | Meta-World | x | x | $\checkmark$ | x |
> | RLBench | $\checkmark$ | x | $\checkmark$ | x |
> | CARL | x | $\checkmark$ | x | $\checkmark$ |
> | **ODRL** | $\checkmark$ | $\checkmark$ | x | $\checkmark$ |
>
> Notably, D4RL only contains single-domain offline datasets and does not address the off-dynamics RL issue. DMC suite contains numerous tasks, but it does not offer offline datasets and also does not address the off-dynamics RL problem. Meta-world is designed for a multi-task setting (i.e., transfer to a varied task). RLBench provides demonstrations for many tasks but it does not address the dynamics shift in a single task. CARL focuses on the setting where the context of the environment (e.g., reward, dynamics) can change between different episodes (i.e., it does not have a *source domain* or *target domain*, but only one domain where the dynamics or rewards can change depending on the context). CARL does not provide offline datasets. ODRL, instead, **focuses on the setting where the agent needs to leverage source domain data to facilitate the policy training in the target domain, where the task in the source domain and the target domain remain identical**. The discussion on different benchmarks would be incorporated into the revision. We have also updated our GitHub page to reflect that.
>
> **Concern 3: The experiments were performed with only 5 random seeds**
>
> We note that 5 random seeds are adopted by numerous prior works like H2O [6], VGDF [7], PAR [8], and benchmarks like CARL [5] (https://arxiv.org/pdf/2110.02102, as recommended by the reviewer). We clarify that the standard deviations of different methods can be affected by the methods themselves (e.g., CQL\_SAC, H2O, and RLPD in Figure 4 have much smaller standard deviations) and the environment (e.g., generally, different methods exhibit smaller standard deviations on the ant task than the walker2d task). Moreover, it would be very expensive and time-consuming if we ran algorithms for all our reported experiments across 10 seeds.
>
> Nevertheless, we added additional 5 seeds to the experiments reported in Figure 4 and Figure 5 of the main text to examine whether there would be a significant difference. We believe that 10 seeds are sufficient to show how the algorithm behaves in the environment and one can draw significant conclusions based on that. Please check Figure 1 and Figure 2 in the **rebuttal pdf** below for a comparison between results under 5 random seeds and results under 10 random seeds. The experiments show that their trends are similar and our observations in the main text are reliable.

---

> > ### Author Rebuttal · Authors · 2024-08-18
> >
> > **[Rebuttal Part 2/3] Continued**
> >
> > **Concern 4: Why should one use ODRL**
> >
> > Thanks for the comment. Yes, the tasks in our benchmark are also self-proposed. ODRL enjoys the following advantages, which we believe are reasons that one should consider using our benchmark:
> >
> > - ODRL covers a wide spectrum of dynamics shift tasks. Instead, prior works only conduct experiments on a limited range of dynamics shift tasks (e.g., H2O [6] only conducts experiments halfcheetah robot under 3 types of dynamics shifts).
> > - All ODRL tasks are adapted from some commonly used tasks (e.g., MuJoCo, Adroit), which should make it easier for newcomers to get familiar with the benchmark
> > - ODRL considers both slight dynamics shift and severe dynamics shift to comprehensively examine the dynamics adaptation ability of the agents under extreme or non-extreme cases (specified by shift level, e.g., the gravity in the target domain of ODRL can be 5 times that of the source domain). While some prior works only seem to include slight dynamics shift (e.g., in H2O [6], the friction in the source domain is only 0.3 times that of the target domain)
> > - ODRL covers varied experimental settings where the source domain and the target domain can be either online or offline, and we provide offline datasets with varied qualities for all benchmark tasks
> > - ODRL has an easy-to-read code style
> > - ODRL collects numerous off-dynamics RL algorithms that are implemented in a single-file manner with a unified code style
> > - ODRL would be actively maintained to include more real-world tasks and more recent off-dynamics RL algorithms. As an example, we would add Humanoid tasks into ODRL, and include the Sawyer robot tasks adapted from Meta-world (please check the `test` branch of our repo for more details: https://github.com/OffDynamicsRL/off-dynamics-rl/tree/test)
> >
> > **Concern 5: On the notation**
> >
> > We respectfully argue that such notation applies to the continuous control scenarios (i.e., the probability of the next state given the current state and action lies in [0, 1]). Such notation is also used by some prior works [9, 10].
> >
> > **Concern 6: Why did you not include the Humanoid robot**
> >
> > We do not include the Humanoid task since ODRL involves offline source domain setting where one needs to leverage source domain offline dataset, and D4RL does not support Humanoid tasks. Meanwhile, Humanoid tasks often consume more CPUs than other tasks like HalfCheetah. Since the reviewer demands, we would include the Humanoid robot in our benchmark. We now include friction shift and gravity shift for the Humanoid robot, and will include kinematic shift and morphology shift for the Humanoid robot in the near future. Please check our GitHub page for more details (https://github.com/OffDynamicsRL/off-dynamics-rl/tree/test/envs/mujoco/assets).
> >
> > **Concern 7: On obs2**
> >
> > PAR exhibits good performance under many dynamics shifts, but fails in the Antmaze domain and dexterous hand manipulation tasks. Here, *other domains* mean Antmaze domain and Adroit domain. We could revise it to *PAR achieves the best performance on locomotion tasks but fails on the Antmaze domain and Adroit domain* if the reviewer deems it misleading.
> >
> > **Concern 8: On the Antmaze task**
> >
> > Thanks for the question. We first clarify that the Antmaze task is very challenging, since it needs to control the ant robot (which is difficult) and navigate it towards the goal position, with its reward extremely sparse (the sparse reward setting is also challenging). Based on our experience and prior experiments, it is hard to get meaningful performance in online Antmaze tasks from D4RL (i.e., vanilla Antmaze) with algorithms like SAC. Nevertheless, it is possible to get strong agents with hierarchical RL methods in online Antmaze tasks (and that is how we gather datasets for Antmaze maps proposed by our own). That indicates that we may blame more on that existing methods are insufficient instead of that the task itself is not solvable (all Antmaze tasks that we propose are solvable). It is possible that existing methods can be enhanced by leveraging some exploration strategies (e.g., random network distillation), or goal-conditioned RL methods (e.g., hindsight experience replay).
> >
> > We believe that if there is a perfect algorithm (say, highly generalizable and efficient dynamics-aware RL algorithm), it should be able to learn a policy across different map layouts and obstacles (e.g., different maps in racing games). Nevertheless, we agree that our proposed Antmaze tasks are very challenging (and we believe this can also be a potential reason for choosing our benchmark as ODRL is challenging). Yet, we observe that methods like SAC\_CQL can learn some good performance on our Antmaze tasks given the online source domain and the offline target domain (please check Figure 18 in the appendix). We believe it is possible that stronger methods can be proposed to solve the Antmaze tasks.
> >
> > **Concern 9: On the failure of PAR in dexterous manipulation tasks**
> >
> > Following the PAR paper [8], PAR incorporates another reward penalty term to the source domain data compared to SAC used in Figure 3. We observe that SAC can exhibit good performance on dexterous hand manipulation tasks while PAR fails in the online-online setting. The only reason is that the reward penalty term given by PAR is poor. In fact, we observe severe Q value overestimation when running PAR on Adroit tasks. Upon inspection, we find that the average reward penalty given by PAR is almost 4 times greater than that given by DARC, leading to the excessive penalty to source domain data (note that we use a *fixed* reward penalty coefficient $\beta=0.1$ of PAR for all tasks in the online-online setting). This results in a poor Q value estimate and negatively affects the policy learning.

---

> > > ### Author Rebuttal · Authors · 2024-08-18
> > >
> > > **[Rebuttal Part 3/3] Continued**
> > >
> > > **Concern 10: What explains the performance of DARC decreasing so much on ant-friction-5.0**
> > >
> > > We believe the *overly pessimism issue* in DARC can explain its poor performance in the target domain of ant-friction-5.0, as criticized by VGDF [7] and PAR [8]. In practice, we observe the same phenomenon as that reported in Figure 6 of the PAR paper, i.e., the reward penalty given by DARC continuously increases in ant-friction-5.0. This shows that DARC can be overly pessimistic and the classifiers may fail to produce suitable reward penalties to compensate source domain data (we would like to note that this seems to rely on the specific environments, e.g., DARC can acquire quite good performance on Adroit tasks while other methods fail).
> > >
> > > Due to the space limit of the main text, we do not introduce how different algorithms transfer to the target domain in the main text. However, we defer all algorithmic details and implementation details to Appendix B, where it is super clear how different methods transfer to the target domain (detailed formulas and loss functions are provided). All off-dynamics RL algorithms reuse the source domain replay buffer while training for the target domain (this is required by the definition of off-dynamics RL setting in Definition 1). Source domain data is valuable since we have a limited budget for target domain data. It is expected that off-dynamics RL algorithms can leverage (possibly biased) source domain data to facilitate policy learning in the target domain through algorithmic advances.
> > >
> > > It is possible that the agent's performance would get hurt if the transitions are too different when the agent simply adopts the same loss function for source domain data and target domain data. However, recent off-dynamics RL methods mitigate this from different perspectives, e.g., PAR penalizes the source domain data that deviate far from the target domain, H2O leverages importance sampling weighting (the weight can be small if the transitions are too different), etc.
> > >
> > >
> > > **Concern 11: on the documentation**
> > >
> > > Thanks for the suggestion. We promise to include documentation of ODRL akin to that on the CARL website.
> > >
> > > **Concern 12: on the citation color**
> > >
> > > We would change the color to blue in the revision to mitigate such concerns.
> > >
> > > Hopefully, these can resolve the concerns. If there is still anything unclear, please let us know!
> > >
> > > **References**
> > >
> > > [1] D4rl: Datasets for deep data-driven reinforcement learning
> > >
> > > [2] Deepmind control suite
> > >
> > > [3] Meta-World: A Benchmark and Evaluation for Multi-Task and Meta Reinforcement Learning
> > >
> > > [4] RLBench: The Robot Learning Benchmark \& Learning Environment
> > >
> > > [5] Contextualize Me - The Case for Context in Reinforcement Learning
> > >
> > > [6] When to trust your simulator: Dynamics-aware hybrid offline-and-online reinforcement learning
> > >
> > > [7] Cross-domain policy adaptation via value-guided data filtering
> > >
> > > [8] Cross-domain policy adaptation by capturing representation mismatch
> > >
> > > [9] Benchmarking Deep Reinforcement Learning for Continuous Control
> > >
> > > [10] A framework for transforming specifications in reinforcement learning

---

> > > > ### Comment · Reviewer_5MsE · 2024-08-18
> > > >
> > > > I thank the authors for taking the reviewer's feedback into account and for the detailed response. I will try to reply to each comment below:
> > > >
> > > > **Concern 1:** Thank you for the clarification. I strongly suggest that you provide a more direct API with corresponding documentation for instantiating the environments in the benchmark. Moreover, it would be extremely useful if the benchmark supported stablished API's (e.g., Gymnasium API), so that users can easily use the environments with stablished RL libraries (e.g., stable-baselines, clean-rl).
> > > >
> > > > **Concern 2:** Thank you for including this discussion, which is very important to more clearly state the relevance of the benchmark.
> > > >
> > > > **Concern 3:** It is very unfortunate that in our field it become acceptable to draw ranking conclusions based on ~5 random seeds. Notice, however, that the number of required seeds is not defined by community common practice. It is OK to say method A appears better than B in our experiments without claiming significance. It is not okay to make ranking claims based on too few seeds (3 and 5 are way too few). How many seeds do we need? This is a statistical question based on the underlying performance distributions.
> > > >
> > > > **Concern 4:** Thanks for this clarification. I suggest including some of these points in the Introduction or Related Work section.
> > > >
> > > > **Concern 5:** For continuous distributions, the probability of any single point is zero, instead of a value in $[0,1]$. If your state space is continuous, you will have a probability density function instead, defined as $P: S \times A \times S \rightarrow [0,\infty]$. Notice that [9] **does not** define $P$ in the same way as you: they define it as $P: S \times A \times S \rightarrow \mathbb{R}$.
> > > >
> > > > **Concern 11:** That is excellent. Good documentation is always very important for any library or benchmark.
> > > >
> > > > I am increasing my score due to the authors' responses.

---

> > > > > ### Author Response · Authors · 2024-08-19
> > > > > **Thank you for your comments!**
> > > > >
> > > > > We thank the reviewer for the prompt response and for raising the score. Please check our replies to the comments.
> > > > >
> > > > > **Concern 1**: Thanks for the kind suggestion. We now include a `call_odrl_env.py` (https://github.com/OffDynamicsRL/off-dynamics-rl/blob/test/envs/call_odrl_env.py) where one can call environments from ODRL through a more direct API, i.e.,
> > > > >
> > > > > ```
> > > > > call_odrl_env(env_type='mujoco',
> > > > >                   env_name='halfcheetah-friction',
> > > > >                   shift_level='0.5')
> > > > > ```
> > > > >
> > > > > One only needs to specify the environment type (e.g., mujoco, antmaze, adroit, sawyer), the environment name, and the shift level. We believe this matches the expectations of the reviewer. Furthermore, we provide another function `call_odrl_dataset` that directly returns the offline target domain datasets in ODRL through a direct API,
> > > > >
> > > > > ```
> > > > > call_odrl_dataset(env_name='halfcheetah-friction',
> > > > >                       shift_level='0.5',
> > > > >                       dataset_type='random',
> > > > >                       )
> > > > > ```
> > > > >
> > > > > We plan to support Gymnasium API in the future.
> > > > >
> > > > > **Concern 3**: Yes, there is no determinant conclusion on how many seeds one should use in the context of RL. We agree that it is a statistical question based on the underlying performance distributions. We could add more seeds to our reported experiments to mitigate this concern, if the reviewer deems it necessary. (This could be very expensive, and we would do our best to fulfill that.)
> > > > >
> > > > > **Concern 4**: We are more than happy to include these points in our paper
> > > > >
> > > > > **Concern 5:** Thanks. We would modify the notations on the transition dynamics in the revision.
> > > > >
> > > > > **Concern 11**: We agree that good documentation is always very important for any library or benchmark. We are proud to announce that we have now included an initial version of our documentation site (http://off-dynamics-rl.readthedocs.io/) where we include some introductions, illustrations, and examples for instantiating the environments, as we promised. We build the documentation site with `mkdocs`. Due to time constraints, only limited content can be included in the documents, and we promise to continuously update the documents to make them clearer and more comprehensive.
> > > > >
> > > > > By the way, we are also happy to share with the reviewer that we also uploaded the `random` and `medium` datasets for the humanoid robot in the source domain (please check https://drive.google.com/drive/folders/1fwkjtXCbMxVP7RM7NSN3mF40Gakwkqei?usp=sharing). More datasets would be included!
> > > > >
> > > > > Please let us know if the reviewer has any further suggestions or questions!

---

### Official Review · Reviewer_Gwro · 2024-07-16
**A simple benchmark with a deep, comprehensive evaluation**

**Rating:** 7
**Confidence:** 4
**Correctness:** The benchmark and evaluation design a…

**Review:**

######## Strengths ########

1. The empirical evaluation is comprehensive, deep, and insightful.
2. There are sufficient details to understand the implementation of the benchmark
3. Including implementations of a collection of transfer RL baselines could lower the barrier of entry into this still-niche field.

######## Opportunities ########

1. The locomotion tasks are strikingly similar to those provided in [1]. More generally, the related work misses any transfer RL or related benchmarks.
2. It is unclear how the AntMaze tasks would constitute a shift in dynamics
3. It would be good to include some discussion about how the single-source/single-target setting continues to be relevant in this age of massive datasets

######## Recommendation ########

I recommend that this paper be accepted for publication. While the proposed collection of tasks is not itself especially novel, the empirical evaluation of a collection of transfer RL methods, including the authors' own proposed baselines, is excellent. The authors provide 11 well written, well analyzed observations drawn from their evaluations, which I believe are interesting and insightful.

######## Arguments ########

The main strength of this submission is the quality of the empirical evaluation and the analysis that follows.
- The baselines are well-chosen and cover a spectrum of plausible transfer methods
- The plots are well designed, especially the radar chart in Fig 3. They enable immediately observing that there are trade-offs across various choices of transfer algorithms
- The analysis covers both online and offline methods.
- Some of the interesting insights include:
    - Training a single SAC agent across source and target domains performs well compared to purpose-built transfer method on some of the benchmark tasks
    - Using higher-quality source data (e.g., expert trajectories instead of medium-performance trajectories, from benchmarks like D4RL) does not necessarily improve the performance on the target domain---this is in direct contrast to the improvement in performance on the source domains
    - Methods that perform poorly in the source domain might still perform well on the target domain

Beyond this, the benchmark itself is well explained and well documented.

This is also the first time I've seen a collection of transfer RL implementations in a single place. Just like baseline implementations like stablebaselines, CleanRL... have lowered the barrier of entrance into deep RL research, I am hopeful that such an implementation of transfer RL methods might lower the barrier of entrance into transfer RL.

The submission does have a significant gap in the related work discussion: it does not include any existing benchmark for transfer or multitask RL. In particular, the authors did not mention gym-extensions [1], which introduces variations across Mujoco environments that are very similar to those in the authors' locomotion tasks (to the point that I could not consider these tasks to be novel). The authors should discuss how the novelty in their benchmark lies primarily in the proposed evaluation setting (gym-extensions was not designed for single-source/single-target transfer, but for batch multitask training) and in executing comprehensive evaluations in this setting. In addition, I encourage the authors to include a more complete discussion of related benchmarks (e.g., [2], [3]...).

Another concern that I have is about the definition of map layout variations as off-dynamics in AntMaze. I believe that in this case, only the state space changes (but the transition function is the same). Technically it could be off-dynamics depending on the observation space (e.g., if the observation is just the ant's pose, without any information about obstacles, then obstacles become part of the dynamics), but that seems like a bit of a stretch. Could the authors comment on this point?

I'm also curious about one more point: how the authors view the relevance of the single-source/single-target setting in today's large-data age. In particular, the variations performed across tasks in this work are relatively small compared to other cross-task variations in the literature. Yet existing works have been able to learn policies across many diverse environments/tasks/dynamics by leveraging access to larger benchmarks that essentially expose the learner to the entire distribution of possible worlds. What is the authors' perspective on how the single-source/single-target setting continues to be relevant today?

[1] Henderson et al. "Benchmark Environments for Multitask Learning in Continuous Domains." ICML Lifelong RL Workshop 2017.

[2] Yu et al. "Meta-World: A Benchmark and Evaluation for Multi-Task and Meta Reinforcement Learning." CoRL 2019.

[3] James et al. "RLBench: The Robot Learning Benchmark & Learning Environment." ICRA 2020.

**Strengths:**

1. The empirical evaluation is comprehensive, deep, and insightful.
2. There are sufficient details to understand the implementation of the benchmark
3. Including implementations of a collection of transfer RL baselines could lower the barrier of entry into this still-niche field.

**Additional Feedback:**

The following points are provided as feedback to hopefully help better shape the submitted manuscript, but did not impact my recommendation in a major way.

Intro
- The intro and abstract are clear enough.
- How do different map structures imply different dynamics? (Fig 1)

Sec 4
- I'd be interested in hearing a bit more about the motivation for having the 4 possible combinations of online/offline for the source/target domain. I find it particularly hard to find settings where we would have an online training setting for the source domain but an offline setting for the target domain. Maybe the source task is trained online in a simulator, but online training in the real-world is unsafe so we use some other data-collection scheme and train via off-line RL? Could the authors provide some justification for each of the settings?

Sec 5
- It would be nice to see how SAC or other vanilla baselines would do by just training on the target domains w/o transfer (as hopefully a sort of lower bound)

**Clarity:**

The paper is easy to follow, though at times it becomes a bit repetitive in the earlier sections.

**Documentation:**

The benchmark is well documented.

**Limitations:**

Limitations are adequately addressed.

**Opportunities For Improvement:**

1. The locomotion tasks are strikingly similar to those provided in [1]. More generally, the related work misses any transfer RL or related benchmarks.
2. It is unclear how the AntMaze tasks would constitute a shift in dynamics
3. It would be good to include some discussion about how the single-source/single-target setting continues to be relevant in this age of massive datasets

**Relation To Prior Work:**

Some key prior work, pointed out above, is missing.

**Summary And Contributions:**

The submission proposes a benchmark to study single-source/single-target transfer in off-dynamics RL. The benchmark contains locomotion, navigation, and dexterous manipulation tasks, with variations in friction, gravity, kinematic constraints, and agent morphology. Each source task is a standard RL task from the literature (e.g., HalfCheetah, AntMaze, pen spinning), and each target task is a variation along one dimension in the dynamics. Both the source and target domains can be trained via online or offline RL. The authors include offline RL datasets from prior work for the source domains, and their own datasets for offline RL on the target domains (smaller datasets, to evaluate transfer data efficiency). The authors evaluate a collection of transfer RL baselines across a subset of their benchmark.

---

> ### Author Rebuttal · Authors · 2024-08-18
>
> **[Rebuttal Part 1/3]**
>
> We thank the reviewer for the thoughtful review. We appreciate that the reviewer comments that our evaluation is comprehensive, deep, and insightful, and ODRL could lower the barrier of entry into this field. Please find our clarification of the concerns below.
>
> **Concern 1: The locomotion tasks are similar to those provided in [1]**
>
> Thank you very much for recommending the gym-extensions benchmark [1]. Yes, it seems that many of the constructed tasks in gym-extensions are similar to those in ODRL (e.g., gravity shift, morphology shift). We would like to clarify that:
>
> - many of the ODRL tasks are actually motivated by prior works, e.g., H2O [2] self-constructs gravity shift tasks for halfcheetah, VGDF [3] self-constructs kinematic shift and morphology shift tasks. What makes ODRL different from them is that ODRL covers both *slight dynamics shift* and *severe dynamics shift*. For example, the target domain gravity is modified to 1.5 times the source domain gravity at most in Gym-extensions, while the target domain gravity can be 5.0 times the source domain gravity.
> - ODRL collects a wider range of tasks and dynamics shifts, e.g., friction shifts, Adroit tasks, etc.
> - Gym-extensions is intrinsically designed for *multi-task* scenarios, this makes the problem setting of Gym-extensions and ODRL different
>
> Nevertheless, they are still quite relevant. We recognize that it is vital to cite [1] and other related papers. We commit to including the discussions on prior benchmarks in the revision (we have updated our GitHub page to include the detailed discussion of ODRL against other relevant benchmarks, please check them out!).
>
> The main novelty of ODRL lies in the following aspects:
>
> - a formal definition of off-dynamics RL setting
> - a wide collection of tasks under diverse dynamics shifts (e.g., gravity, friction, morphology) and under diverse domains (MuJoCo, Antmaze, Adroit)
> - a collection of four different learning paradigms
> - a collection of various off-dynamics RL algorithms (and some baselines proposed by our own), all in a single-file implementation manner
> - a collection of offline datasets for each dynamics shift task
> - comprehensive evaluations under benchmark tasks
>
> **Concern 2: It is unclear how the AntMaze tasks would constitute a shift in dynamics**
>
> Yes, varied map layouts in Antmaze can be regarded as off-dynamics since obstacles can be seen as part of the dynamics. For example, the *antmaze-small-empty* map and *antmaze-small-centerblock* map can have distinct dynamics (the ant robot in the centerblock map can only bypass the obstacle in the middle of the maze to get to the target location, while the ant robot in the empty map can reach the middle of the maze). The existence of different obstacles in the source domain and the target domain incur differences in their transition dynamics $P\_{\rm src},P\_{\rm tar}$. This meets the definition of off-dynamics RL setting in the paper.
>
> We also would like to note that Gym-extensions also include tasks with obstacles, and the DARC paper [2] also includes environments containing obstacles for evaluating the off-dynamics RL algorithm. We then believe the Antmaze map layout setup in our paper belongs to the off-dynamics setting.
>
> **Concern 3: Lacking a discussion on existing benchmark for transfer or multitask RL**
>
> Thanks for the comment. We are sorry for missing the comparison of ODRL to some multi-task RL benchmark or transfer RL benchmark. As a remedy, we discuss the following widely used benchmarks: Gym-extension [1], D4RL [3], DMC suite [4], Meta-World [5], RLBench [6], CARL [7], Continual World [8], and summarize the comparison below
>
> | Benchmark | Offline datasets | Diverse Domains | Multi-task | Single-task Dynamics Shift |
> | ----    | :---: | :---: | :---: | :---: |
> | D4RL | $\checkmark$ | $\checkmark$ | x | x |
> | DMC suite | x | $\checkmark$ | x | x |
> | Meta-World | x | x | $\checkmark$ | x |
> | RLBench | $\checkmark$ | x | $\checkmark$ | x |
> | CARL | x | $\checkmark$ | x | $\checkmark$ |
> | Gym-extensions | x | x | $\checkmark$ | $\checkmark$ |
> | Continual World | x | x | $\checkmark$ | x |
> | **ODRL** | $\checkmark$ | $\checkmark$ | x | $\checkmark$ |
>
> Among these benchmarks, D4RL only contains single-domain offline datasets and does not focus on the off-dynamics RL issue. DMC suite contains a wide range of tasks, but it does not offer offline datasets and does not handle the off-dynamics RL. Meta-world is designed for the multi-task RL setting. RLBench provides demonstrations for numerous tasks but it does not involve the dynamics shift in a single task. CARL focuses on the setting where the context of the environment (e.g., reward, dynamics) can change between different episodes (i.e., it does not have a *source domain* or *target domain*, but only one domain where the dynamics or rewards can change depending on the context). CARL also does not provide offline datasets. Continual World is a benchmark for continual learning in RL. It also supports multi-task learning and can be used for transfer RL policies. ODRL, instead, **focuses on the setting where the agent can leverage source domain data to facilitate the policy training in the target domain, where the task in the source domain and the target domain remain identical**.

---

> > ### Author Rebuttal · Authors · 2024-08-18
> >
> > **[Rebuttal Part 2/3] Continued**
> >
> > **Concern 4: how the authors view the relevance of the single-source/single-target setting in today's large-data age**
> >
> > Interesting question! We totally agree that existing works on cross-task transfer and methods that learn policies across many diverse environments/tasks by leveraging larger benchmarks or datasets are impressive. For example, Multi-game decision transformer [9] trains a single model across multiple Atari games and generalizes to unseen Atari games. Nevertheless, the success of such scenarios does not necessarily indicate that single-source/single-target setting is meaningless. First, ODRL focuses on the problem of policy adaptation under varied dynamics, and with the same task, instead of multi-task or cross-domain policy transfer. They generally solve different problems and have different potential applications. It is possible that algorithms that train on large data/environments/tasks can exhibit some generalization ability on tasks with dynamics shifts. However, training such an algorithm can be expensive and may be negatively affected by some training tasks when transferring to the new environment (e.g., the model may exhibit some bias). The single-source/single-target setting, instead, is cheaper and can be more efficient (e.g., one can find that multi-game decision transformer still underperforms policies that are trained in the single domain). Second, sometimes it is quite difficult and expensive to collect diverse, large-scale robotic data or have numerous robotic environments for the agent to interact with, especially in real-world scenarios. These pose challenges for large-data training. However, it is easier to acquire one source domain and one target domain. Third, large-scale data training methods may not be applicable in some scenarios, e.g., game AI. Different games have quite difficult and complex dynamics, features and characters, etc. It is nearly impossible to train the agent across varied games and then transfer it to the new game (even if it is possible, the performance can be quite poor). It is less possible that one can get access to diverse tasks/environments in a fixed game. The single-source/single-target setting can hence be better applicable here. Fourth, developing methods for training on large-scale environments/tasks is not contradictory with developing methods that can adapt policy from the single source domain to the single target domain, as the latter can be incorporated into the former (e.g., one can borrow some methods from off-dynamics RL when adapting the pre-trained agent to a new environment with dynamics shift).
> >
> > To summarize, the rationality of single-source/single-target setting lies in:
> >
> > - single-source/single-target can exhibit better performance than training on large-scale tasks/environments, and is less affected by the bias in the training data
> > - single-source/single-target setting can be cheaper and more efficient, while training on large-scale data can be expensive. It can be difficult to collect large-scale data/environments in real-world scenarios
> > - single-source/single-target setting has its unique value, e.g., in the field of game AI
> > - single-source/single-target setting is not contradictory against cross-task methods or methods that train on large-scale data, and one can borrow some off-the-shelf methods/ideas from off-dynamics RL to facilitate cross-task or large-scale data training or policy transfer.
> >
> > Moreover, ODRL can be naturally extended to the multi-source/single-target or multi-source/multi-target setting, or other settings like cross-task policy transfer (as we wrote in Line 1803-1807 in Appendix G). That said, one can also use ODRL for training cross-task policies or train agents by leveraging multiple environments/tasks.
> >
> > **Concern 5: Justifications for each of the experimental settings**
> >
> > We justify the rationality of each experimental setting below by using robotics and game AI as examples:
> >
> > - **Online-online**. The robot can interact with a (possibly biased) simulator and the real world simultaneously. Indeed, it can be unsafe to directly learn online in the real world. One can incorporate an action filtering procedure and filter out those unsafe actions. Furthermore, we limit the interaction steps in the target domain (as emphasized in the definition of off-dynamics RL), which can mitigate such concerns to some extent. For game AI, one may develop another version of the game (say, Version A and B). In Version B, the skills of some characters may change and we want to leverage Version A data for training game AI in Version B instead of learning from scratch.
> >
> > - **Offline-online**. We may have some previously logged real-world robot datasets, while the robot may exhibit dynamics shift (e.g., kinematic shift due to joint component deterioration, morphology shift due to body part replacement). This means that even the same real-world robot may undergo dynamics shift. Also, the offline datasets can be gathered in the (possibly biased) simulator. Again, the action filtering procedure can be applied to exclude unsafe actions. For game AI, we may have some offline datasets from Version A (either gathered by game AI or human players in Version A), and can interact with Version B to train game AI in Version B.
> >
> > - **Online-offline**. The robot can interact with the (possibly biased) simulator and can simultaneously get access to a real-world (target domain) offline dataset with a limited size (e.g., human demonstrations gathered by humans). For game AI, the agent can interact with the Version A environment while can only get access to some Version B offline datasets (e.g., collected by some human players or random policies).

---

> > > ### Author Rebuttal · Authors · 2024-08-18
> > >
> > > **[Rebuttal Part 3/3] Continued**
> > >
> > > - **Offline-offline**. The robot can get access to offline datasets from the simulator (or previous real-world robot datasets), as well as limited offline datasets from the real-world robot (e.g., collected by humans). For game AI, the dataset from Version A is offline (can be gathered by either AI in Version A or human players), and the agent can also get access to Version B dataset.
> > >
> > > **Concern 6: how SAC or other vanilla baselines would do by just training on the target domains w/o transfer in Sec 5**
> > >
> > > Thanks for the suggestion. To check how SAC behaves without transfer, we run SAC purely in the online target domain for 1M steps on the same environments in Figure 3 of our paper. In the main text, the agent can interact with the source domain for 1M steps while can only get access to the target domain for 0.1M steps in the online-online setting. Then we use the average return of SAC at 0.1M steps in the target domain as the *lower bound* while its performance at 1M steps in the target domain as the *upper bound*. We report the normalized score on each task here and also compare the performance of SAC in the target domain against the algorithms in Figure 3. The results are averaged across 5 seeds. Please check the results in Table 1 and Table 2 of the **rebuttal pdf** below.
> > >
> > > Hopefully, these can resolve the concerns. If there is still something unclear, please let us know!
> > >
> > > **References**
> > >
> > > [1] Benchmark Environments for Multitask Learning in Continuous Domains
> > >
> > > [2] Off-dynamics reinforcement learning: Training for transfer with domain classifiers
> > >
> > > [3] D4rl: Datasets for deep data-driven reinforcement learning
> > >
> > > [4] Deepmind control suite
> > >
> > > [5] Meta-World: A Benchmark and Evaluation for Multi-Task and Meta Reinforcement Learning
> > >
> > > [6] RLBench: The Robot Learning Benchmark \& Learning Environment
> > >
> > > [7] Contextualize Me - The Case for Context in Reinforcement Learning
> > >
> > > [8] Continual World: A Robotic Benchmark For Continual Reinforcement Learning
> > >
> > > [9] Multi-Game Decision Transformers

---

> > > > ### Comment · Reviewer_Gwro · 2024-08-21
> > > >
> > > > Thank you for your comprehensive responses. I do not have any further questions and will review my score with the other reviewers' during the discussion period.

---

> > > > > ### Author Response · Authors · 2024-08-21
> > > > >
> > > > > Thanks for the prompt response. We would incorporate the suggestions from the reviewer into the revision (e.g., comparison against other benchmarks, citing and discussing Gym-extensions and other related papers). We are more than happy to answer new questions if any arise.

---

### Official Review · Reviewer_XwbG · 2024-07-18
**Review for ODRL**

**Rating:** 6
**Confidence:** 3
**Correctness:** Yes
**Clarity:** Yes

**Review:**

The introduction of ODRL as a benchmark tailored for off-dynamics RL is a significant contribution, addressing the critical need for standardized evaluation in this field. The benchmark's comprehensive nature, covering a wide array of dynamics shifts and task categories, is a major strength, offering researchers a versatile platform for rigorous testing of RL algorithms' adaptability.

However, the paper could be strengthened by addressing its limitations more thoroughly. Specifically, a deeper exploration of the benchmark's applicability to real-world scenarios would be valuable. While simulation-based tasks provide a controlled environment for initial evaluations, real-world applications often present more complex and unpredictable dynamics shifts. Discussing strategies for bridging this gap could make the research more impactful. Additionally, the paper could benefit from a more explicit comparison with existing benchmarks in the field of RL. By highlighting the unique features and advantages of ODRL, the authors could better position their contribution within the broader research landscape.

Besides, I have one question: Is universal off-dynamics RL really necessary due to the wide variety of real-world tasks? Isn't it great that one algorithm works well in a limited domain? Isn't it too harsh to expect an algorithm to work well in all task domains?

**Strengths:**

1. ODRL's wide-ranging spectrum of dynamics shifts and tasks across different domains significantly contributes to the field of RL. Its comprehensive nature ensures that it can be a valuable tool for testing the robustness and adaptability of off-dynamics RL algorithms.

2.  The paper presents a thorough experimental evaluation of different algorithms across various settings, providing insightful observations and highlighting the challenges in developing universally effective off-dynamics RL methods. This depth of analysis is beneficial for guiding future research directions.

**Additional Feedback:**

cf. detailed review above

**Documentation:**

Yes

**Limitations:**

Yes

**Opportunities For Improvement:**

1. While the paper mentions the challenge of adapting policies to real-world scenarios, it does not deeply explore how the benchmark can be extended or adapted for real-world applications. Given the complexity of real-world dynamics, further discussion on this aspect could enhance the paper's relevance.

2. This paper introduces ODRL as a pioneering benchmark, but does not provide a detailed comparison with existing benchmarks or explain how ODRL fills the gaps left by these benchmarks. The significance and novelty of ODRL can be better demonstrated by comparative analysis. There are no other comprehensive benchmarks available, but the shortcomings of the evaluation benchmarks used in other papers should be discussed.

**Relation To Prior Work:**

Yes

**Summary And Contributions:**

The paper introduces ODRL, a benchmark designed for evaluating off-dynamics reinforcement learning (RL), focusing on the transfer of policies across different domains with dynamics mismatches. ODRL offers a unified framework that encompasses a wide variety of dynamics shifts across numerous task categories, including locomotion, navigation, and dexterous manipulation. It includes both online and offline domains, allowing for diverse experimental settings. The benchmark is equipped with numerous algorithms and baselines, aiming to facilitate a comprehensive evaluation of the adaptation capabilities of RL agents. Extensive experiments across varied dynamics shifts reveal that no single method uniformly excels, underscoring the complexity of off-dynamics RL and the necessity for further research in developing more generalized dynamics-aware algorithms.

---

> ### Author Rebuttal · Authors · 2024-08-18
>
> **[Rebuttal Part 1/2]**
>
> We thank the reviewer for the insightful review. We appreciate that the reviewer acknowledges that ODRL is comprehensive and our analysis is in-depth and insightful. Please find our clarifications below.
>
> **Concern 1: A deeper exploration of the benchmark's applicability to real-world scenarios**
>
> We agree that real-world applications often present more complex and unpredictable dynamics shifts. ODRL bridges this gap with the following strategies:
>
> - ODRL covers numerous kinds of dynamics shifts (kinematic, morphology, gravity, friction, map layout) with various shift levels. In this way, many (potentially unpredictable) dynamics shifts that may occur in real-world applications can be included. Though many of our adopted robots are simple (e.g., halfcheetah), the problem setting is general and common, and we believe that there are conclusions/observations that can be borrowed from these simulated off-dynamics tasks.
>
> - ODRL also includes dexterous hand manipulation tasks, which are complex and resemble real-world robotic hand tasks.
>
> - ODRL can serve as a reliable testbed for off-dynamics RL algorithms. If one algorithm exhibits quite strong performance across numerous tasks from ODRL, then one can feel more confident to apply it in real-world applications, otherwise, it is better not to deploy the specific algorithm. Since simulation is cheap and can run fast, one can choose which algorithm to use in practice with ODRL. It is also a potential applicability of our benchmark to real-world scenarios.
>
> To further mitigate such concern, we provide other strategies:
>
> - **Mixing dynamics shifts in ODRL**. In existing ODRL benchmark tasks, all environments only involve one single dynamics shift. We could add more tasks where two or more dynamics shifts occur at the same time, e.g., the kinematic shift and the friction shift occur simultaneously. This aligns with the real-world scenarios where multiple dynamics shifts may exist.
>
> - **Expanding benchmark tasks to include more real-world robotic tasks**. This is perhaps the most intuitive way of enhancing the applicability. We are proud to announce that we are trying to include dynamics shift tasks using the Sawyer robots from Meta-World [1]. We now support kinematic shifts (easy/medium/hard) and morphology shifts (easy/medium/hard) for five Sawyer robot tasks. The Sawyer robot is widely used in academic or real-world robotics research. In the future, we plan to support all 50 Meta-World tasks. Please check our code (https://github.com/OffDynamicsRL/off-dynamics-rl/tree/test) for more details. We also attach the visualizations of our modified Sawyer robot tasks to the **rebuttal pdf** below. Please check them out!
>
> **[NOTE: the Sawyer robot environments are still under development, please allow us some time to test them.]**
>
> **Concern 2: A more explicit comparison with existing RL benchmarks or tasks used in prior work**
>
> Thanks for the kind suggestion. We agree that it counts to include explicit comparisons against existing RL benchmarks. We include the following widely used benchmarks: D4RL [2], DMC suite [3], Meta-World [1], RLBench [4], CARL [5], and summarize the comparison below (we have also updated our Github page on comparison against other benchmarks)
>
> | Benchmark | Offline datasets | Diverse Domains | Multi-task | Single-task Dynamics Shift |
> | ----    | :---: | :---: | :---: | :---: |
> | D4RL | $\checkmark$ | $\checkmark$ | x | x |
> | DMC suite | x | $\checkmark$ | x | x |
> | Meta-World | x | x | $\checkmark$ | x |
> | RLBench | $\checkmark$ | x | $\checkmark$ | x |
> | CARL | x | $\checkmark$ | x | $\checkmark$ |
> | **ODRL** | $\checkmark$ | $\checkmark$ | x | $\checkmark$ |
>
> As seen, D4RL only contains single-domain offline datasets and does not address the off-dynamics RL issue. DMC suite contains numerous kinds of tasks, but it does not offer offline datasets and also does not handle the off-dynamics RL. Meta-world is designed for a multi-task setting. RLBench provides demonstrations for many tasks but it does not address the dynamics shift in a single task. CARL focuses on the setting where the context of the environment (e.g., reward, dynamics) can change between different episodes (i.e., it does not have a *source domain* or *target domain*, but only one domain where the dynamics or rewards can change depending on the context). CARL does not provide offline datasets. ODRL, instead, **focuses on the setting where the agent needs to leverage source domain data to facilitate the policy training in the target domain, where the task in the source domain and the target domain remain identical**.
>
> We also compare ODRL against some off-dynamics RL papers (where they use their self-proposed environments) and summarize the comparison below.
>
> - DARC [6] proposes three broken tasks and one task with obstacles. The dynamics shift tasks used for experiments are relatively limited (e.g., mainly considering the broken tasks) and the modified task is relatively simple (e.g., the broken torque is only applied to one joint in ant robot). Meanwhile, it only considers the online source domain and online target domain setting
>
> - H2O [7] conducts experiments merely on the simulated halfcheetah robot with gravity/friction/action shift. Its experimental scope is quite limited and it only considers online source domain and offline target domain setting.
>
> - VGDF [8] and PAR [9] only conduct experiments on MuJoCo kinematic and morphology shifts, and do not consider other tasks like map layout or more complex tasks like dexterous hand manipulation tasks.
>
> - BOSA [10] and DARA [11] consider the offline source domain and offline target domain setting, where they only use mass shift and joint noise shift.

---

> > ### Author Rebuttal · Authors · 2024-08-18
> >
> > **[Rebuttal Part 2/2] Continued**
> >
> > ODRL, instead, covers a wide spectrum of tasks including gravity shift, friction shift, kinematic shift, morphology shift, and map layout difference, all with varied shift levels. Some tasks in ODRL are far more challenging than tasks used in prior work since the dynamics shift is more severe here (e.g., adroit shrink finger tasks, the gravity in the target domain can be 5 times that in the source domain, etc.) We cover not only MuJoCo tasks but also Antmaze and Adroit tasks. We offer four distinct experimental settings, where either the source domain or the target domain can be online or offline. We provide offline datasets for all tasks and include numerous off-dynamics RL algorithms that are implemented in a single-file manner. These are all notable contributions that ODRL brings.
> >
> > Please let us know if the reviewer suggests a comparison against any other benchmarks or papers.
> >
> > **Concern 3: Is universal off-dynamics RL really necessary due to the wide variety of real-world tasks?**
> >
> > Yes, we think the universal off-dynamics RL is necessary. As the reviewer commented, *real-world applications often present more complex and unpredictable dynamics shifts*. Then it is natural that one expects general and universal off-dynamics RL algorithms that can quickly adapt to the possibly unknown dynamics in the target domain. It is totally fine to have a method that can work on limited domains. However, it may be expensive to choose an appropriate algorithm for the specific domain since one needs to do trial and error. Furthermore, we firmly believe that generalization and transfer capability are always expected for strong AI agents. Hence, enhancing the generalization and transfer ability of the agent and developing a universal off-dynamics RL algorithm is a valuable and meaningful direction.
> >
> > We further would like to clarify that we never claim that one algorithm must exhibit strong performance on all task domains, but one algorithm should have strong policy adaptation capability across numerous domains (we agree that it would be too harsh to expect an algorithm to work well in all task domains, at least at the current stage). ODRL can serve as a reliable benchmark for evaluating the true policy adaptation of the algorithm. Then one can decide which algorithm to choose under different scenarios. With the advances in RL itself and other fields like LLM, it may be possible that future off-dynamics RL algorithms can be general enough that handle most dynamics shifts.
> >
> > Hopefully, these can resolve the concerns. If there is still something unclear, please let us know!
> >
> > **References**
> >
> > [1] Meta-World: A Benchmark and Evaluation for Multi-Task and Meta Reinforcement Learning
> >
> > [2] D4rl: Datasets for deep data-driven reinforcement learning
> >
> > [3] Deepmind control suite
> >
> > [4] RLBench: The Robot Learning Benchmark \& Learning Environment
> >
> > [5] Contextualize Me - The Case for Context in Reinforcement Learning
> >
> > [6] Off-dynamics reinforcement learning: Training for transfer with domain classifiers
> >
> > [7] When to trust your simulator: Dynamics-aware hybrid offline-and-online reinforcement learning
> >
> > [8] Cross-domain policy adaptation via value-guided data filtering
> >
> > [9] Cross-domain policy adaptation by capturing representation mismatch
> >
> > [10] Beyond ood state actions: Supported cross-domain offline reinforcement learning
> >
> > [11] Dara: Dynamics-aware reward augmentation in offline reinforcement learning

---

> > > ### Author Response · Authors · 2024-08-28
> > > **Following up with Reviewer XwbG**
> > >
> > > Dear Reviewer XwbG, thanks for your helpful review. As the author-reviewer discussion period is about to close in 4 days, we wonder if our rebuttal addresses your concerns or if there are any new questions, such that we would still have enough time to address them. It would be great if you could check our repo (https://github.com/OffDynamicsRL/off-dynamics-rl/tree/test) for new Sawyer tasks, and a comparison of ODRL against other benchmarks. We also include a documentation website for ODRL (http://off-dynamics-rl.readthedocs.io/) following the suggestion of Reviewer 5MsE.
> > >
> > > Please let us know if there is anything unclear! We are ready to answer any new questions or provide further clarifications.
> > >
> > > The authors

---

### Official Review · Reviewer_ciZk · 2024-07-21

**Rating:** 8
**Confidence:** 5
**Correctness:** Yes
**Clarity:** Yes

**Review:**

Quality:
1. The paper clearly defines the off-dynamics RL setting, which is a key strength. The formal problem definition establishes a solid theoretical foundation for the benchmark.
2. The development of the ODRL benchmark is an important contribution to the off-dynamics RL community, as this field previously lacked a unified standard for evaluation. The comprehensive and diverse nature of the benchmark tasks is a major strength.

Clarity:
1. The paper provides clear and detailed descriptions of the dynamics modifications in the locomotion, AntMaze, and dexterous manipulation environments. The technical explanations are well-executed.
2. While Figure 3 may be challenging to interpret, the authors could consider alternative visualization methods, such as bar plots or heatmaps, to better convey the performance comparison across the different task categories.

Originality:
1. The paper's approach of evaluating algorithms across the four experimental settings and the insightful observations derived from the results are valuable contributions that can guide future research.

Significance:
1. This paper is of high quality and effectively addresses the lack of a unified standard in the off-dynamics RL field. The ODRL benchmark represents an important and impactful contribution.
2. Expanding the benchmark to include more real-world or robotic environments could further enhance the significance and impact of the work.
3. The inclusion of a diverse set of baseline algorithms is a strength, and the authors could potentially incorporate even more methods in the future to make the benchmark more comprehensive.

**Strengths:**

1. The clear definition of the off-dynamics RL setting is a strength of the paper, as it provides a solid theoretical foundation for the benchmark.
2. The development of a standardized benchmark for off-dynamics RL is a significant contribution, as it addresses an important gap in the field.
3. The detailed descriptions of the benchmark tasks, including the dynamics modifications, are well-executed and enhance the understanding of the problem.
4. The comprehensive evaluation across the four experimental settings and the insightful observations are valuable contributions that can guide future research.

**Additional Feedback:**

No

**Documentation:**

Yes

**Limitations:**

Yes

**Opportunities For Improvement:**

1. While Figure 3 may be challenging to interpret, the authors could consider alternative visualization methods, such as bar plots or heatmaps, to better convey the performance comparison across the different task categories.
2. Expanding the benchmark to include more real-world or robotic environments could further enhance the significance and impact of the work.
3. The inclusion of a diverse set of baseline algorithms is a strength, and the authors could potentially incorporate even more methods in the future to make the benchmark more comprehensive.

**Relation To Prior Work:**

Yes

**Summary And Contributions:**

This paper introduces ODRL, the first benchmark for off-dynamics reinforcement learning (RL).  The paper formally defines the general off-dynamics RL setting, where the source and target domains differ only in their transition dynamics. The ODRL benchmark offers diverse task categories (locomotion, navigation, dexterous manipulation) and a wide spectrum of dynamics shifts (friction, gravity, kinematic, morphology) under four experimental settings (online-online, offline-online, online-offline, offline-offline). The paper implements various recent off-dynamics RL algorithms and baseline methods within a unified framework. Extensive empirical evaluation of the implemented methods across the ODRL benchmark are given, leading to several key observations.

---

> ### Author Rebuttal · Authors · 2024-08-18
>
> We thank the reviewer for the thoughtful review and positive rating of our work. We appreciate that the reviewer thinks that our benchmark is clear and is of high quality. Please find our clarifications to the concerns below.
>
> **Concern 1: Figure 3 can be improved**
>
> Thanks for catching that. Since different methods have distinct performances under specific dynamics shifts, we agree that it may be kind of difficult to interpret the results in Figure 3. We would use the bar plots or heatmaps (maybe put them alongside the radar chart) as suggested by the reviewer to assist the readers in understanding the figure.
>
> **Concern 2: Expanding the benchmark to include more real-world or robotic environments**
>
> Thanks for your kind suggestion. In our benchmark, the dexterous hand manipulation tasks indeed lie in the category of robotic hand environment. We fully agree that it would be great to expand the benchmark to include more real-world environments. We are proud to announce that we have included a new set of tasks that leverage the Sawyer robot to complete various tasks. These tasks are adapted from Meta-World [1]. The Sawyer robot is widely used in academic or real-world robotics research. Note that we do not consider the multi-task setting, but still the off-dynamics setting, where the tasks are identical and only dynamics between the source domain and the target domain vary. It costs us some time to fully understand the `xml` files of the sawyer robots in Meta-World and make our own modifications. We include two kinds of dynamics shifts akin to Adroit robots, kinematic shifts (easy/medium/hard) and morphology shifts (easy/medium/hard). Please check our codes for more details (since the Sawyer tasks are still under development, we put them in a new branch, https://github.com/OffDynamicsRL/off-dynamics-rl/tree/test). We also attach the visualizations of our modified Sawyer robot tasks to the **rebuttal pdf** below. It would be great if the reviewer could check that. Please let us know if you have any suggestions for robotic environments!
>
> **[NOTE: the Sawyer robot environments are still under development, please allow us some time to fully examine all environments we propose to ensure that they are solvable and the corresponding offline datasets are provided.]**
>
> [1] Meta-World: A Benchmark and Evaluation for Multi-Task and Meta Reinforcement Learning
>
> **Concern 3: Incorporate even more methods in the future**
>
> Yes, as we wrote in the main text (line 327), *our benchmark is promised to be actively maintained*. This, of course, includes implementing more recent off-dynamics RL methods to make our benchmark more comprehensive.
>
> If there are still some remaining concerns, we are happy to have further discussions with the reviewer.

---

> > ### Author Response · Authors · 2024-08-28
> > **Following up with Reviewer ciZk**
> >
> > Dear Reviewer ciZk, thanks for your thoughtful review. It would be great if you could give us some comments on the new Sawyer tasks (please check https://github.com/OffDynamicsRL/off-dynamics-rl/tree/test). We promise to include all your suggestions in the revision. We are more than happy to answer new questions if any arise.
> >
> > The authors

---

### Decision · Program_Chairs · 2024-09-26

**Decision:**

Accept (Poster)

**Comment:**

Great paper!  It describes a new benchmark for off-dynamics generalization in RL.  There doesn't exist a centralized benchmark at the moment.  This paper fulfills this gap. The empirical evaluation is thorough and comprehensive.  This benchmarks will help future research to advance in terms of generalization across RL domains with different dynamics.  The authors addressed well the reviewers' concerns in the rebuttal.